# When can transformers reason with abstract symbols?

**Enric Boix-Adserà\***
Apple, MIT
eboix@mit.edu

**Omid Saremi**
Apple
osaremi@apple.com

**Emmanuel Abbe**
Apple, EPFL
emmanuel.abbe@epfl.ch

**Samy Bengio**
Apple
bengio@apple.com

**Etai Littwin**
Apple
elittwin@apple.com

**Joshua Susskind**
Apple
jsusskind@apple.com

## Abstract

We investigate the capabilities of transformer models on *relational reasoning* tasks. In these tasks, models are trained on a set of strings encoding abstract relations, and are then tested out-of-distribution on data that contains symbols that did not appear in the training dataset. We prove that for any relational reasoning task in a large family of tasks, transformers learn the abstract relations and generalize to the test set when trained by gradient descent on sufficiently large quantities of training data. This is in contrast to classical fully-connected networks, which we prove fail to learn to reason. Our results inspire modifications of the transformer architecture that add only two trainable parameters per head, and that we empirically demonstrate improve data efficiency for learning to reason.

## 1 Introduction

As large language models (LLMs) are trained with increasing quantities of data, they begin to exhibit the ability to reason mathematically (Kaplan et al., 2020; Yuan et al., 2023). Why does more data help an LLM learn to reason? And can we make LLMs more data-efficient at learning to reason?

In this paper, we study *relational reasoning* with abstract symbols, which is a basic capability that has been hypothesized to underlie more complex abilities in human cognition (Fodor, 1975; Newell, 1980; Snow et al., 1984; Marcus, 1998; Holyoak, 2012; Kriete et al., 2013; Webb et al., 2020b). One example is in mathematics or computer science, where relational reasoning is necessary to parse a proof or a program: variable names are abstract symbols and the functionality of the proof or program only depends on how they relate to each other and not on the variable names themselves.

Our contributions are threefold: (i) we formalize relational reasoning through "template tasks"; (ii) we conduct an analysis of when transformers can learn template tasks when trained by gradient descent and show a separation with classical fully-connected neural network architectures; (iii) we propose modifications to transformers that improve data efficiency for learning to reason.

### 1.1 Capturing relational reasoning with template tasks

Building on a line of work in neuroscience (Marcus, 1998; Martinho III & Kacelnik, 2016; Kim et al., 2018; Webb et al., 2020b; Kerg et al., 2022; Altabaa et al., 2023; Webb et al., 2023a; Geiger et al., 2023), we formalize a framework of reasoning tasks called *template tasks*.

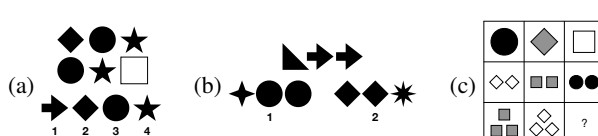

(a) (b) (c)

Figure 1: Tasks from Raven (1938); Webb et al. (2020b) which fall under our theory. Networks are trained with one alphabet of symbols and then tested on held-out symbols. Details in Appendix A.

**Regression setting** In the regression setting, a template task is specified by a collection of "template" strings labeled by real numbers, which are used to generate the train and test data. The simplest way to describe these is through an example. Consider, for instance, the templates

$$\text{``}\alpha\text{=1;}\beta\text{=-1;}\texttt{print}(\alpha)\text{''} \rightarrow \text{label=+1} \quad \text{and} \quad \text{``}\alpha\text{=1;}\beta\text{=-1;}\texttt{print}(\beta)\text{''} \rightarrow \text{label=-1.} \quad (1)$$

These are used to generate the datasets in Figure 2, where every sample $(\boldsymbol{x}_i, y_i) \in \mathcal{X}^k \times \mathcal{Y}$ is formed by picking a template and replacing the placeholders $\alpha, \beta$ (which we call "wildcards") with variable names. Memorizing the training data is easy (Zhang et al., 2021a), but we wish to measure reasoning: will the model learn to treat the variable names as abstract symbols, enabling generalization beyond its training distribution? To evaluate this, we adopt an out-of-distribution setting, where the train and test data distributions differ (Marcus, 1998; Abbe et al., 2023). The test dataset consists of the same programs, but with *new variable names never seen during training*. By *testing on symbols unseen in the train set*, we measure the ability of an LLM to learn logical rules on the relations between symbols. To succeed, the LLM must effectively infer the templates from training data, and at test time match samples to the corresponding templates to derive their labels.

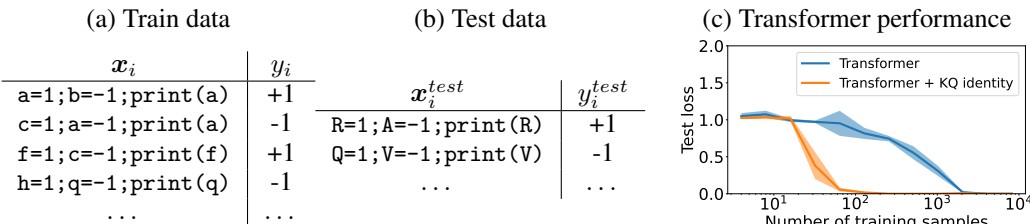

| (a) Train data | | (b) Test data | | (c) Transformer performance |
|---|---|---|---|---|

| $\boldsymbol{x}_i$ | $y_i$ |
|---|---|
| `a=1;b=-1;print(a)` | +1 |
| `c=1;a=-1;print(a)` | -1 |
| `f=1;c=-1;print(f)` | +1 |
| `h=1;q=-1;print(q)` | -1 |
| ... | ... |

| $\boldsymbol{x}_i^{test}$ | $y_i^{test}$ |
|---|---|
| `R=1;A=-1;print(R)` | +1 |
| `Q=1;V=-1;print(V)` | -1 |
| ... | ... |

Figure 2: (a,b) Variable names in the test data never appear in the train data (indicated by lower/upper-case names). (c) Remarkably, as the training set size increases, the LLM's ability to reason outside of its training data improves, as it learns to use the relations between the variable names to classify, instead of simply memorizing the training data. Our theory motivates a modified transformer architecture (see Observation 1.2), which solves the reasoning task with less training data. Details in Appendix A.

Apart from programming tasks as in Figure 2, this framework captures several natural problems:

- *Same/different task*. The simplest relational reasoning task is when the templates are "$\alpha\alpha$" and "$\alpha\beta$" labeled by +1 and −1. This encodes learning to classify two symbols as equal (e.g., $AA$, $BB$) or as distinct (e.g., $AB$, $BC$), even when the symbols were unseen in the training data. This task has been studied empirically in animal behavior (Martinho III & Kacelnik, 2016) and in neural networks (Kim et al., 2018; Webb et al., 2020b).
- *Word problems*. Word problems often have building blocks that follow simple templates. For example, the template "If $\alpha$ gives $\beta$ 5 $\gamma$, how many $\gamma$ does $\beta$ have?" labeled by +5, could generate the data "If Alice gives Bob 5 oranges, how many oranges does Bob have?" or the data "If Rob gives Ada 5 apples, how many apples does Ada have?"
- *Psychometric tests*. Psychometric tests of relational reasoning, which have recently been used to probe LLMs (Raven, 1938; Webb et al., 2020b; Altabaa et al., 2023; Kerg et al., 2022; Webb et al., 2023a;b), are often template tasks. Figure 1 illustrates some examples.

**Next-token-prediction setting** In the next-token-prediction setting, there is one extra layer of complexity: each sample is labeled with a symbol. For the LLM to generalize to symbols unseen at train time, not only must it learn to track the value stored in a variable, but it also must learn to predict labels at test time that might not occur in its training data. For example, the train and test datasets in Figure 3 are generated by:

$$\text{``}\alpha\text{=}\text{"}\gamma\text{";}\beta\text{=}\text{"}\delta\text{";}\texttt{print}(\alpha)\text{''} \rightarrow \text{label=}\gamma \quad \text{and} \quad \text{``}\alpha\text{=}\text{"}\gamma\text{";}\beta\text{=}\text{"}\delta\text{";}\texttt{print}(\beta)\text{''} \rightarrow \text{label=}\delta, \quad (2)$$

where $\alpha, \beta, \gamma, \delta$ are wildcards. Other problems covered by these tasks include:

- *Programming*. The template "`print("`$\alpha$`")`" labeled with $\alpha$ generates (`print("A")`,`A`) or (`print("dog")`,`dog`), and so an LLM that learns on the corresponding task can robustly evaluating print statements on symbols not seen in the training data.

- *Mathematical functions.* For example, the set of templates $\{\alpha\alpha\alpha, \alpha\beta\alpha, \alpha\alpha\beta, \beta\alpha\alpha\}$ labeled by $\alpha$ encode the task of outputting the majority token in a length-3 string with a vocabulary of two symbols. Similarly, for length-$k$ strings, the task of outputting the majority element can be encoded with $2^{k-1}$ templates.

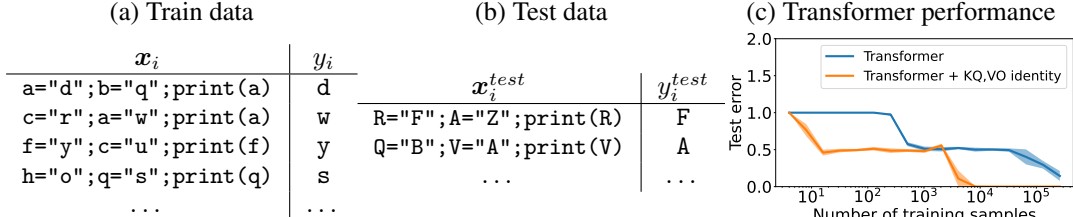

| (a) Train data | | | (b) Test data | | (c) Transformer performance |
| --- | --- | --- | --- | --- | --- |

| $\boldsymbol{x}_i$ | $y_i$ |
| --- | --- |
| a="d";b="q";print(a) | d |
| c="r";a="w";print(a) | w |
| f="y";c="u";print(f) | y |
| h="o";q="s";print(q) | s |
| ... | ... |

| $\boldsymbol{x}_i^{test}$ | $y_i^{test}$ |
| --- | --- |
| R="F";A="Z";print(R) | F |
| Q="B";V="A";print(V) | A |
| ... | ... |

Figure 3: (a,b) The labels are symbols. (c) We propose a modified that transformer learns the reasoning task with less data (see Observation 1.2 and Theorem 1.4). Details in Appendix A.

## 1.2 MAIN RESULTS

The phenomenon from Figures 2 and 3 that we seek to understand is: why does the out-of-distribution performance of the transformer architecture improve as the number of training samples increases? We analyze the regression and next-token-prediction settings separately.

**(1) MLPs fail to generalize to unseen symbols** A classical criticism of connectionism by Marcus (1998) is that neural networks do not learn relational reasoning when trained. We support this criticism in Appendix I by proving that classical MLP architectures (a.k.a. fully-connected networks) trained by SGD or Adam will not generalize in template tasks on symbols unseen during training, even in the regression setting. This failure to reason relationally occurs regardless of the training data size. The proof uses a permutation equivariance property of MLP training (Ng, 2004; Shamir, 2018; Li et al., 2020; Abbe et al., 2022; Abbe & Boix-Adsera, 2022).

**(2) Transformers generalize to unseen symbols, but require large data diversity** Nevertheless, we prove that he criticism of Marcus (1998) is not valid for modern transformer architectures (Vaswani et al., 2017). We analyze the training dynamics of a transformer model and establish that it can learn to reason relationally:

**Theorem 1.1** (Informal Theorem 3.4). *For any regression template task, a wide-enough transformer architecture trained by gradient flow on sufficiently many samples generalizes on unseen symbols.*

Here the key points are: *(a) Universality.* The transformer architecture generalizes on symbols unseen in train data regardless of which and how many templates are used to define the reasoning task. *(b) Large enough number of samples.* Our theoretical guarantees require the training dataset size to be large, and even for very basic tasks like the two-template task in Figure 2, good generalization begins to occur only at a very large number of training samples considering the simplicity of the task. This raises the question of how the inductive bias of the transformer can be improved.

The proof of Theorem 1.1 inspires a parametrization modification that empirically lowers the quantity of data needed by an order of magnitude. A standard transformer attention head that takes in an input $\boldsymbol{X} \in \mathbb{R}^{k \times d_{emb}}$ is given by

$$\mathrm{smax}(\boldsymbol{X}\boldsymbol{W}_K\boldsymbol{W}_Q^T\boldsymbol{X}^T)\boldsymbol{X}\boldsymbol{W}_V\boldsymbol{W}_O^T, \tag{3}$$

where $\boldsymbol{W}_K, \boldsymbol{W}_Q, \boldsymbol{W}_V, \boldsymbol{W}_O$ are trainable parameters. Our modification makes it easier for the transformer to access the incidence matrix $\boldsymbol{X}\boldsymbol{X}^T \in \mathbb{R}^{k \times k}$ of the input, which is invariant to permutations of the symbol alphabet and can be used to solve the relational reasoning task:

**Observation 1.2.** *Adding one trainable parameter $a$ to each attention head so that $\boldsymbol{W}_K\boldsymbol{W}_Q^T$ is replaced by $\boldsymbol{W}_K\boldsymbol{W}_Q^T + a\boldsymbol{I}$ improves transformers' data-efficiency on template tasks.*

**(3) Transformers fail at copying unseen symbols** The story is slightly different for next-token-prediction tasks, because of the bottleneck of learning to output a symbol that was never seen in the training dataset. Transformers' performance degrades as the model grows (an "inverse scaling" law (McKenzie et al., 2023)). Large transformers fail even for the task of copying the input.

**Theorem 1.3** (Informal Theorem 4.1). *Transformers with large embedding dimension fail to generalize on unseen symbols for the copy-task outputting label "$\alpha$" on template "$\alpha$".*

However, we propose adding an *attention-modulated skip connection*, which corrects this failure, making it easy for the transformer to learn to copy data between its residual streams:

**Theorem 1.4** (Informal Theorem 4.2). *Adding one trainable parameter $b$ to each head so that $\boldsymbol{W}_V \boldsymbol{W}_O^T$ is replaced by $\boldsymbol{W}_V \boldsymbol{W}_O^T + b\boldsymbol{I}$ makes transformers generalize on the task of Theorem 1.3.*

**(4) Experiments** We conclude with experimental validation of our architecture modifications, and find that they improve data efficiency on relational reasoning tasks by an order of magnitude, and improve language-modeling performance when training the GPT-2 architecture on Wikitext.

## 1.3 RELATED LITERATURE

A spate of recent work studies whether and how LLMs perform various reasoning tasks, each focusing on one component of reasoning: these include recognizing context-free grammars (Zhao et al., 2023; Allen-Zhu & Li, 2023), learning sparse functions (Edelman et al., 2022), learning compositionally (Hupkes et al., 2020), generalizing out-of-distribution when learning Boolean functions (Abbe et al., 2023), performing arithmetic (Nanda et al., 2023), learning in context (Garg et al., 2022; Ahn et al., 2023; Zhang et al., 2023), and evaluating indexing (Zhang et al., 2021b). Our setting is closest to that of empirical work studying neural networks on relational reasoning tasks (Geiger et al., 2023; Webb et al., 2023b). For example, the four tasks in Webb et al. (2020b), the matrix digits task in Webb et al. (2023a), the SET game task in Altabaa et al. (2023), and most of the tasks in Kerg et al. (2022) (with the exception of the relational games tasks), are examples of regression template tasks that fall under our theory. Furthermore, Kim et al. (2018) shows experimentally that MLPs fail on the same/different template task, and we provide a proof for this in Appendix I. There is also a literature on modifying training to improve relational reasoning: (Webb et al., 2020a) proposes applying Temporal Context Normalization during training, and Santoro et al. (2017; 2018); Palm et al. (2018); Shanahan et al. (2020); Webb et al. (2020b); Kerg et al. (2022); Altabaa et al. (2023) propose new architectures. Finally, some recent works in mechanistic interpretability look for subnetworks within trained networks that are responsible for tasks such as variable binding (Olsson et al., 2022; Davies et al., 2023). In contrast, our focus is on proving when the transformer architecture learns or fails to learn, and on applying this theoretical understanding to improve its data efficiency for relational reasoning.

## 2 FORMAL DEFINITION OF TEMPLATE TASKS

We formally define *regression* template tasks. For next-token prediction, see Appendix J.

**Definition 2.1.** A **template** is a string $\boldsymbol{z} \in (\mathcal{X} \cup \mathcal{W})^k$, where $\mathcal{X}$ is an alphabet of tokens, and $\mathcal{W}$ is an alphabet of "wildcards". A **substitution map** is an injective function $s : \mathcal{W} \to \mathcal{X}$. We write $\mathrm{sub}(\boldsymbol{z}, s) \in \mathcal{X}^k$ for the string where each wildcard is substituted with the corresponding token: $\mathrm{sub}(\boldsymbol{z}, s)_i = z_i$ if $z_i \in \mathcal{X}$, and $\mathrm{sub}(\boldsymbol{z}, s)_i = s(z_i)$ if $z_i \in \mathcal{W}$. The string $\boldsymbol{x} \in \mathcal{X}^k$ **matches** the template $\boldsymbol{z}$ if $\boldsymbol{x} = \mathrm{sub}(\boldsymbol{z}, s)$ for some substitution map $s$ and also $s(\mathcal{W}) \cap \{z_i\}_{i \in [k]} = \emptyset$: i.e., the substituted tokens did not already appear in the template $\boldsymbol{z}$.

**Example** Using Greek letters to denote the wildcards and Latin letters to denote regular tokens, the template "$\alpha\alpha\beta ST$" matches the string "QQRST", but *not* "QQQST" (because the substitution map is not injective) and *not* "QQSST" (because $\beta$ is replaced by S which is already in the template).

A template task's training data distribution is generated by picking a template randomly from a distribution, and substituting its wildcards with a random substitution map.

**Definition 2.2.** A template data distribution $\mathcal{D} = \mathcal{D}(\mu_{\mathsf{tmplt}}, \{\mu_{sub,\boldsymbol{z}}\}_{\boldsymbol{z}}, f_*, \sigma)$ is given by

- a template distribution $\mu_{\text{tmplt}}$ supported on templates in $(\mathcal{X} \cup \mathcal{W})^k$,

- for each $\boldsymbol{z} \in \text{supp}(\mu_{\text{tmplt}})$, a distribution $\mu_{sub,\boldsymbol{z}}$ over substitution maps $s : \mathcal{W} \to \mathcal{X}$,

- template labelling function $f_* : \text{supp}(\mu_{\text{tmplt}}) \to \mathbb{R}$, and a label-noise parameter $\sigma \geq 0$.

We draw a sample $(\boldsymbol{x}, y) = (\text{sub}(\boldsymbol{z}, s), f_*(\boldsymbol{z}) + \xi) \sim \mathcal{D}$, by drawing a template $\boldsymbol{z} \sim \mu_{\text{tmplt}}$, a substitution map $s \sim \mu_{sub,\boldsymbol{z}}$, and label noise $\xi \sim \mathcal{N}(0, \sigma^2)$.

Finally, we define what it means for a model to solve the template task and generalize on unseen symbols; namely, the model should output the the correct label for any string $\boldsymbol{x} \in \mathcal{X}^k$ matching a template, regardless of whether the string is in the support of the training distribution.

**Definition 2.3.** A (random) estimator $\hat{f} : \mathcal{X}^k \to \mathbb{R}$ **generalizes on unseen symbols** with $(\epsilon, \delta)$-error if the following is true. For any $\boldsymbol{x} \in \mathcal{X}^k$ that matches a template $\boldsymbol{z} \in \text{supp}(\mu_{\text{tmplt}})$, we have

$$(\hat{f}(\boldsymbol{x}) - f_*(\boldsymbol{z}))^2 \leq \epsilon,$$

with probability at least $1 - \delta$ over the randomness of the estimator $\hat{f}$.

**Example** If the training data is generated from a uniform distribution on templates "$\alpha\alpha$" with label 1 and "$\alpha\beta$" for label -1, then it might consist of the data samples $\{(AA, 1), (BB, 1), (AB, -1), (BA, -1)\}$. An estimator that generalizes to unseen symbols must correctly label string $CC$ with $+1$ and string $CD$ with $-1$, even though these strings consist of symbols that do not appear in the training set. This is a nontrivial reasoning task since it requires learning to use the *relations* between the symbols to classify rather than the identities of the symbols.

## 3 ANALYSIS FOR TEMPLATE TASKS IN THE REGRESSION SETTING

We establish that one-layer transformers of large enough width generalize to unseen symbols, when trained with enough data on regression template tasks. It is important to note that this is not true for all architectures, as we prove in Appendix I that MLPs trained by SGD or Adam will not succeed.

### 3.1 TRANSFORMER RANDOM FEATURES KERNEL

The one-layer transformer architecture that we analyze consists of an embedding layer, a multihead attention mechanism, an MLP layer, and an unembedding layer $\boldsymbol{w}_U$. This is written mathematically in Appendix H. We analyze training only the final $\boldsymbol{w}_U$ layer of the transformer, keeping the other weights fixed at their random Gaussian initialization. Surprisingly, even though we only train the final layer of the transformer, this is enough to guarantee generalization on unseen symbols. Taking the width and embedding and head dimensions to infinity, and the step size to 0, the SGD training algorithm with weight decay converges to kernel gradient flow with the following kernel $K_{\text{trans}}$ in the infinitely-wide, infinitely-small-step-size limit. Here and throughout the remainder of the paper, we interchangeably denote an input by a string $\boldsymbol{x} \in \mathcal{X}^k$ or a matrix $\boldsymbol{X} \in \mathbb{R}^{k \times m}$ constructed by stacking the one-hot vectors $\boldsymbol{X} = [\boldsymbol{e}_{x_1}, \ldots, \boldsymbol{e}_{x_k}]^T$ of the string's tokens. $\phi : \mathbb{R} \to \mathbb{R}$ is the MLP activation layer, $\beta, \gamma \in \mathbb{R}$ are hyperparameters controlling the temperature and magnitude of positional activations.

$$K_{\text{trans}}(\boldsymbol{X}, \boldsymbol{Y}) = \mathbb{E}_{u,v}[\phi(u)\phi(v)] \text{ for } u, v \sim N(\boldsymbol{0}, \begin{bmatrix} K_{\text{attn}}(\boldsymbol{X}, \boldsymbol{X}) & K_{\text{attn}}(\boldsymbol{X}, \boldsymbol{Y}) \\ K_{\text{attn}}(\boldsymbol{Y}, \boldsymbol{X}) & K_{\text{attn}}(\boldsymbol{Y}, \boldsymbol{Y}) \end{bmatrix}) \quad (4)$$

where $K_{\text{attn}}(\boldsymbol{X}, \boldsymbol{Y}) = \mathbb{E}_{\boldsymbol{m}(\boldsymbol{X}), \boldsymbol{m}(\boldsymbol{Y})}[\text{smax}(\beta\boldsymbol{m}(\boldsymbol{X}))^T (\boldsymbol{X}\boldsymbol{Y}^T + \gamma^2 \boldsymbol{I})\text{smax}(\beta\boldsymbol{m}(\boldsymbol{Y}))]$

$$[\boldsymbol{m}(\boldsymbol{X}), \boldsymbol{m}(\boldsymbol{Y})] \sim N(\boldsymbol{0}, \begin{bmatrix} \boldsymbol{X}\boldsymbol{X}^T + \gamma^2 \boldsymbol{I} & \boldsymbol{X}\boldsymbol{Y}^T + \gamma^2 \boldsymbol{I} \\ \boldsymbol{Y}\boldsymbol{X}^T + \gamma^2 \boldsymbol{I} & \boldsymbol{Y}\boldsymbol{Y}^T + \gamma^2 \boldsymbol{I} \end{bmatrix}).$$

The function outputted by kernel gradient flow is known to have a closed-form solution in terms of the samples, the kernel, and the weight-decay parameter $\lambda$, which we recall in Proposition 3.1.

**Proposition 3.1** (How kernel gradient flow generalizes; see e.g., (Welling, 2013).). *Let* $(\boldsymbol{X}_1, y_1), \ldots, (\boldsymbol{X}_n, y_n)$ *be training samples. With the square loss and ridge-regularization of magnitude $\lambda$, kernel gradient flow with kernel $K$ converges to the following solution*

$$\hat{f}(\boldsymbol{X}) = \boldsymbol{y}^T(\hat{\boldsymbol{K}} + \lambda \boldsymbol{I})^{-1}\boldsymbol{k}(\boldsymbol{X}), \quad (5)$$

where $\boldsymbol{y} = [y_1, \ldots, y_n] \in \mathbb{R}^n$ are the train labels, $\hat{\boldsymbol{K}} \in \mathbb{R}^{n \times n}$ is the empirical kernel matrix and has entries $\hat{K}_{ij} = K(\boldsymbol{X}_i, \boldsymbol{X}_j)$, and $\boldsymbol{k}(\boldsymbol{X}) \in \mathbb{R}^n$ has entries $k_i(\boldsymbol{X}) = K(\boldsymbol{X}_i, \boldsymbol{X})$.

## 3.2 Transformers generalize on unseen symbols

We prove that transformers will generalize out-of-distribution on unseen symbols when trained on template tasks. We require the templates in the distribution $\mu_{\mathsf{tmplt}}$ to be "disjoint", since otherwise the correct label for a string $\boldsymbol{x}$ is not uniquely defined, as $\boldsymbol{x}$ could match more than one template:

**Definition 3.2.** Two templates $\boldsymbol{z}, \boldsymbol{z}' \in (\mathcal{X} \cup \mathcal{W})^k$ are **disjoint** if no $\boldsymbol{x} \in \mathcal{X}^k$ matches both $\boldsymbol{z}$ and $\boldsymbol{z}'$.

Furthermore, in order to ensure that the samples are not all copies of each other (which would not help generalization), we have to impose a diversity condition on the data.

**Definition 3.3.** The **data diversity** is measured by $\rho = \min_{\boldsymbol{z} \in \mathrm{supp}(\mu_{\mathsf{tmplt}})} \min_{t \in \mathcal{X}} \frac{1}{\mathbb{P}_{s \sim \mu_{sub,\boldsymbol{z}}}[t \in s(\mathcal{W})]}$.

When the data diversity $\rho$ is large, then no token is much more likely than others to be substituted. If $\rho$ is on the order of the number of samples $n$, then most pairs of data samples will not be equal.

**Theorem 3.4** (Transformers generalize on unseen symbols). *Let $\mu_{\mathsf{tmplt}}$ be supported on a finite set of pairwise-disjoint templates ending with [CLS] tokens. Then, for almost any $\beta, \gamma, b_1, b_2$ parameters (except for a Lebesgue-measure-zero set), the transformer random features with $\phi(t) = \cos(b_1 t + b_2)$ generalizes on unseen symbols.[1] Formally, there are constants $c, C > 0$ and ridge regularization parameter $\lambda > 0$ that depend only $\beta, \gamma, b_1, b_2, \mu_{\mathsf{tmplt}}, f_*, \sigma$, such that for any $\boldsymbol{x}$ matching a template $\boldsymbol{z} \in \mathrm{supp}(\mu_{\mathsf{tmplt}})$ the kernel ridge regression estimator $\hat{f}$ in (5) with kernel $K_{\mathsf{trans}}$ satisfies*

$$|\hat{f}(\boldsymbol{x}) - f_*(\boldsymbol{z})| \leq C\sqrt{\log(1/\delta)/n} + C\sqrt{1/\rho},$$

*with probability at least $1 - \delta - \exp(-cn)$ over the random samples.*

The first term is due to the possible noise in the labels. The second term quantifies the amount of sample diversity in the data. Both the sample diversity and the number of samples must tend to infinity for an arbitrarily small error guarantee.

**Proof sketch** (1) In Lemma 3.5 we establish with a sufficient condition for kernel ridge regression to generalize on unseen symbols. (2) We prove that $K_{\mathsf{trans}}$ satisfies it.

*(1) Sufficient condition.* Let $\mu_{\mathsf{tmplt}}$ be supported on templates $\boldsymbol{z}_1, \ldots, \boldsymbol{z}_r$. Let $\mathcal{R} = \cup_{i \in [k], j \in [r]} \{z_{j,i}\}$ be the tokens that appear in the templates. Let $[n] = \mathcal{I}_1 \sqcup \mathcal{I}_2 \sqcup \cdots \sqcup \mathcal{I}_n$ be the partition of the samples such that if $a \in \mathcal{I}_j$ then sample $(\boldsymbol{x}_a, y_a)$ is drawn by substituting the wildcards of template $\boldsymbol{z}_j$. Two samples $\boldsymbol{x}_a, \boldsymbol{x}_b$ that are drawn from the same template $\boldsymbol{z}_j$ may be far apart as measured by the kernel: i.e., the kernel inner product $K(\boldsymbol{x}_a, \boldsymbol{x}_b)$ may be small. However, these samples will have similar relationship to most other samples:

$$K(\boldsymbol{x}_a, \boldsymbol{x}_i) = K(\boldsymbol{x}_b, \boldsymbol{x}_i) \quad \text{for most } i \in [n]. \tag{6}$$

Specifically, if the wildcards of $\boldsymbol{x}_a, \boldsymbol{x}_b$ and $\boldsymbol{x}_i$ are substituted by disjoint sets of tokens that do not appear in the templates, then (6) holds. Therefore, as the sample diversity $\rho$ increases, the empirical kernel matrix $\hat{\boldsymbol{K}}$ becomes approximately block-structured with blocks $\mathcal{I}_j \times \mathcal{I}_{j'}$. For most samples $\boldsymbol{x}_a, \boldsymbol{x}_b$ corresponding to template $\boldsymbol{z}_j$, and most $\boldsymbol{x}_{a'}, \boldsymbol{x}_{b'}$ corresponding to template $\boldsymbol{z}_{j'}$ we have

$$K(\boldsymbol{x}_a, \boldsymbol{x}_{a'}) = K(\boldsymbol{x}_b, \boldsymbol{x}_{b'}) = K(\mathrm{sub}(\boldsymbol{z}_j, s), \mathrm{sub}(\boldsymbol{z}_{j'}, s')) := N_{j,j'}, \tag{7}$$

where $s, s' : \mathcal{W} \to \mathcal{X}$ are substitution maps satisfying

$$s(\mathcal{W}) \cap s'(\mathcal{W}) = 0 \quad \text{and} \quad s(\mathcal{W}) \cap \mathcal{R} = s'(\mathcal{W}) \cap \mathcal{R} = \emptyset. \tag{8}$$

One can check that (7) and (8) uniquely define a matrix $\boldsymbol{N} \in \mathbb{R}^{r \times r}$ which gives the entries in the blocks of $\hat{\boldsymbol{K}}$, with one block for each pair of templates.[2] See Figure 4.

---

[1] We analyze the shifted and rescaled cosine activation function $\phi(t) = \cos(b_1 t + b_2)$ out of technical convenience, but conjecture that most non-polynomial activation functions should succeed.

[2] This assumes a "token-symmetry" property of $K$ that is satisfied by transformers; details in the full proof.

Figure 4: Illustration of structure of $\hat{K}$ and $N$ for the same/different task, which has $r = 2$ templates $z_1 = \alpha\alpha$ and $z_2 = \alpha\beta$. As the sample diversity $\rho$ increases and the number of samples $n$ increases, the empirical kernel matrix $\hat{K} \in \mathbb{R}^{n \times n}$ becomes approximately $(r \times r)$-block-structured, and within each block most of the entries are given by $N \in \mathbb{R}^{r \times r}$; exceptions where this is not true, including the diagonals, are drawn in black. Furthermore, the spectrum of $\hat{K}$ is increasingly determined by the spectrum of $N$, and if $N$ is nonsingular then the top eigenspace increasingly aligns with the span of the indicator vectors on $\mathcal{I}_1, \ldots, \mathcal{I}_r$.

If the matrix $N$ is nonsingular and the number of samples is large, then the span of the top $r$ eigenvectors of $\hat{K}$ will align with the span of the indicator vectors on the sets $\mathcal{I}_1, \ldots, \mathcal{I}_r$. Furthermore, when testing a string $x^{test}$ that matches template $z_j$, but might not have appeared in the training set, it holds that for most $a \in \mathcal{I}_j$, we have

$$k(x^{test}) = [K(x^{test}, x_1), \ldots, K(x^{test}, x_n)] \approx [K(x_a, x_1), \ldots, K(x_a, x_n)] = \hat{K}_{a,:} \,.$$

In words, the similarity relationship of $x^{test}$ to the training samples is approximately the same as the similarity relationship of $x_a$ to the training samples. So the kernel ridge regression solution (5) approximately equals the average of the labels of the samples corresponding to template $z_j$, which in turn is approximately equal to the template label by a Chernoff bound,

$$y^T(\hat{K} + \lambda I)^{-1} k(x^{test}) \approx \frac{1}{|\mathcal{I}_j|} \sum_{a \in \mathcal{I}_j} y_i \approx f_*(z_j) \,. \tag{9}$$

Therefore, kernel ridge regression generalizes on $x^{test}$. It is important to note that the number of samples needed until (9) is a good approximation depends on the nonsingularity of $N$. This yields the sufficient condition for kernel ridge regression to succeed (proof in Appendix C).

**Lemma 3.5** (Informal Lemma C.3). *If $N$ is nonsingular, then* (5) *generalizes to unseen symbols.*

*(2) $K_{\text{trans}}$ satisfies the sufficient condition.* We now show that for *any* collection of disjoint templates $z_1, \ldots, z_r$, the matrix $N_{\text{trans}} := N \in \mathbb{R}^{r \times r}$ defined with kernel $K = K_{\text{trans}}$ is nonsingular. The challenging is that $K_{\text{trans}}$ does not have a closed-form solution because of the expectation over softmax terms in its definition (4). Therefore, our analysis of the transformer random feature kernel is, to the best of our knowledge, the first theoretical analysis showing that the transformer random features learn a nontrival class of functions of sequences. We proceed by analyzing the MLP layer and the attention layer separately, observing that a"weak" condition on $K_{\text{attn}}$ can be lifted into the "strong" result that $N_{\text{trans}}$ is nonsingular. The intuition is that as long as $K_{\text{attn}}$ is not a very degenerate kernel, it is unlikely that the MLP layer has the cancellations that to make $N_{\text{trans}}$ nonsingular.

**Lemma 3.6** (Nonsingularity of $N_{\text{trans}}$). *Suppose for every non-identity permutation $\tau \in S_r \setminus \{\text{id}\}$,*

$$\sum_{i \in [r]} K_{\text{attn}}(\text{sub}(z_i, s), \text{sub}(z_i, s')) \neq \sum_{i \in [r]} K_{\text{attn}}(\text{sub}(z_i, s), \text{sub}(z_{\tau(i)}, s')) \,, \tag{10}$$

*where $s, s'$ are the substitution maps in the definition of $N_{\text{trans}}$ in* (8). *Let the MLP layer's activation function be $\phi(t) = \cos(b_1 t + b_2)$. Then for almost any choice of $b_1, b_2$ (except for a Lebesgue-measure-zero set), the matrix $N_{\text{trans}}$ is nonsingular.*

This is proved in Appendix E, by evaluating a Gaussian integral and showing $N_{\text{trans}}$ has Vandermonde structure. Although we use the cosine activation function, we conjecture that this result holds for most non-polynomial activation functions. Next, we prove the condition on $N_{\text{attn}}$.

**Lemma 3.7** (Non-degeneracy of $K_{\text{attn}}$). *The condition* (10) *holds for Lebesgue-almost any $\beta, \gamma$.*

The proof is in Appendix F. First, we prove the analyticity of the kernel $K_{\text{attn}}$ in terms of the hyperparameters $\beta$ and $\gamma$. Because of the identity theorem for analytic functions, it suffices to show at least one choice of hyperparameters $\beta$ and $\gamma$ satisfies (10) for all non-identity permutations $\tau$. Since $K_{\text{attn}}$ does not have a closed-form solution, we find such a choice of $\beta$ and $\gamma$ by analyzing the Taylor-series expansion of $K_{\text{attn}}$ around $\beta = 0$ and $\gamma = 0$ up to order-10 derivatives.

### 3.3 Improving transformer data-efficiency with $W_K W_Q^T + aI$ parametrization

Can we use these insights to improve transformers' data-efficiency in template tasks? In the proof, the nonsingularity of $N$ in Lemma 3.5 drives the model's generalization on unseen symbols. This suggests that an approach to improve data-efficiency is to make $N$ better-conditioned by modifying the transformer parametrization. We consider here the simplest task, with templates "$\alpha\alpha$" and "$\alpha\beta$" labeled with $+1$ and $-1$, respectively. For tokens $A, B, C, D \in \mathcal{X}$, the matrix $N$ is

$$N = \begin{bmatrix} K(AA, BB) & K(AA, BC) \\ K(BC, AA) & K(AB, CD) \end{bmatrix}$$

If $K$ is an inner-product kernel, $K(\boldsymbol{x}, \boldsymbol{x}') = \kappa(\sum_{i \in [k]} 1(x_i = x_i'))$, as from an MLP, then $K(AA, BB) = K(AA, BC) = K(BC, AA) = K(AB, CD) = \kappa(0)$, so $N$ is singular and generalization is not achieved. Intuitively, every sample $\boldsymbol{x}_i$ has approximately the same "similarity profile to other data" $\hat{\boldsymbol{K}}_{i,:} = [K(\boldsymbol{x}_i, \boldsymbol{x}_1), \dots, K(\boldsymbol{x}_i, \boldsymbol{x}_n)]$, so the kernel method cannot identify the samples that come from the same template as $\boldsymbol{x}^{test}$. In contrast, the transformer kernel (4) succeeds by using information about the incidence matrix $\boldsymbol{X}\boldsymbol{X}^T$, which differs between templates, and does not depend on the symbol substitution. We thus propose to emphasize the incidence matrix $\boldsymbol{X}\boldsymbol{X}^T$ by reparametrizing each head to $\boldsymbol{W}_K \boldsymbol{W}_Q^T + a\boldsymbol{I}$, where $a$ is a trainable parameter. This adds a scaling of $\boldsymbol{X}\boldsymbol{X}^T$ in the attention, and can empirically improve data efficiency by an order of magnitude on several template tasks (see Figures 2 and 3, as well as additional experiments in Appendix B).

## 4 Analysis for template tasks in next-token-prediction setting

We switch gears to the next-token prediction setting with the cross-entropy loss, where the output label may be a token as in the example of Figure 3; formal definition is in Appendix J. The simplest task consists of template "$\alpha$" labeled by "$\alpha$". An example train set is $\{(A, A), (B, B), (C, C)\}$, where $A, B, C \in \mathcal{X}$ are tokens, and then we test with $(x^{test}, y^{test}) = (D, D)$ which is not in the train set. This task captures the ability of a model to learn how to copy a symbol, which is important for LLMs that solve problems with multi-stage intermediate computations and must copy these to later parts of a solution (Csordás et al., 2021). From now on, we only consider this "copying" task.

We consider an architecture $f_{\mathsf{attn}}(\boldsymbol{x}; \boldsymbol{\theta})$ with just a multi-head attention layer, and we tie the embedding and unembedding weights as in practice (Brown et al., 2020). Define the train loss and test loss as follows, where $\ell$ is the cross-entropy loss and $x^{test}$ is a token unseen in the training data: $\mathcal{L}_{train}(\boldsymbol{\theta}) = \frac{1}{n} \sum_{i=1}^n \ell(f_{\mathsf{attn}}(x_i; \boldsymbol{\theta}), y_i)$ and $\mathcal{L}_{test}(\boldsymbol{\theta}) = \ell(f_{\mathsf{attn}}(x^{test}), y^{test})$. We prove this network does not generalize on unseen symbols when trained, as we take the embedding dimension large. Our evidence is from analyzing the early time of training, and showing that the test loss on unseen symbols does not decrease.

**Theorem 4.1** (Failure of transformers at copying). *For any learning rates such that* $-\frac{\partial \mathcal{L}_{train}}{\partial t}\big|_{t=0} = O(1)$, *we must have that* $\frac{\partial \mathcal{L}_{test}}{\partial t}\big|_{t=0} \to 0$ *as* $d_{emb} \to \infty$.

The proof idea is that since the input string has length $k = 1$, the architecture simplifies: all softmaxes in the attention heads output 1, and the network is a sum of attention heads of the form $\boldsymbol{X}\boldsymbol{W}_E \boldsymbol{W}_V \boldsymbol{W}_O^T \boldsymbol{W}_E^T$. At early times the evolution of the weights $\boldsymbol{W}_V \boldsymbol{W}_O^T$ will roughly lie in the span of $\{\boldsymbol{W}_E^T \boldsymbol{e}_{x_i} \boldsymbol{e}_{x_i}^T \boldsymbol{W}_E\}_{i \in [n]}$, which as the embedding dimension becomes large will be approximately orthogonal to the direction $\boldsymbol{W}_E^T \boldsymbol{e}_{x^{test}} \boldsymbol{e}_{x^{test}}^T \boldsymbol{W}_E$ that would lower the test loss. This suggests the following modification to transformers allows them to copy symbols never seen at training:

**Theorem 4.2** (Adding one parameter allows copying). *After reparametrizing the attention* (3) *so that in each head* $\boldsymbol{W}_V \boldsymbol{W}_O^T$ *is replaced by* $\boldsymbol{W}_V \boldsymbol{W}_O^T + b\boldsymbol{I}$ *where* $b$ *is a trainable parameter, there are learning rates such that* $-\frac{\partial \mathcal{L}_{train}}{\partial t}\big|_{t=0} = O(1)$ *and* $-\frac{\partial \mathcal{L}_{test}}{\partial t}\big|_{t=0} = \Omega(1)$ *as* $d_{emb} \to \infty$.

Figures 3 and 5 illustrate the benefit of this additional per-head parameter on the copying task. It is not equivalent to adding a trainable skip connection as in ResNet (He et al., 2016). Instead, the addition of $b_h \boldsymbol{I}$ encodes an *attention-modulated* skip connection that allows copying tokens between the transformer's streams. A related modification of adding a head with the hardcoded $\boldsymbol{X}\boldsymbol{X}^T$ as its attention matrix was proposed in Zhang et al. (2022).

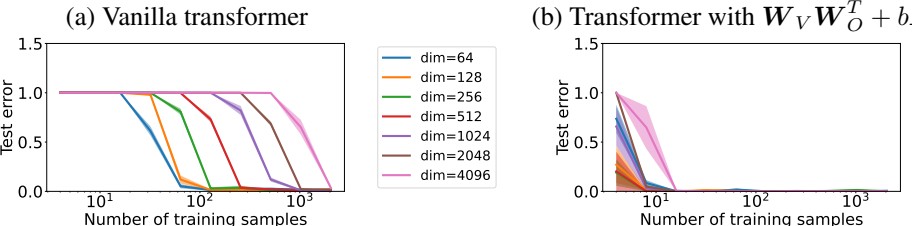

Figure 5: (a) Transformers fail on the copying task as embedding dimension $d_{emb}$ grows (Theorem 4.1); (b) Success when reparametrizing $\boldsymbol{W}_V \boldsymbol{W}_O^T$ as $\boldsymbol{W}_V \boldsymbol{W}_O^T + b\boldsymbol{I}$ (Theorem 4.2). Details in Appendix A.

## 5 EXPERIMENTS

Figures 2 and 3 (and additional experiments in Appendix B) show that our reparametrizations can give a significant data-efficiency benefit on template tasks. Figure 6 shows they can also give improvements on real data. In Figure 7, we see that pretraining outperforms random initialization on a template task. This might be explained by several heads of the pretrained model with diagonals stronger from other weights (originally observed in (Trockman & Kolter, 2023)). These learned diagonals resemble our proposed transformer modifications and so might be driving the data-efficiency of fine-tuning a pretrained model. Appendix B provides extensive experiments on the effect of hyperparameters, inductive biases of different models, and varying levels of task difficulty.

| Dataset | GPT-2 | GPT-2 + trainable identity scalings (ours) |
|---|---|---|
| Wikitext2 | 64.00 | **60.46** |
| Wikitext103 | 16.83 | **16.40** |

Figure 6: Perplexity of GPT-2 trained from random initialization with Adam learning rate 3e-4 for 20 epochs on Wikitext (smaller perplexity is better). GPT-2 has 117M parameters, and we add an extra 288 parameters (2 per head). Interestingly, even though the task is Wikipedia modeling, and therefore is not a pure reasoning task, the transformer modifications still give an improvement.

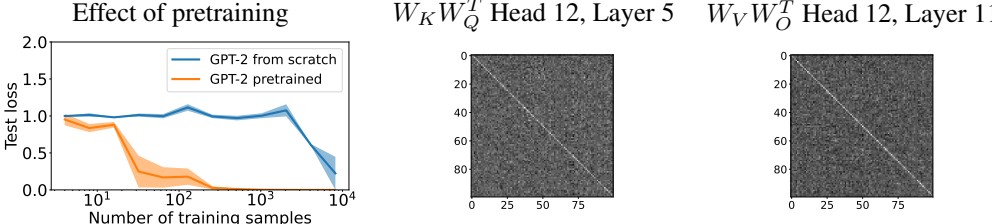

Figure 7: Left: Pretrained versus randomly-initialized GPT-2 test loss when fine-tuned on $\alpha\beta\alpha$ vs. $\alpha\beta\beta$ template task. Right: some GPT-2 pretrained heads have strong diagonals (zoomed to 100x100 top-left corner).

## 6 DISCUSSION

We show that transformers are a universal architecture for template tasks in the regression setting: when trained with gradient descent with enough training data they learn to reason relationally. However, transformers are not optimal – empirically they require large amounts of data to learn basic tasks, and in the next-token-prediction setting they fail at copying unseen symbols. Thus, we have proposed architectural modifications to improve their inductive bias towards logical reasoning. It seems promising to explore other reasoning tasks (for example, reasoning with syllogisms, reasoning by symmetry, and compositional reasoning). It may also be fruitful to study data augmentation approaches (e.g., concatenating the tensorization $\boldsymbol{X}\boldsymbol{X}^T$ to the input, so as to encourage use of relational information). Additionally, tight quantitative upper and lower bounds on the data and width of the architecture needed, depending on the template task, are an interesting open direction.

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

CONTENTS

## A   DETAILS FOR FIGURES IN MAIN TEXT

Code is available at `https://github.com/eboix/relational-reasoning/`.

**Psychometric tasks**   We describe how the tasks in Figure 1 fall under the template framework.

- *(a) Distribution of 3*. The task is to complete the bottom row so that the set of elements is the same as in the top row (answer: 2). To input this task into a language model, a token is used to represent each symbol. The example in the figure matches template "$\alpha\beta\gamma \ \gamma\alpha\square \ \epsilon\alpha\beta\gamma$", with label +2. There are other templates for this task, corresponding to different arrangements of the objects, such as "$\alpha\beta\gamma \ \beta\gamma\square \ \alpha\gamma\epsilon\beta$" with label +1, and "$\alpha\beta\gamma \ \gamma\beta\square \ \epsilon\beta\alpha\gamma$" with label +3. In total there are 144 templates, since the first 3 elements of the template are always $\alpha\beta\gamma$, and then there are 6 choices for the permutation in the next row, and finally 24 choices for the permutation in the final row.

- *(b) Relational match-to-sample*. The task is to match the first row to one of two alternative patterns (answer: 1). Again, a token is used to represent each symbol. The example in the figure matches "$\alpha\beta\beta \ \gamma\delta\delta \ \epsilon\epsilon\tau$" with label +1. A simple combinatorial calculation gives a total of 40 templates (5 possible patterns in the first row, times 2 choices for whether the first option or the second option is correct, times 4 choices for the pattern of alternative option).

- *(c) Raven's progressive matrices*. A standard Raven's progressive matrices task (Raven, 1938) (answer: three dark circles). For each of the dimensions of shape, number, and color, we have a "distribution of 3" task with a symbolic label. For example, for the shapes in the figure, the task is "$\alpha\beta\gamma \ \beta\gamma\alpha \ \gamma\beta$?" with label $\alpha$. Since another possibility is for each row to be constant (as in, e.g., the case of numbers), another possible template is "$\alpha\alpha\alpha \ \beta\beta\beta \ \gamma\gamma$?" with label $\gamma$, and so there is a total of 36+1 = 37 possible templates per dimension. This discussion assumes that the only patterns in the progressive matrices are distribution of 3, and constant. If progressions are also allowed as in Webb et al. (2023a), these can be incorporated by adding corresponding templates.

**Transformer performance** In all experiments, standard transformer architectures are used. In Figure 2, The architecture is a 2-layer transformer with 16 heads per layer, embedding dimension 128, head dimension 64, MLP dimension 256, trained with Adam with learning rate 1e-3 and batch-size 1024. The $n$ training samples are chosen by picking the variable names at random from an alphabet of $n$ tokens. The test set is the same two programs but with disjoint variable names. The reported error bars are on average over 5 trials. The learning rate for each curve is picked as the one achieving best generalization in $\{10^{-5}, 10^{-4}, 10^{-3}, 10^{-2}\}$. In Figure 3, the setting is the same except that the transformer is 4-layer transformer and has embedding dimension 512. In Figure 5 the same hyperparameters as in Figure 2 are used. In order to measure the generalization performance of the learned model on unseen symbols, we evaluate it on a test set and a validation set which each consist of 100 samples drawn in the same way as the training dataset, but each using a disjoint alphabet of size 100. Therefore, there is no overlap in the support of the train, test, and validation distributions. We use the validation loss to select the best epoch of training out of 1000 epochs. We report the test loss on this saved model.

## B ADDITIONAL EXPERIMENTS

We report extensive additional experiments probing the template task framework. In each of these, the training dataset consists of $n$ random training samples. Each sample is drawn according to a template distribution. The following are template tasks on which we test.

- *$\alpha\beta\alpha$ vs. $\alpha\beta\beta$ task.* Uniform on two templates $\alpha\beta\alpha$ and $\alpha\beta\beta$ with labels 1, -1 respectively and $\alpha$ and $\beta$ are wildcards.

- *$\alpha\beta\alpha\beta$ vs. $\alpha\alpha\beta\beta$ task.* Same as above, except with templates $\alpha\beta\alpha\beta$ and $\alpha\alpha\beta\beta$.

- *Length-$k$ majority task.* Uniform on $2^{k-1}$ templates $\alpha \times \{\alpha, \beta\}^{k-1}$ where $\alpha$ and $\beta$ are wildcards. A template $\boldsymbol{z}$ has label 1 if its first token occurs in the majority of the rest of the string, and -1 otherwise. Namely, $f_*(\boldsymbol{z}) = \begin{cases} 1, & |\{i : z_1 = z_i\}| > (k+1)/2 \\ -1, & \text{otherwise} \end{cases}$.

- *Random template task.* A certain number $r$ of templates are drawn uniformly from $(\mathcal{W} \cup \mathcal{X})^k$, conditioned on being pairwise distinct. The task is the uniform distribution over these $r$ templates, with random Gaussian labels centered and scaled so that the trivial MSE is 1.

For any of these tasks, we generate $n$ training samples as follows. We substitute the wildcards for regular tokens using a randomly chosen injective function $s : \mathcal{W} \to \mathcal{X}$ where $\mathcal{X}$ is an alphabet of size $n$ (which is the same size as the number of samples). For example, if a given sample is generated from template $\alpha\beta\alpha$ with substitution map $s$ mapping $s(A) = 12$, $s(B) = 5$, then the sample will be $[12, 5, 12]$. Error bars are over 5 trials, unless otherwise noted.

### B.1 EFFECT OF TRANSFORMER HYPERPARAMETERS

We test a standard transformer architecture on the $\alpha\beta\alpha$ vs. $\alpha\beta\beta$ task, varying some of the hyperparameters of the transformer to isolate their effect while keeping all other hyperparameters fixed. The base hyperparameters are depth 2, embedding dimension 128, head dimension 64, number of heads per layer 16, trained with Adam with minibatch size 1024 for 1000 epochs. Our experiments are as follows:

- *Learning rate and $n$.* In Figure 8 we vary the learning rate and $n$.

- *Learning rate and depth.* In Figure 9 and Figure 10, we vary the learning rate and the depth, for $n = 512$ and $n = 1024$, respectively.

- *Learning rate and number of heads.* In Figure 11 and 12, we vary the learning rate and number of heads, for $n = 512$ and $n = 1024$, respectively.

- *Learning rate and embedding dimension.* In Figure 13 we vary the learning rate and embedding dimension for $n = 1024$.

- *Learning rate and batch size.* In Figure 14, we vary the learning rate and batch-size for $n = 512$. In Figure 16 we vary the batch-size and $n$ for learning rate 0.001.

- *Training just the last layer.* In Figure 15, we train just the last layer, and see that the network does learn to generalize out of distribution, as predicted by our theory. However, the number of samples and number of epochs needed is larger than when all parameters are trained. We train for 10000 epochs and have 64 heads per layer in this experiment.

## B.2 EFFECT OF COMPLEXITY OF TASK

We test an out-of-the-box transformer architecture with depth 2, embedding dimension 128, head dimension 64, number of heads 16, trained with Adam with batch-size 1024 for 1000 epochs, on various template tasks.

- *Comparing difficulty of various tasks.* Figure 17 we plot the performance on various simple tasks.
- *Random tasks.* In Figures 18, 19, 20, and 21, we test on random template tasks, and investigate the effects of template length, wildcard alphabet size, regular token alphabet size, number of templates.

## B.3 EFFECT OF INDUCTIVE BIAS OF MODEL

We provide experiments probing the effect of the inductive bias of the model:

- *Different architectures.* In Figure 22, we plot the test loss for different architectures on the $\alpha\beta\alpha$ vs. $\alpha\beta\beta$ template task, including transformers with trainable identity perturbations to $W_Q W_K^T$, to $W_V W_O^T$, to both $W_Q W_K^T$ and $W_V W_O^T$, or to neither. Figure 23 illustrates on the beneficial effect of the transformer modification for the majority task with different lengths, lowering the amount of data needed by an order of magnitude.
- *Size of model.* In Figure 24 we compare the test loss of fine-tuning small, medium and large pretrained GPT-2 networks on the $\alpha\beta\alpha$ vs. $\alpha\beta\beta$ template task.
- *MLP with $XX^T$ data augmentation vs. transformer.* In Figure 25, we compare the test loss of a transformer with the test loss of an MLP where the input data has been augmented by concatenating $\text{vec}(XX^T)$, which is a data augmentation that improves performance under the NTK criterion similarly to the discussion in Section 3.3 and the discussion section.

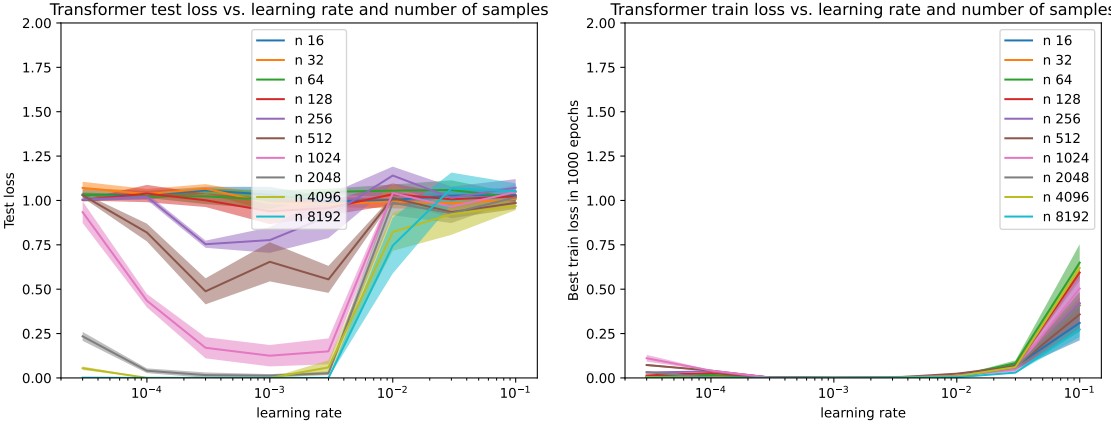

Figure 8: Learning rate versus $n$ = number of samples = training alphabet size. Taking too large or too small of a learning rate can hurt generalization even when the train loss is close to zero.

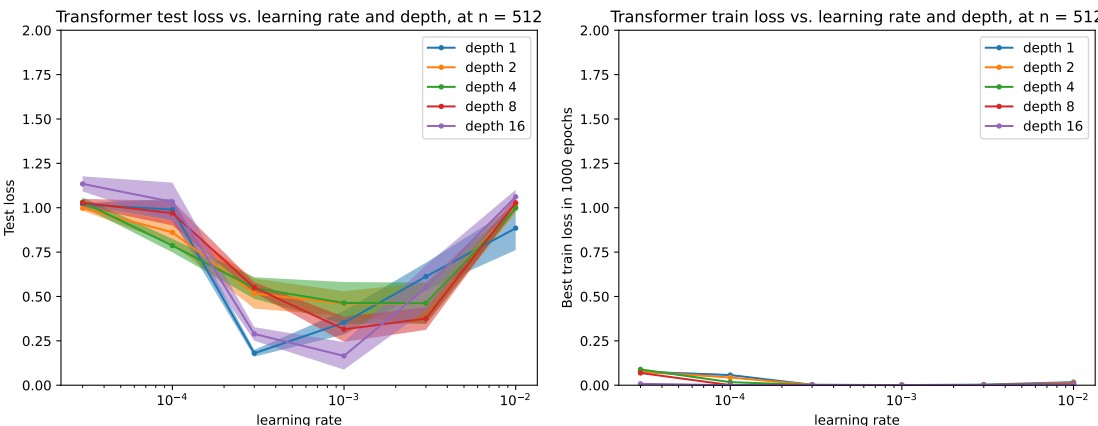

Figure 9: Learning rate vs. depth at $n = 512$. No clear relationship between depth and generalization. Too large or too small of a learning rate can hurt generalization.

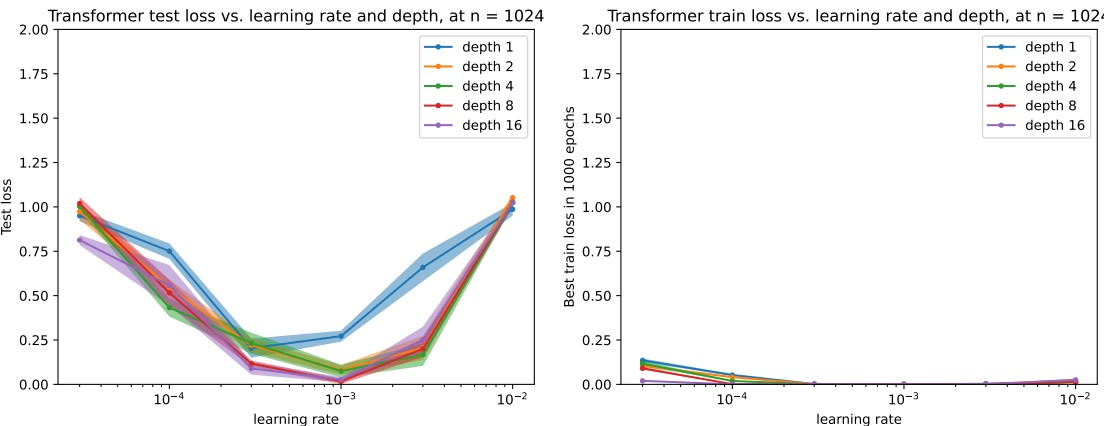

Figure 10: Learning rate vs. depth at $n = 1024$. Unlike $n = 512$ case, in previous figure, larger depth typically performs better.

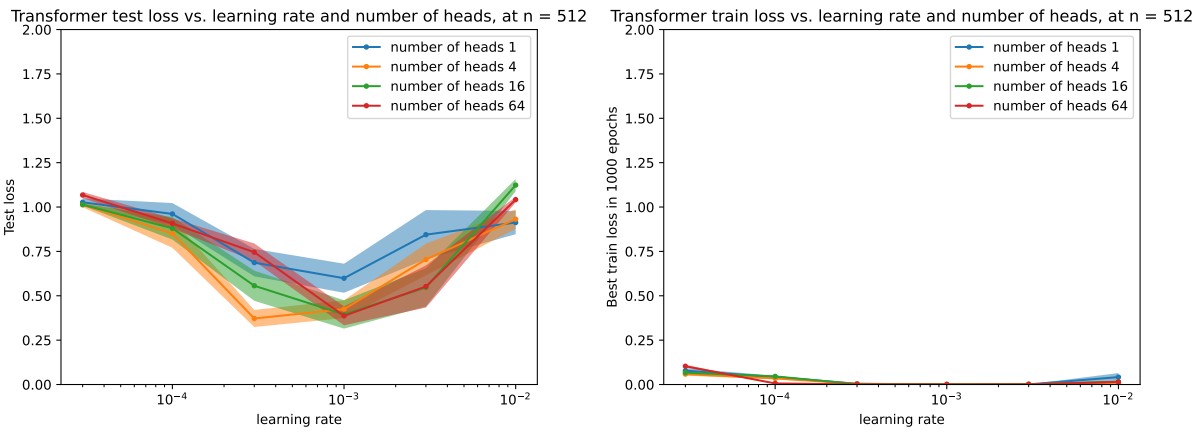

Figure 11: Learning rate vs. number of heads per layer at $n = 512$. More heads are better than one head.

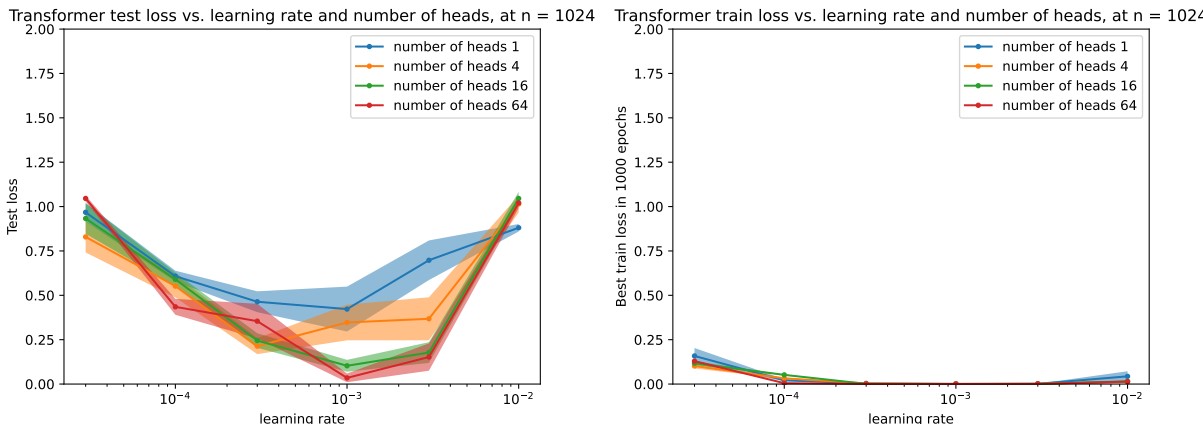

Figure 12: Learning rate vs. number of heads at $n = 1024$. More heads are better.

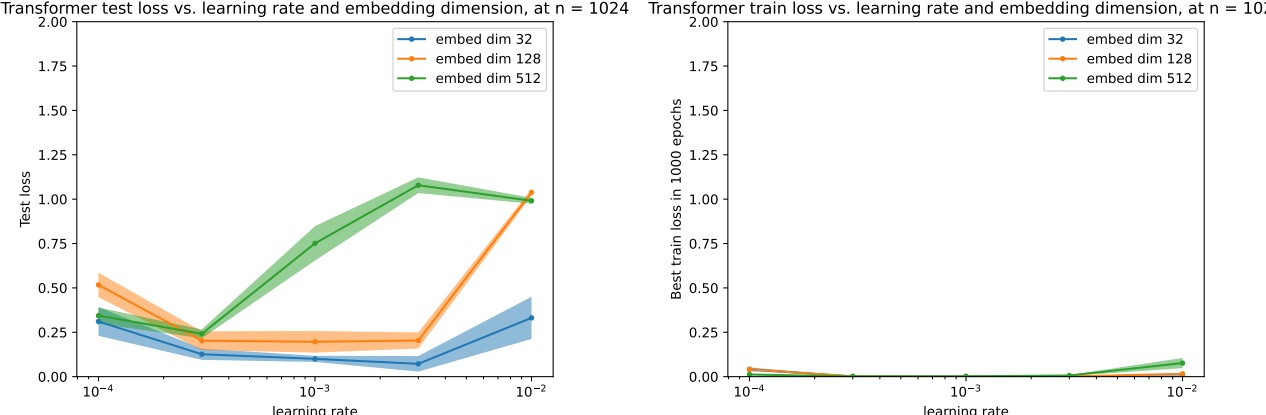

Figure 13: Learning rate vs. embedding dimension at $n = 1024$. Smaller embedding dimension is generally better.

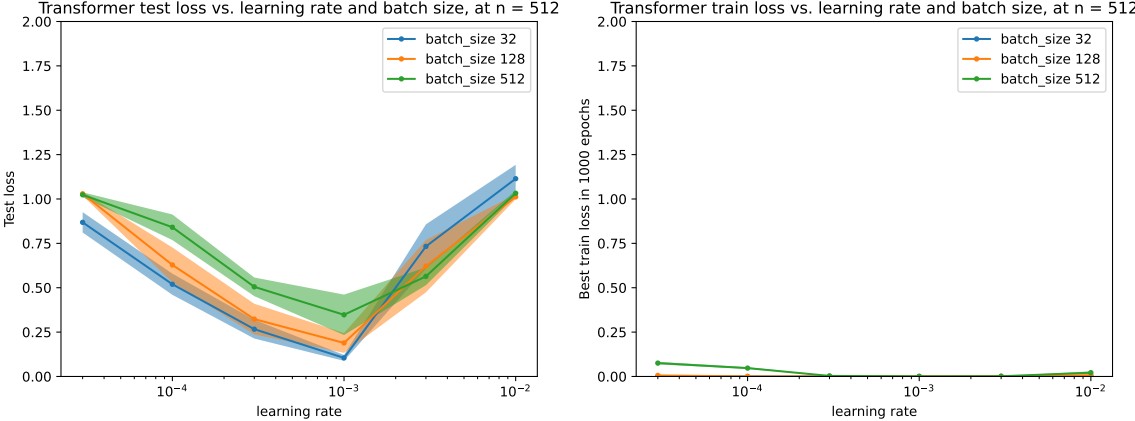

Figure 14: Learning rate vs. batch-size at $n = 512$. Smaller batch size is better.

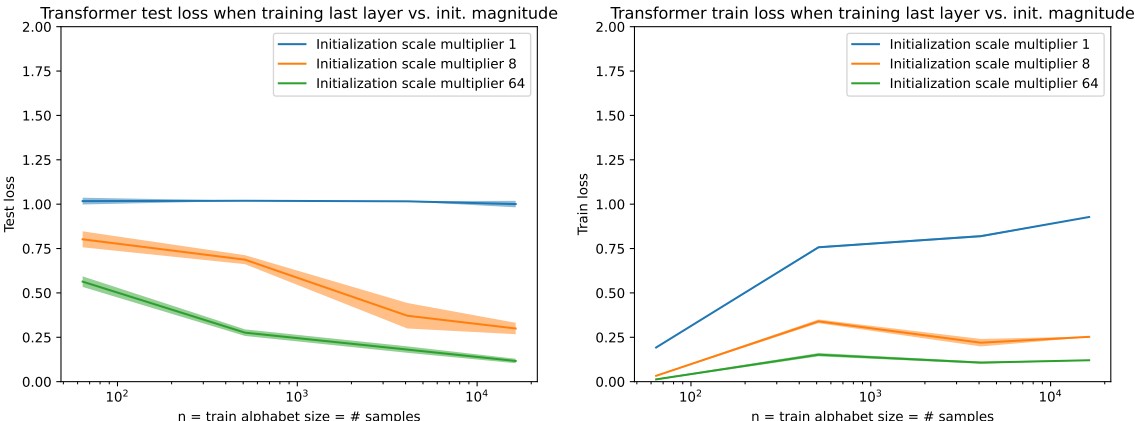

Figure 15: Training just the final unembedding layer suffices for the transformer to generalize out of distribution, as predicted by our theory. However, the number of samples and number of epochs needed is larger than when all parameters of the network are trained. Understanding why training all parameters gives better performance than training just the last layer is an interesting future direction. We report results for 3 different magnitudes of initializing the weights of attention mechanism (1 times, 8 times, and 64 times the standard initialization), and find that larger initialization helps, which we conjecture is due to the softmax being in the saturated regime, which leads to more weight on the relational features.

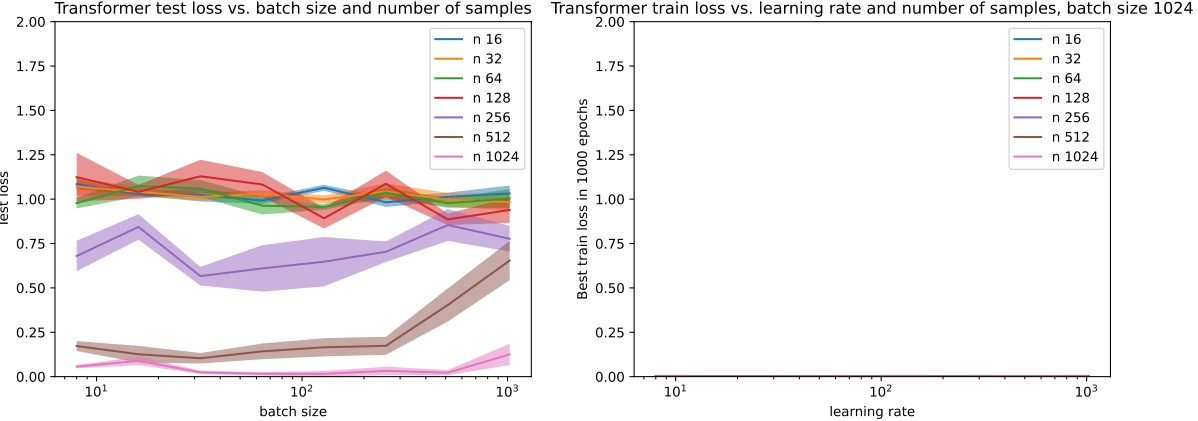

Figure 16: Batch size vs. $n$ = number of training samples = training alphabet size. Smaller batch size is generally better, which is most visible at $n = 512$.

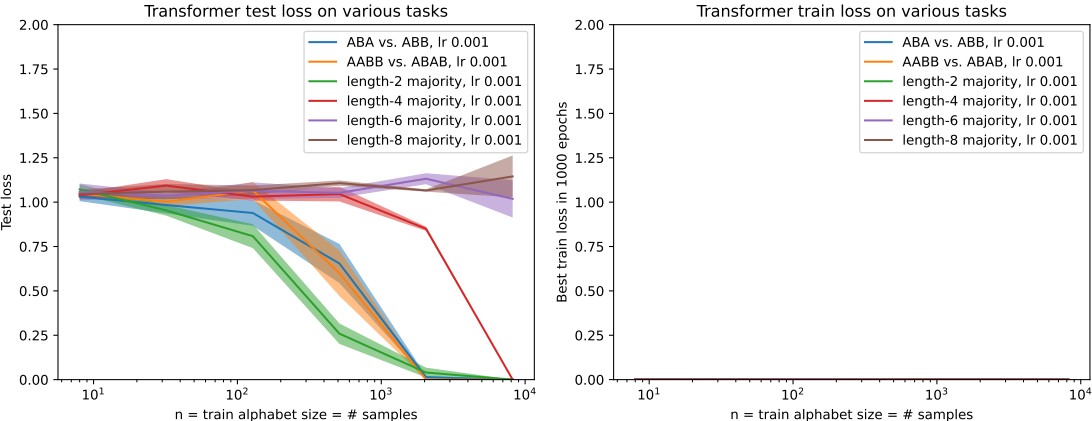

Figure 17: Test and train loss of transformer for various tasks. The $\alpha\beta\alpha$ vs. $\alpha\beta\beta$ task consists of two templates $\alpha\beta\alpha$ and $\alpha\beta\beta$ with labels +1, -1. The $\alpha\alpha\beta\beta$ vs. $\alpha\beta\alpha\beta$ task has templates +1, -1. For each $k$, the length-$k$ majority task consists of all templates in $\{\alpha\} \times \{\alpha, \beta\}^{k-1}$, where each template has label 1 if $\alpha$ occurs more times in the last $k-1$ entries, and label +1 if $\alpha$ occurs fewer times in the last $k-1$ entries. The trivial model that outputs 0 always will achieve test loss of 1.

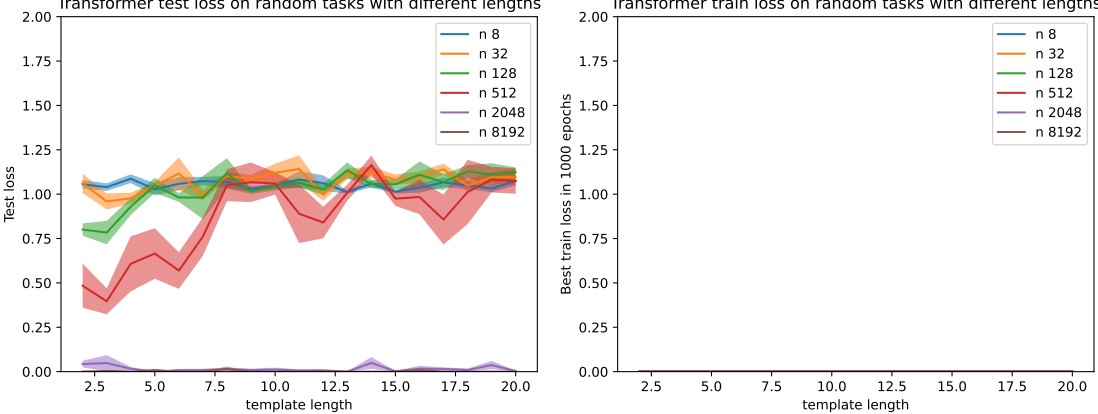

Figure 18: Performance on tasks corresponding of two, distinct random templates with two wildcards $\alpha, \beta$, and with labels $1, -1$, respectively. Performance degrades as the template length increases.

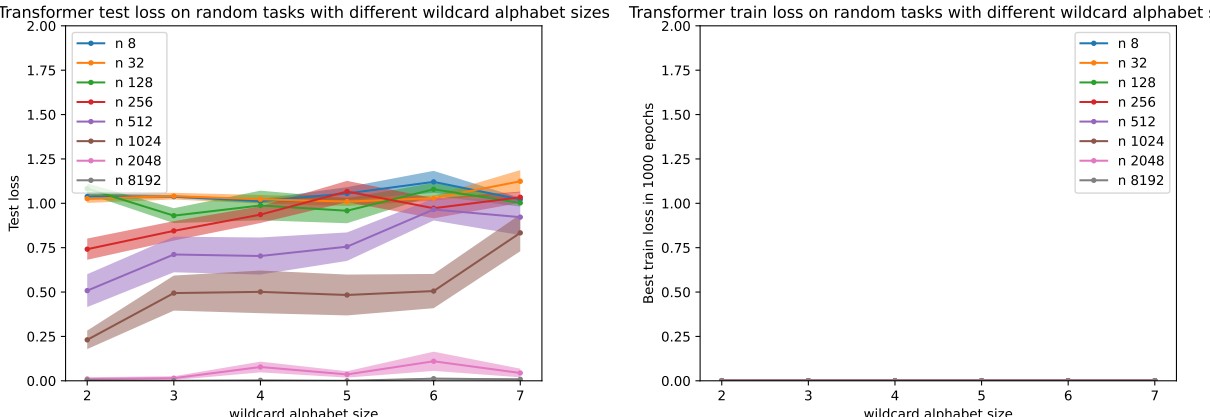

Figure 19: Performance on tasks corresponding of two random templates of length 5, labeled with $1, -1$, respectively. Each template is sampled randomly from $\mathcal{W}^5$, conditioned on the two templates being distinct. We vary the wildcard alphabet size $|\mathcal{W}|$. Performance generally degrades as the wildcard alphabet size increases.

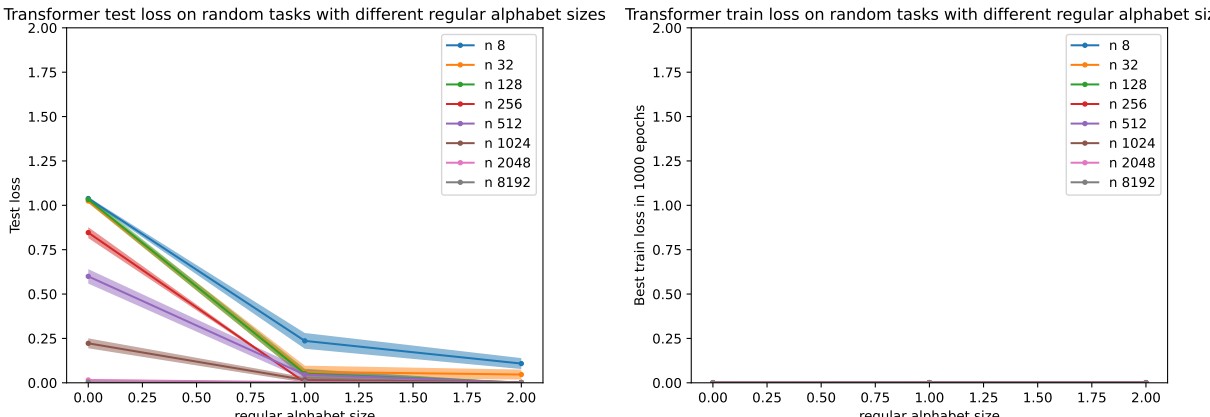

Figure 20: Performance on tasks corresponding of two random templates of length 5, labeled with $1, -1$, respectively. Each template is sampled randomly from $(\mathcal{W} \cup \mathcal{X})^5$, conditioned on the two templates being distinct. We keep $|\mathcal{W}| = 2$ and vary the regular token alphabet size $|\mathcal{X}|$ between 0 and 2. Performance quickly improves as the regular token alphabet size increases.

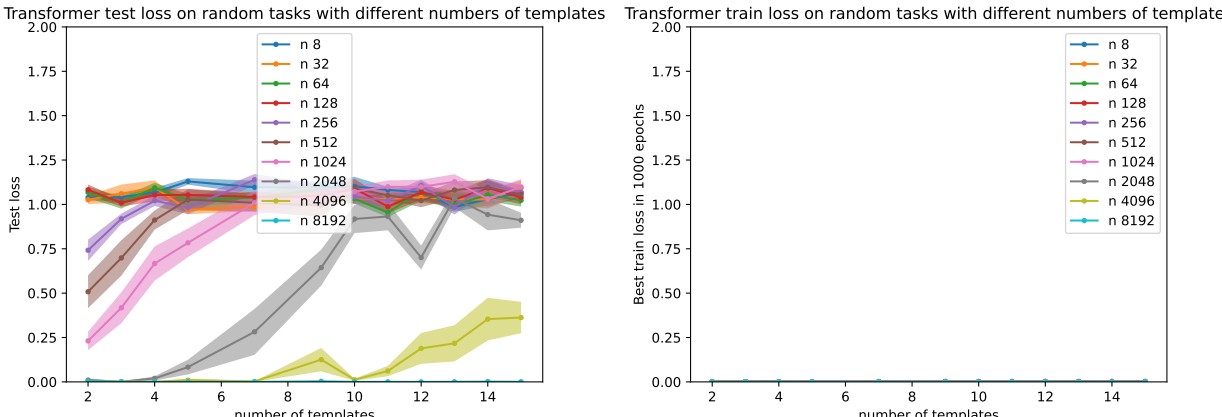

Figure 21: Performance on tasks corresponding of two random templates of length 5, labeled with $1, -1$, respectively. Each template is sampled randomly from $(\mathcal{W} \cup \mathcal{X})^5$, conditioned on the two templates being distinct. We keep $|\mathcal{W}| = 2$ and vary the regular token alphabet size $|\mathcal{X}|$ between 0 and 2. Performance quickly improves as the regular token alphabet size increases.

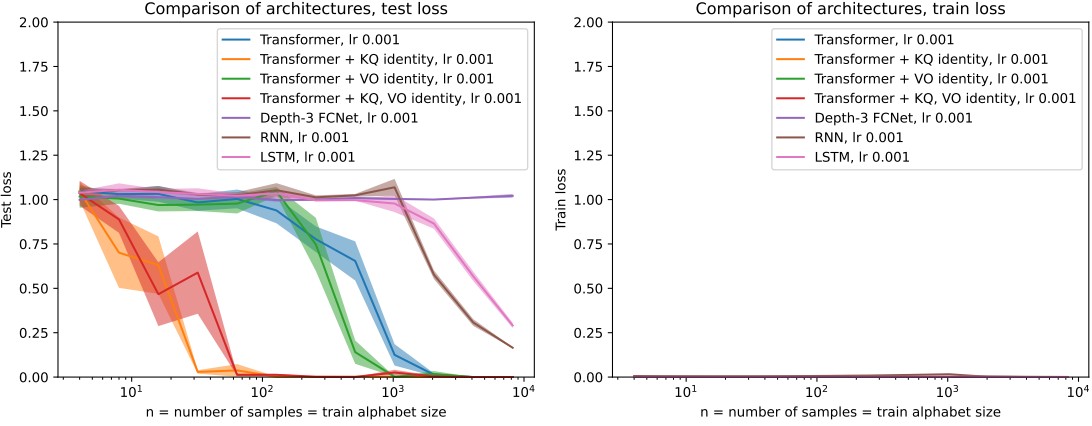

Figure 22: Different architectures on $\alpha\beta\alpha$ vs. $\alpha\beta\beta$ task. Transformer outperforms the other architectures, especially with the reparametrization that prioritizes identities in heads.

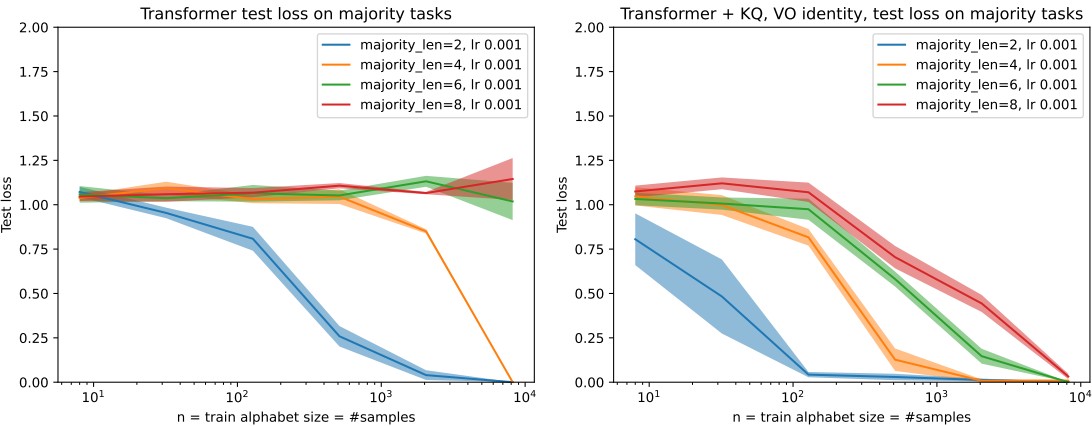

Figure 23: Comparison of test loss of architectures on length-$k$ majority task with different $k$. Left: vanilla transformer architecture. Right: transformer architecture plus the trainable identity scalings on each attention head's $W_K W_Q^T$ and $W_V W_O^T$ matrices. Notice that again the transformer reparametrization lowers the amount of data needed by at least an order of magnitude.

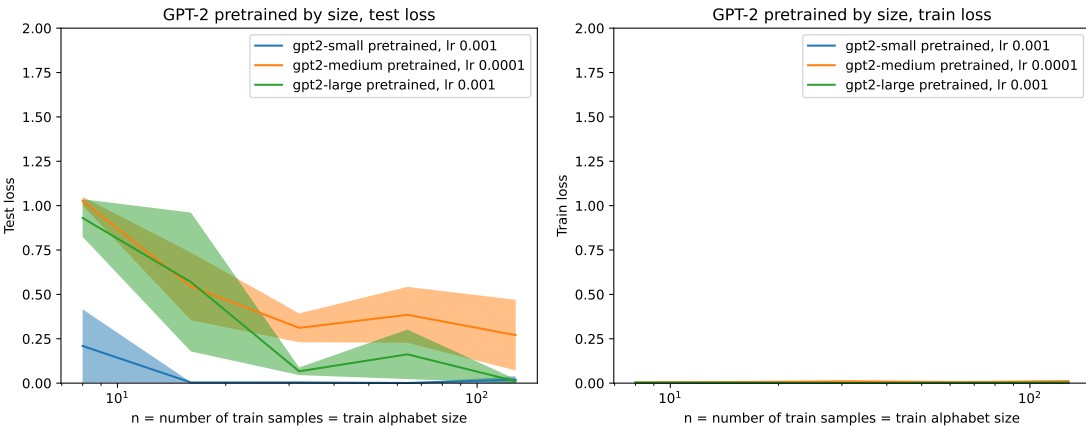

Figure 24: Pretrained GPT-2 of different sizes fine-tuned on $\alpha\beta\alpha$ vs. $\alpha\beta\beta$ task.

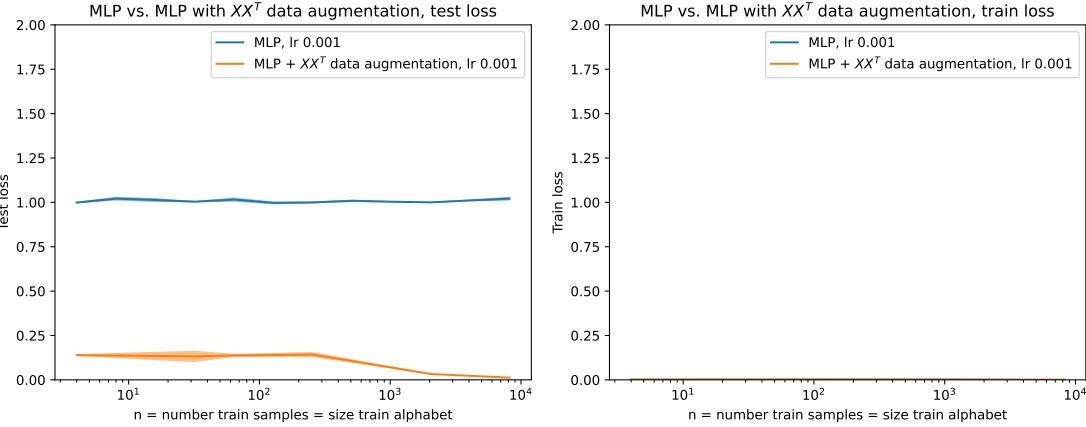

Figure 25: Test loss of MLP with $X X^T$ data augmentation, where it is concatenated to input, versus MLP without data augmentation, versus transformer.

## C    PROOF OF THEOREM 3.4

There are two main parts to the proof. First, in Section C.1 we establish a lemma with a sufficient condition for a kernel method to have good test loss. Second, in Section C.2 we prove that the transformer random features kernel $K_{\text{trans}}$ satisfies this condition for almost any $\beta, \gamma, b_1, b_2$ parameters. We conclude in Section C.3.

**Remark C.1.** The reason that we state our result with mean-squared error loss is that we have the closed-form solution (5) for the function that the kernel method learns in terms of its kernel and the data. Such an expression is not known for the cross-entropy loss.

### C.1    PART 1. GENERAL SUFFICIENT CONDITION FOR GOOD TEST LOSS

We restrict ourselves to token-symmetric kernels, which are kernels whose values are unchanged if the tokens are relabeled by a permutation.

**Definition C.2** (Token-symmetric kernel). $K$ is token-symmetric if for any permutation $\pi : \mathcal{X} \to \mathcal{X}$ we have $K(\boldsymbol{x}, \boldsymbol{y}) = K([\pi(x_1), \ldots, \pi(x_k)], [\pi(y_1), \ldots, \pi(y_k)])$.

Token-symmetry is a mild condition, as most network architectures used in practice (including transformers) have token-symmetric neural tangent kernels at initialization. We emphasize that token-symmetry is not sufficient for good test loss since MLPs are a counterexample (see Appendix I.)

To state the sufficient condition for good test loss, let $\{\boldsymbol{z}_1, \ldots, \boldsymbol{z}_r\} = \text{supp}(\mu_{\text{tmplt}})$ be the template distribution support. Define also the set $\mathcal{R} = \cup_{i \in [k], j \in [r]} \{z_{j,i}\}$ of tokens that appear in the templates. Finally, define $\boldsymbol{N} \in \mathbb{R}^{r \times r}$ by

$$N_{ij} = K(\text{sub}(\boldsymbol{z}_i, s), \text{sub}(\boldsymbol{z}_j, s')), \tag{11}$$

where $s, s' : \mathcal{W} \to \mathcal{X}$ are substitution maps satisfying

$$s(\mathcal{W}) \cap s'(\mathcal{W}) = 0 \quad \text{and} \quad s(\mathcal{W}) \cap \mathcal{R} = s'(\mathcal{W}) \cap \mathcal{R} = \emptyset. \tag{12}$$

One can check that because of the token-symmetry of the kernel $K$, the matrix $\boldsymbol{N}$ is uniquely-defined regardless of the substitution maps $s, s'$ chosen, as long as they satisfy (12).

**Lemma C.3** (It suffices for $\boldsymbol{N}$ to be nonsingular). *If $K$ is a token-symmetric kernel, and $\boldsymbol{N}$ is nonsingular, then kernel ridge regression achieves vanishing test loss.*

*Formally, there are constants $c, C > 0$ and ridge regularization parameter $\lambda > 0$ depending only on $\mu_{\text{tmplt}}$, $\sigma$, $|\mathcal{W}|$, $\|\boldsymbol{N}^{-1}\|$ and $\|K\|_\infty = \max_{\boldsymbol{x}} K(\boldsymbol{x}, \boldsymbol{x})$, such that for any $\boldsymbol{x}$ matching a template $\boldsymbol{z} \in \text{supp}(\mu_{\text{tmplt}})$ the kernel ridge regression estimator $\hat{f}$ in (5) with kernel $K$ satisfies*

$$|\hat{f}(\boldsymbol{x}) - f_*(\boldsymbol{z})| \leq C\sqrt{\frac{\log(1/\delta)}{n}} + C\sqrt{\frac{1}{\rho}},$$

*with probability at least $1 - \delta - \exp(-cn)$ over the random samples.*

The proof is in Appendix D, but we develop an intuition here on why the nonsingularity of the matrix $\boldsymbol{N}$ is important. Let $[n] = \mathcal{I}_1 \sqcup \mathcal{I}_2 \sqcup \cdots \sqcup \mathcal{I}_n$ be the partition of the samples such that if $i \in \mathcal{I}_j$ then sample $(\boldsymbol{x}_i, y_i)$ is drawn by substituting the wildcards of template $\boldsymbol{z}_j$ with substitution map $s_i : \mathcal{W} \to \mathcal{X}$. We show that for any string $\boldsymbol{x}$ matching template $\boldsymbol{z}_j$, the kernel ridge regression solution (5) is approximately equal to the average of the labels of the samples corresponding to template $j$,

$$\boldsymbol{y}^T (\hat{\boldsymbol{K}} + \lambda \boldsymbol{I})^{-1} \boldsymbol{k}(\boldsymbol{x}) \approx \frac{1}{|\mathcal{I}_j|} \sum_{i \in \mathcal{I}_j} y_i \approx f_*(\boldsymbol{z}_j). \tag{13}$$

In order to see why this is true, consider the regime in which the sample diversity is very high, i.e., $\rho \gg 1$. Since $\rho$ is large, any particular token is highly unlikely to be substituted. This has the following implications:

- For most sample pairs $i \neq i' \in [n]$, the maps $s_i$ and $s_{i'}$ have disjoint range: $s_i(\mathcal{W}) \cap s_i'(\mathcal{W})$.
- For most samples $i \in [n]$, the substituted tokens are not in the templates: $s_i(\mathcal{W}) \cap \mathcal{R} = \emptyset$.

These are the same conditions as in (8). So by the token-symmetry of the kernel, for most pairs of samples the empirical kernel matrix is given by $N$:

$$\hat{K}_{i,i'} := K(\boldsymbol{x}_i, \boldsymbol{x}_{i'}) = N_{j,j'} \text{ for most } i \in \mathcal{I}_j, i' \in \mathcal{I}_{j'} .$$

So if $N$ is nonsingular, then $\hat{K}$ has $r$ large eigenvalues, and $n - r$ much smaller eigenvalues. This turns out to be sufficient for (9) to hold. We refer the reader to Appendix D for more details.

## C.2 PART 2. ANALYZING THE TRANSFORMER RANDOM FEATURES KERNEL

We show that the transformer random features kernel $K_{\text{trans}}$ satisfies the sufficient condition of Lemma C.3 for vanishing test loss. It is clear that the kernel is token-symmetric because the definition is invariant to the permutation relabelings of the tokens. The difficult part is to show that the matrix $N_{\text{trans}} := N$ defined with kernel $K = K_{\text{trans}}$ in (11) is nonsingular. The main challenge is that the transformer kernel does not have a known closed-form solution because of the softmax terms in its definition (4). Furthermore, the result is especially challenging to prove because it must hold for *any* collection of disjoint templates $\boldsymbol{z}_1, \ldots, \boldsymbol{z}_r$.

We analyze the MLP layer and the attention layer of the transformer separately. We observe that a "weak" condition on $K_{\text{attn}}$ can be lifted into the "strong" result that $N_{\text{trans}}$ is nonsingular. Intuitively, as long as $K_{\text{attn}}$ is not a very degenerate kernel, it is very unlikely that the MLP layer has the cancellations that would be needed to make $N_{\text{trans}}$ nonsingular.

**Lemma C.4** (Nonsingularity of $N_{\text{trans}}$, restatement of Lemma 3.6). *Suppose for every non-identity permutation $\tau \in S_r \setminus \{\text{id}\}$,*

$$\sum_{i \in [r]} K_{\text{attn}}(\text{sub}(\boldsymbol{z}_i, s), \text{sub}(\boldsymbol{z}_i, s')) \neq \sum_{i \in [r]} K_{\text{attn}}(\text{sub}(\boldsymbol{z}_i, s), \text{sub}(\boldsymbol{z}_{\tau(i)}, s')) , \tag{14}$$

*where $s, s'$ are the substitution maps in the definition of $N_{\text{trans}}$ in (12). Let the MLP layer's activation function be $\phi(t) = \cos(b_1 t + b_2)$. Then for almost any choice of $b_1, b_2$ (except for a Lebesgue-measure-zero set), the matrix $N_{\text{trans}}$ is nonsingular.*

This lemma is proved in Appendix E, by explicitly evaluating the Gaussian integral, which is possible since the activation function is the cosine function. Although in our proof we use the cosine activation function, we conjecture that this result should morally hold for sufficiently generic non-polynomial activation functions. Next, we prove the condition on $N_{\text{attn}}$.

**Lemma C.5** (Non-degeneracy of $K_{\text{attn}}$, restatement of Lemma 3.7). *The condition (14) holds for Lebesgue-almost any $\beta, \gamma$.*

The proof is in Appendix F. First, we prove the analyticity of the kernel $K_{\text{attn}}$ in terms of the hyperparameters $\beta$ and $\gamma$ which control the softmax inverse temperature and the positional embeddings. Because of the identity theorem for analytic functions, it suffices to show at least one choice of hyperparameters $\beta$ and $\gamma$ satisfies (14) for all non-identity permutations $\tau$. Since $K_{\text{attn}}$ does not have a closed-form solution, we find such a choice of $\beta$ and $\gamma$ by analyzing the Taylor-series expansion of $K_{\text{attn}}$ around $\beta = 0$ and $\gamma = 0$ up to order-10 derivatives, which happens to suffice.

## C.3 CONCLUDING THE PROOF OF THEOREM 3.4

By Lemma C.3, it suffices to prove the nonsingularity of the matrix $N_{\text{trans}}$ defined in (11) with kernel $K = K_{\text{trans}}$. Lemma 3.6 gives a condition for nonsingularity that holds for almost any $b_1, b_2$. Lemma 3.7 proves this condition for almost any $\beta, \gamma$. Therefore, Theorem 3.4 follows.

## D SUFFICIENT CONDITION FOR KERNEL METHOD TO GENERALIZE ON UNSEEN SYMBOLS (PROOF OF LEMMA C.3)

We restate and prove Lemma C.3. Let $K$ be a token-symmetric kernel as in Definition C.2. Let $\mu_{\text{tmplt}}$ be a distribution supported on disjoint templates $\boldsymbol{z}_1, \ldots, \boldsymbol{z}_r$ and define $\mathcal{R} = \cup_{i \in [r], j \in [k]} \{z_{i,j}\}$. Recall the definiton of the matrix $N \in \mathbb{R}^{r \times r}$ with

$$N_{i,i'} = K(\text{sub}(\boldsymbol{z}_i, s), \text{sub}(\boldsymbol{z}_{i'}, s')).$$

for substitution maps $s : \mathcal{W} \to \mathcal{X}$, $s' : \mathcal{W} \to \mathcal{X}$ satisfying $s(\mathcal{W}) \cap s'(\mathcal{W}) = s(\mathcal{W}) \cap \mathcal{R} = s'(\mathcal{W}) \cap \mathcal{R} = \emptyset$. Recall that this is well-defined by the token-symmetry of the kernel $K$.

**Lemma D.1** (Restatement of Lemma C.3). *Suppose that $K$ is token-symmetric and $\boldsymbol{N}$ is nonsingular. Then there are constants $0 < c < C$ and $0 < c' < C'$ depending only on $\mu_{\mathsf{tmplt}}$, $\sigma$, $|\mathcal{W}|$, $\|\boldsymbol{N}^{-1}\|$ and $\|K\|_\infty = \max_{\boldsymbol{x}} K(\boldsymbol{x}, \boldsymbol{x})$ such that the following holds. Consider any regularization parameter $\lambda \in [c'n, C'n]$, and any string $\boldsymbol{x}$ matching template $\boldsymbol{z} \in \mathrm{supp}(\mu_{\mathsf{tmplt}})$. Then with probability $\geq 1 - \delta - \exp(-cn)$, the kernel ridge regression estimator $\hat{f}$ achieves good accuracy on $\boldsymbol{x}$:*

$$|\hat{f}(\boldsymbol{x}) - f_*(\boldsymbol{z})| \leq C\sqrt{\frac{\log(1/\delta)}{n}} + C\sqrt{\frac{1}{\rho}}.$$

*Proof.* Note that some proofs of helper claims are deferred to Section D.1. Let $(\boldsymbol{x}_1, y_1), \ldots, (\boldsymbol{x}_n, y_n)$ be the samples seen by the kernel method. We know from (5) that kernel ridge regression outputs the estimator

$$\hat{f}(\boldsymbol{x}) = \boldsymbol{y}^T (\hat{\boldsymbol{K}} + \lambda \boldsymbol{I})^{-1} \boldsymbol{v}(\boldsymbol{x}), \qquad \text{(Kernel ridge regression)}$$

where the empirical kernel matrix $\hat{\boldsymbol{K}} \in \mathbb{R}^{n \times n}$ is

$$\hat{K}_{i,j} = K(\boldsymbol{x}_i, \boldsymbol{x}_j),$$

and $\boldsymbol{y} = [y_1, \ldots, y_n]$, and $\boldsymbol{v}(\boldsymbol{x}) = [K(\boldsymbol{x}_1, \boldsymbol{x}), \ldots, K(\boldsymbol{x}_n, \boldsymbol{x})] \in \mathbb{R}^n$.

**Idealized estimator when sample diversity is high** If the sample diversity is sufficiently high, then for most pairs of samples $i \neq i' \in [n]$, it will be the case that $\boldsymbol{x}_i$ and $\boldsymbol{x}_{i'}$ do not share any of the wildcard substitution tokens. In other words, the wildcard substitution map used to form $\boldsymbol{x}_i$ will have disjoint range from the wildcard substitution map used to form $\boldsymbol{x}_{i'}$. This means that we should expect the estimator $\hat{f}$ to perform similarly to the following idealized estimator:

$$\hat{f}^{ideal}(\boldsymbol{x}) = \boldsymbol{y}^T (\hat{\boldsymbol{K}}^{ideal} + \lambda \boldsymbol{I})^+ \boldsymbol{v}^{ideal}(\boldsymbol{x}), \qquad (15)$$

where $\hat{\boldsymbol{K}}^{ideal} \in \mathbb{R}^{n \times n}$ and $\boldsymbol{v}^{ideal}(\boldsymbol{x}) \in \mathbb{R}^n$ are idealized versions of $\hat{\boldsymbol{K}}$ and $\boldsymbol{v}(\boldsymbol{x})$, formed below. They correspond to the limit of infinitely-diverse samples, when all token substitution maps have disjoint range. For each $j \in [r]$, let $\mathcal{I}_j \subseteq [n]$ be the indices of samples $\boldsymbol{x}_i$ formed by substituting from template $\boldsymbol{z}_j$. For any $i \in \mathcal{I}_j, i' \in \mathcal{I}_{j'}$, let

$$\hat{K}_{i,i'}^{ideal} = N_{j,j'}, \qquad (16)$$

Also, similarly define $\boldsymbol{v}^{ideal}(\boldsymbol{x}) \in \mathbb{R}^n$. For any $i \in \mathcal{I}_j$, let

$$v_i^{ideal}(\boldsymbol{x}) = K(\mathrm{sub}(\boldsymbol{z}_j, s), \boldsymbol{x}), \qquad (17)$$

where $s : \mathcal{W} \to \mathcal{X}$ is a substitution map with $s(\mathcal{W}) \cap \mathcal{R} = s(\mathcal{W}) \cap \{x_i\}_{i \in [k]} = \emptyset$, i.e., it does not overlap with the templates or with $\boldsymbol{x}$ in the tokens substituted for the wildcards. The expressions (16) and (17) are well-defined because of the token-symmetry of the kernel.

If the sample diversity is high, then we show that the idealized estimator $\hat{f}^{ideal}$ is indeed close to the kernel ridge regression solution $\hat{f}$.

**Claim D.2** (Idealized estimator is good approximation to true estimator). *Suppose $\|K\|_\infty = \max_{\boldsymbol{x}} |K(\boldsymbol{x}, \boldsymbol{x})| < \infty$. Then there are constants $C, c > 0$ depending only on $|\mathcal{W}|, \|K\|_\infty, k, r$ such that the following holds. For any $\boldsymbol{x}$, with probability at least $1 - \exp(-cn)$,*

$$|\hat{f}^{ideal}(\boldsymbol{x}) - \hat{f}(\boldsymbol{x})| \leq \frac{C}{\lambda} + \frac{Cn}{\lambda\sqrt{\rho}},$$

*where $\rho$ is defined in Definition 3.3 and measures the diversity of the substitution map distribution.*

**Analyzing the idealized estimator using its block structure** The matrix $\hat{K}^{ideal}$ has block structure with blocks $\mathcal{I}_1, \ldots, \mathcal{I}_r$. Namely, it equals $\hat{K}_{i,i'} = N_{j,j'}$ for all $i \in \mathcal{I}_j, i' \in \mathcal{I}_{j'}$. Similarly, $\boldsymbol{v}^{ideal}(\boldsymbol{x})$ also has block structure with blocks $\mathcal{I}_1, \ldots, \mathcal{I}_r$. This structure allows us to analyze estimator $\hat{f}^{ideal}$ and to prove its accuracy.

In order to analyze the estimator, we prove the following technical claim. The interpretation of this claim is that if $\boldsymbol{x}$ matches template $\boldsymbol{z}_a$, then $\boldsymbol{v}^{ideal}(\boldsymbol{x})$ is equal to any of the rows in $\hat{K}^{ideal}$ that correspond to template $a$. In other words, we should have $(\hat{\boldsymbol{K}}^{ideal})^+ \boldsymbol{v}^{ideal}(\boldsymbol{x}) = \mathbf{1}_{\mathcal{I}_a}/|\mathcal{I}_a|$, which is the indicator vector for samples that come from template $a$. The following technical claim is a more robust version of this observation.

**Claim D.3.** *Let $\boldsymbol{x}$ be a string that matches template $\boldsymbol{z}_a$. Suppose that $0 < \lambda < \tau := \min_{j \in [r]} |\mathcal{I}_j|/\|\boldsymbol{N}^{-1}\|$. Then $(\hat{\boldsymbol{K}}^{ideal} + \lambda \boldsymbol{I})$ is invertible and the following are satisfied*

$$\|(\hat{\boldsymbol{K}}^{ideal} + \lambda \boldsymbol{I})^{-1} \boldsymbol{v}^{ideal}(\boldsymbol{x})\| \leq \sqrt{\frac{1}{|\mathcal{I}_a|}(\frac{\tau}{\tau - \lambda})} \,,$$

*and, letting $\mathbf{1}_{\mathcal{I}_a} \in \mathbb{R}^n$ be the indicator vector for set $\mathcal{I}_a$,*

$$\|\frac{\mathbf{1}_{\mathcal{I}_a}}{|\mathcal{I}_a|} - (\hat{\boldsymbol{K}}^{ideal} + \lambda \boldsymbol{I})^{-1} \boldsymbol{v}^{ideal}(\boldsymbol{x})\| \leq \sqrt{\frac{1}{|\mathcal{I}_a|}(\frac{\tau}{\tau - \lambda} - 1)} \,.$$

Using the above technical claim, we can prove that $\hat{f}^{ideal}$ is an accurate estimator. The insight is that since $(\hat{\boldsymbol{K}}^{ideal} + \lambda \boldsymbol{I})^{-1} \boldsymbol{v}^{ideal}(\boldsymbol{x})$ is approximately the indicator vector $\mathbf{1}_{\mathcal{I}_a}/|\mathcal{I}_a|$ for samples corresponding to template $a$, the output of the idealized estimator is the average of the labels for samples corresponding to template $a$.

**Claim D.4** (Idealized estimator gets vanishing test loss on unseen symbols). *There are $c, C > 0$ depending only on $|\mathcal{W}|, \mu_{\mathsf{tmplt}}, \sigma, \|K\|_\infty$ such that the following holds for any $0 < \lambda < cn/\|\boldsymbol{N}^{-1}\|$. Let $\boldsymbol{x}$ be any string that matches template $\boldsymbol{z} \in \mathrm{supp}(\mu_{\mathsf{tmplt}})$. Then, for any $\delta > 0$, with probability $\geq 1 - \delta - \exp(-cn)$ over the random samples, the idealized estimator has error upper-bounded by*

$$|\hat{f}^{ideal}(\boldsymbol{x}) - f_*(\boldsymbol{z})| \leq C\sqrt{\frac{\log(1/\delta)}{n}} \,.$$

*Proof of Claim D.4.* Let $E_1$ be the event that $|\mathcal{I}_j| \geq n\mu_{\mathsf{tmplt}}(\boldsymbol{z}_j)/2$ for all $j \in [r]$, i.e., all templates are well-represented in the dataset. By a Hoeffding bound,

$$\mathbb{P}[E_1] \geq 1 - \exp(-cn).$$

Suppose that $\boldsymbol{x}$ matches template $\boldsymbol{z}_a$. By Claim D.3, under event $E_1$, there is a constant $C > 0$ such that

$$|\hat{f}^{ideal}(\boldsymbol{x}) - f_*(\boldsymbol{z}_a)| = |\boldsymbol{y}^T(\hat{\boldsymbol{K}}^{ideal} + \lambda \boldsymbol{I})^{-1} \boldsymbol{v}^{ideal}(\boldsymbol{x}) - f_*(\boldsymbol{z}_a)|$$

$$\leq |\boldsymbol{y}^T \frac{\mathbf{1}_{\mathcal{I}_a}}{|\mathcal{I}_a|} - f_*(\boldsymbol{z}_a)| + \sqrt{\frac{1}{|\mathcal{I}_a|}(\frac{\tau}{\tau - \lambda} - 1)}$$

$$\leq |\boldsymbol{y}^T \frac{\mathbf{1}_{\mathcal{I}_a}}{|\mathcal{I}_a|} - f_*(\boldsymbol{z}_a)| + C\sqrt{\frac{1}{n}} \,.$$

We conclude since $\mathbb{P}[|\boldsymbol{y}^T \frac{\mathbf{1}_{\mathcal{I}_a}}{|\mathcal{I}_a|} - f_*(\boldsymbol{z}_a)| > C\sqrt{\frac{\log(1/\delta)}{n}} \mid E_1] \leq \delta$ by a tail bound for Gaussians. $\square$

**Putting the elements together to conclude the proof of the lemma** Combined, Claims D.2 and D.4 imply the lemma if we take $\lambda = \Theta(n)$, then we obtain error $O(\sqrt{\log(1/\delta)/n} + \sqrt{1/\rho})$ with probability at least $1 - \delta - \exp(-\Omega(n))$. $\square$

## D.1 DEFERRED PROOFS OF CLAIMS

*Proof of Claim D.3.* Let $\boldsymbol{w}_1, \ldots, \boldsymbol{w}_n$ be an orthogonal basis of eigenvectors for $\hat{\boldsymbol{K}}^{ideal}$ with eigenvalues $\nu_1, \ldots, \nu_n$. Notice that these are also eigenvectors of $\hat{\boldsymbol{K}}^{ideal} + \lambda \boldsymbol{I}$. Because of the block structure of $\hat{\boldsymbol{K}}^{ideal}$, its eigenvectors and eigenvalues have a simple form. Define

$$\boldsymbol{M} = \mathrm{diag}([\sqrt{|\mathcal{I}_1|}, \ldots, \sqrt{|\mathcal{I}_r|}]) \boldsymbol{N} \mathrm{diag}([\sqrt{|\mathcal{I}_1|}, \ldots, \sqrt{|\mathcal{I}_r|}]) \,.$$

The nonzero eigenvalues of $\hat{\boldsymbol{K}}^{ideal}$ correspond to the nonzero eigenvalues of $\boldsymbol{M}$, because for any eigenvector $\boldsymbol{u} \in \mathbb{R}^r$ of $\boldsymbol{M}$ there is a corresponding eigenvector of $\hat{\boldsymbol{K}}^{ideal}$ with the same eigenvalue by letting each of the blocks $\mathcal{I}_j$ consist of copies of the entry $u_j/\sqrt{|\mathcal{I}_j|}$. Therefore, all nonzero eigenvalues of $\hat{\boldsymbol{K}}^{-1}$ have magnitude at least

$$|\nu_1|, \ldots, |\nu_n| \geq 1/\|\boldsymbol{M}^{-1}\| \geq \min_{j \in [r]} |\mathcal{I}_j|/\|\boldsymbol{N}^{-1}\| = \tau > \lambda.$$

So $\hat{\boldsymbol{K}}^{ideal} + \lambda \boldsymbol{I}$ is invertible, which is the first part of the claim. Write $\frac{\boldsymbol{1}_{\mathcal{I}_a}}{|\mathcal{I}_a|}$ in the eigenbasis as

$$\frac{\boldsymbol{1}_{\mathcal{I}_a}}{|\mathcal{I}_a|} = \sum_i c_i \boldsymbol{w}_i \,,$$

for some coefficients $c_i$. By construction,

$$\boldsymbol{v}^{ideal}(\boldsymbol{x}) = \hat{\boldsymbol{K}}^{ideal} \frac{\boldsymbol{1}_{\mathcal{I}_a}}{|\mathcal{I}_a|} = \sum_i \nu_i c_i \boldsymbol{w}_i \,,$$

so

$$\|(\hat{\boldsymbol{K}}^{ideal} + \lambda \boldsymbol{I})^{-1} \boldsymbol{v}^{ideal}(\boldsymbol{x})\|^2 = \|\sum_i \frac{\nu_i}{\nu_i + \lambda} c_i \boldsymbol{w}_i\|^2 = \sum_i (\frac{\nu_i}{\nu_i + \lambda})^2 c_i^2$$

$$\leq \max_i (\frac{\nu_i}{\nu_i + \lambda})^2 \frac{1}{|\mathcal{I}_a|} \leq \max_i (\frac{\tau}{\tau - \lambda})^2 \,.$$

Similarly,

$$\|\frac{\boldsymbol{1}_{\mathcal{I}_a}}{|\mathcal{I}_a|} - (\hat{\boldsymbol{K}}^{ideal} + \lambda \boldsymbol{I})^{-1} \boldsymbol{v}^{ideal}(\boldsymbol{x})\|^2 = \|\sum_i (1 - \frac{\nu_i}{\nu_i + \lambda}) c_i \boldsymbol{w}_i\|^2 = \sum_i (1 - \frac{\nu_i}{\nu_i + \lambda})^2 c_i^2$$

$$\leq \max_i (1 - \frac{\nu_i}{\nu_i + \lambda})^2 \frac{1}{|\mathcal{I}_a|} \leq \max_i (1 - \frac{\tau}{\tau - \lambda})^2 \,.$$

$\square$

**Claim D.5** (Bound on difference between kernel regressions). *Suppose that $\hat{\boldsymbol{K}}$ is p.s.d and that $(\hat{\boldsymbol{K}}^{ideal} + \lambda \boldsymbol{I})^{-1} \boldsymbol{v}^{ideal}(\boldsymbol{x})$ is well-defined. Then, for any $\lambda > 0$,*

$$|\hat{f}^{ideal}(\boldsymbol{x}) - \hat{f}(\boldsymbol{x})| \leq \frac{\|\boldsymbol{y}\|}{\lambda} (\|\boldsymbol{v}^{ideal}(\boldsymbol{x}) - \boldsymbol{v}(\boldsymbol{x})\| + \|\hat{\boldsymbol{K}} - \hat{\boldsymbol{K}}^{ideal}\| \|(\hat{\boldsymbol{K}}^{ideal} + \lambda \boldsymbol{I})^{-1} \boldsymbol{v}^{ideal}(\boldsymbol{x})\|)$$

*Proof of Claim D.5.* By triangle inequality,

$$|\hat{f}(\boldsymbol{x}) - \hat{f}^{ideal}(\boldsymbol{x})| = \|\boldsymbol{y}^T(\hat{\boldsymbol{K}} + \lambda \boldsymbol{I})^{-1} \boldsymbol{v}(\boldsymbol{x}) - \boldsymbol{y}^T(\hat{\boldsymbol{K}}^{ideal} + \lambda \boldsymbol{I})^{-1} \boldsymbol{v}^{ideal}(\boldsymbol{x})\|$$

$$\overset{(a)}{\leq} \|\boldsymbol{y}\| \cdot \underbrace{\|(\hat{\boldsymbol{K}} + \lambda \boldsymbol{I})^{-1} \boldsymbol{v}(\boldsymbol{x}) - (\hat{\boldsymbol{K}} + \lambda \boldsymbol{I})^{-1} \boldsymbol{v}^{ideal}(\boldsymbol{x})\|}_{\text{Term 1}}$$

$$+ \|\boldsymbol{y}\| \cdot \underbrace{\|(\hat{\boldsymbol{K}} + \lambda \boldsymbol{I})^{-1} \boldsymbol{v}^{ideal}(\boldsymbol{x}) - (\hat{\boldsymbol{K}}^{ideal} + \lambda \boldsymbol{I})^{-1} \boldsymbol{v}^{ideal}(\boldsymbol{x})\|}_{\text{Term 2}}$$

The first term can be upper-bounded because $\|(\hat{\boldsymbol{K}} + \lambda\boldsymbol{I})^{-1}\| \leq \|(\lambda\boldsymbol{I})^{-1}\| = 1/\lambda$, so

$$\text{Term 1} \leq \frac{\|\boldsymbol{v}^{ideal}(\boldsymbol{x}) - \boldsymbol{v}(\boldsymbol{x})\|}{\lambda}$$

The second term can be upper-bounded by

$$\text{Term 2} = \|(\hat{\boldsymbol{K}} + \lambda\boldsymbol{I})^{-1}((\hat{\boldsymbol{K}} + \lambda\boldsymbol{I})(\hat{\boldsymbol{K}}^{ideal} + \lambda\boldsymbol{I})^{-1} - (\hat{\boldsymbol{K}}^{ideal} + \lambda\boldsymbol{I})(\hat{\boldsymbol{K}}^{ideal} + \lambda\boldsymbol{I})^{-1})\boldsymbol{v}^{ideal}(\boldsymbol{x})\|$$

$$= \|(\hat{\boldsymbol{K}} + \lambda\boldsymbol{I})^{-1}(\hat{\boldsymbol{K}} - \hat{\boldsymbol{K}}^{ideal})(\hat{\boldsymbol{K}}^{ideal} + \lambda\boldsymbol{I})^{-1}\boldsymbol{v}^{ideal}(\boldsymbol{x})\|$$

$$\leq \frac{1}{\lambda}\|\hat{\boldsymbol{K}} - \hat{\boldsymbol{K}}^{ideal}\|\|(\hat{\boldsymbol{K}}^{ideal} + \lambda\boldsymbol{I})^{-1}\boldsymbol{v}^{ideal}(\boldsymbol{x})\|.$$

$\square$

*Proof of Claim D.2.* Let $E_1$ be the event that $|\mathcal{I}_j| \geq n\mu_{\mathsf{tmplt}}(\boldsymbol{z}_j)$ for all $j \in [r]$. By Hoeffding, there is a constant $c > 0$ such that $\mathbb{P}[E_1] \geq 1 - \exp(-cn)$. By Claim D.3, under event $E_1$, there is a constant $C > 0$ such that

$$\|(\hat{\boldsymbol{K}}^{ideal} + \lambda\boldsymbol{I})^{-1}\boldsymbol{v}^{ideal}(\boldsymbol{x})\| \leq \frac{C}{\sqrt{n}}. \tag{18}$$

Next, recall the parameter $\rho$ used to measure the spread of the substitution map distributions $\{\mu_{sub,\boldsymbol{z}}\}_{\boldsymbol{z}\in\mathrm{supp}(\mu_{\mathsf{tmplt}})}$, as defined in (3.3). For each $i \in [n]$, let $s_i : \mathcal{W} \to \mathcal{X}$ be the substitution map used to generate the sample $\boldsymbol{x}_i$. Let $P_1$ be the number of samples $(i, i')$ such that their substitution maps overlap, or have range that overlaps with the regular tokens in the templates. Formally:

$$P_1 = |\{1 \leq i < i' \leq n : s_i(\mathcal{W}) \cap s_{i'}(\mathcal{W}) \neq \emptyset \text{ or } s_i(\mathcal{W}) \cap \mathcal{R} \neq \emptyset \text{ or } s_{i'}(\mathcal{W}) \cap \mathcal{R} \neq \emptyset\}|.$$

Similarly, let $P_2$ be the number of samples that $(i, i')$ such that their substitution maps overlap with that used to generate $\boldsymbol{x}$, or they overlap with the regular tokens in the templates:

$$P_2 = |\{1 \leq i \leq n : s_i(\mathcal{W}) \cap \mathcal{R} \neq \emptyset \text{ or } s_i(\mathcal{W}) \cap \{x_j\}_{j\in[k]} \neq \emptyset\}|.$$

By the definition of $\rho$, we can upper-bound the expected number of "bad" pairs $P_1$ and "bad" indices $P_2$ by:

$$\mathbb{E}[P_1] \leq \left(\sum_{i,i'\in[n]}\sum_{w,w'\in\mathcal{W}}\mathbb{P}[s_i(w) = s_{i'}(w')]\right) + n\sum_{i\in[n]}\sum_{t\in\mathcal{R}}\mathbb{P}[t \in s_i(\mathcal{W})] \leq \frac{Cn^2}{\rho} + \frac{Cn}{\rho} \leq \frac{Cn^2}{\rho}$$

$$\mathbb{E}[P_2] \leq \sum_{i\in[n]}\sum_{t\in\{x_j\}_{j\in[k]}\cup\mathcal{R}}\mathbb{P}[t \in s_i(\mathcal{W})] \leq \frac{Cn}{\rho}.$$

By Hoeffding's inequality, the event $E_2$ that $P_1 \leq \frac{Cn^2}{\rho}$ and $P_2 \leq \frac{Cn}{\rho}$ occurs with probability $\geq 1 - \exp(-cn)$. Under event $E_2$,

$$\|\hat{\boldsymbol{K}} - \hat{\boldsymbol{K}}^{ideal}\| \leq C + Cn/\sqrt{\rho} \quad \text{and} \quad \|\boldsymbol{v}(\boldsymbol{x}) - \boldsymbol{v}^{ideal}(\boldsymbol{x})\| \leq C\sqrt{n/\rho}. \tag{19}$$

By Claim D.5 and (18) and (19), under events $E_1, E_2$, and using that $\|\boldsymbol{y}\| \leq C\sqrt{n}$, we have

$$|\hat{f}^{ideal}(\boldsymbol{x}) - \hat{f}(\boldsymbol{x})| \leq \frac{C\sqrt{n}}{\lambda}(C\sqrt{n/\rho} + (C + Cn/\sqrt{\rho})\frac{C}{\sqrt{n}}) \leq \frac{C}{\lambda} + \frac{Cn}{\lambda\sqrt{\rho}}.$$

## D.2 Remark: explicit dependence on $\|\boldsymbol{N}^{-1}\|$

In the case that $\rho = \infty$, let us obtain explicit dependence on $\|\boldsymbol{N}^{-1}\|$ in the bound of Lemma D.1.

**Lemma D.6.** *Suppose that $K$ is token-symmetric and $\boldsymbol{N}$ is nonsingular. Suppose also that $\rho = \infty$. Then there are constants $0 < c < C$ and $0 < c' < C'$ depending only on $\mu_{\mathsf{tmplt}}$, $\sigma$, $|\mathcal{W}|$, and $\|K\|_\infty = \max_{\boldsymbol{x}} K(\boldsymbol{x},\boldsymbol{x})$ such that the following holds. Consider any regularization parameter $\lambda \in [c'n/\|\boldsymbol{N}^{-1}\|, C'n/\|\boldsymbol{N}^{-1}\|]$, and any string $\boldsymbol{x}$ matching template $\boldsymbol{z} \in \mathrm{supp}(\mu_{\mathsf{tmplt}})$. Then with probability $\geq 1 - \delta - \exp(-cn)$, the kernel ridge regression estimator $\hat{f}$ achieves good accuracy on $\boldsymbol{x}$:*

$$|\hat{f}(\boldsymbol{x}) - f_*(\boldsymbol{z})| \leq C\sqrt{\frac{\log(1/\delta)}{n}} + C\frac{\|\boldsymbol{N}^{-1}\|}{n}.$$

*Proof.* First, by Claim D.2, we have $|\hat{f}^{ideal}(\boldsymbol{x}) - \hat{f}(\boldsymbol{x})| \leq \frac{C}{\lambda}$. Next, by Claim D.4, we have $|\hat{f}^{ideal}(\boldsymbol{x}) - f_*(\boldsymbol{z})| \leq C\sqrt{\frac{\log(1/\delta)}{n}}$. ☐

☐

# E    NONSINGULARITY OF RANDOM FEATURES AFTER MLP LAYER (PROOF OF LEMMA 3.6)

Consider a kernel $K_2$ formed from a kernel $K_1$ as follows:

$$K_2(\boldsymbol{x}, \boldsymbol{y}) = \mathbb{E}_{u,v \sim \Sigma_1(\boldsymbol{x}, \boldsymbol{y})}[\phi(u)\phi(v)], \quad \Sigma_1(\boldsymbol{x}, \boldsymbol{y}) = \begin{bmatrix} K_1(\boldsymbol{x}, \boldsymbol{x}) & K_1(\boldsymbol{x}, \boldsymbol{y}) \\ K_1(\boldsymbol{x}, \boldsymbol{y}) & K_1(\boldsymbol{y}, \boldsymbol{y}) \end{bmatrix}.$$

Here $\phi : \mathbb{R} \to \mathbb{R}$ is a nonlinear activation function. Such a random features kernel arises in a neural network architecture by appending an infinite-width MLP layer with Gaussian initialization to a neural network with random features with kernel $K_1$.

We wish to prove that a certain matrix $N \in \mathbb{R}^{r \times r}$ given by

$$N_{ij} = K_2(\boldsymbol{x}_i, \boldsymbol{y}_j), \tag{20}$$

is nonsingular, where $\boldsymbol{x}_1, \ldots, \boldsymbol{x}_r, \boldsymbol{y}_1, \ldots, \boldsymbol{y}_r$ are inputs. The intuition is that if $\phi$ is a "generic" activation function, then only a weak condition on $K_1$ is required for the matrix $N$ to be invertible. We provide a general lemma that allows us to guarantee the invertibility if the activation function is a shifted cosine, although we conjecture such a result to be true for most non-polynomial activation functions $\phi$. This is a generalization of Lemma 3.6, so it implies Lemma 3.6.

**Lemma E.1** (Criterion for invertibility of $N$). *Consider the matrix $N \in \mathbb{R}^{r \times r}$ defined in* (20) *where* $\boldsymbol{x}_1, \ldots, \boldsymbol{x}_r$ *and* $\boldsymbol{y}_1, \ldots, \boldsymbol{y}_r$ *are inputs. Suppose that for all nontrivial permutations* $\tau \in S_r \setminus \{\mathrm{id}\}$ *we have*

$$\sum_{i \in [r]} K_1(\boldsymbol{x}_i, \boldsymbol{y}_i) \neq \sum_{i \in [r]} K_1(\boldsymbol{x}_i, \boldsymbol{y}_{\tau(i)}). \tag{21}$$

*Suppose also that the MLP activation function is* $\phi(t) = \cos(kt + c)$ *for two hyperparameters* $k, c$. *Then, $N$ is nonsingular for all* $(k, c) \in \mathbb{R}^2$ *except for a Lebesgue-measure-zero subset of* $\mathbb{R}^2$.

*Proof.* Let $f(k, c) := \det(N)$. We wish to show that $\{(k, c) : f(k, c) = 0\}$ is a measure-zero set. By Claim E.2, is an analytic function of $c$ and $k$, and by the identity theorem for analytic functions (Mityagin, 2020), it suffices to show that $f \not\equiv 0$. Fixing $c = \pi/4$, by Claim E.2,

$$K_2(\boldsymbol{x}, \boldsymbol{y}) = \frac{1}{2}\exp(-\frac{k^2}{2}(K_1(\boldsymbol{x}, \boldsymbol{x}) + K_1(\boldsymbol{y}, \boldsymbol{y}) - 2K_1(\boldsymbol{x}, \boldsymbol{y}))).$$

Therefore

$$\begin{aligned} f(k, \pi/4) &= \sum_{\tau \in S_r} \mathrm{sgn}(\tau) \prod_{i \in [r]} K_2(\boldsymbol{x}_i, \boldsymbol{y}_{\tau(i)}) \\ &= e^{-\frac{k^2}{2}(\sum_{i \in [r]} K_1(\boldsymbol{x}_i, \boldsymbol{x}_i) + K_1(\boldsymbol{y}_i, \boldsymbol{y}_i))} \sum_{\tau \in S_r} \mathrm{sgn}(\tau) \exp(k^2 \sum_{i \in [r]} K_1(\boldsymbol{x}_i, \boldsymbol{y}_{\tau(i)})). \end{aligned}$$

It remains to prove that as a function of $k$ we have

$$\sum_{\tau \in S_r} \mathrm{sgn}(\tau) \exp(k^2 \sum_{i \in [r]} K_1(\boldsymbol{x}_i, \boldsymbol{y}_{\tau(i)})) \not\equiv 0,$$

This holds because for any distinct $c_1, \dots, c_l$ the functions $\exp(c_1 t), \dots, \exp(c_l t)$ are linearly independent functions of $t$, since their Wronskian is a rescaled Vandermonde determinant

$$\begin{vmatrix} \exp(c_1 t) & \dots & \exp(c_l t) \\ \frac{d}{dt}\exp(c_1 t) & \dots & \frac{d}{dt}\exp(c_l t) \\ \vdots & & \vdots \\ \frac{d^{l-1}}{dt^{l-1}}\exp(c_1 t) & \dots & \frac{d^{l-1}}{dt^{l-1}}\exp(c_l t) \end{vmatrix} = \exp(\sum_{i=1}^l c_i t) \begin{vmatrix} 1 & \dots & 1 \\ c_1 & \dots & c_l \\ \vdots & & \vdots \\ c_1^{l-1} & \dots & c_l^{l-1} \end{vmatrix}$$

$$= \exp(\sum_{i=1}^l c_i t) \prod_{1 \le i < j \le l} (c_j - c_i) \not\equiv 0$$

$\square$

Below is the technical claim used in the proof of the lemma.

**Claim E.2.** *Let* $U, V \sim N(0, \begin{bmatrix} a & \rho \\ \rho & b \end{bmatrix})$. *Then for any* $k, c \in \mathbb{R}$,

$$\mathbb{E}[\cos(kU + c)\cos(kV + c)] = \frac{1}{2}e^{-\frac{1}{2}k^2(a+b)}(e^{-k^2\rho}\cos(2c) + e^{k^2\rho}).$$

*Proof.* By Mathematica, we have the following Gaussian integrals

$$\mathbb{E}[e^{ikU+ikV}] = \mathbb{E}[e^{-ikU-ikV}] = e^{-\frac{1}{2}k^2(a+b+2\rho)},$$
$$\mathbb{E}[e^{ikU-ikV}] = \mathbb{E}[e^{-ikU+ikV}] = e^{-\frac{1}{2}k^2(a+b-2\rho)}.$$

Since $\cos(kt + c) = (e^{ikt+ic} + e^{-ikt-ic})/2$,

$$\mathbb{E}[\cos(kU + c)\cos(kV + c)] = \frac{1}{4}\mathbb{E}[(e^{ikU+ic} + e^{-ikU-ic})(e^{ikV+ic} + e^{-ikV-ic})]$$

$$= \frac{1}{4}(e^{-\frac{1}{2}k^2(a+b+2\rho)}(e^{2ic} + e^{-2ic}) + 2e^{-\frac{1}{2}k^2(a+b-2\rho)})$$

$$= \frac{1}{2}e^{-\frac{1}{2}k^2(a+b)}(e^{-k^2\rho}\cos(2c) + e^{k^2\rho}).$$

$\square$

# F    ANALYSIS OF ATTENTION LAYER FEATURES (PROOF OF LEMMA 3.7)

For any inputs $X, Y$, we write the kernel of the random features of the attention layer as

$$K_{\mathsf{attn}}(X, Y) = \mathbb{E}_{\boldsymbol{m}(\boldsymbol{X}),\boldsymbol{m}(\boldsymbol{Y})}[\mathrm{smax}(\beta\boldsymbol{m}(\boldsymbol{X}))^T(\boldsymbol{X}\boldsymbol{Y}^T + \gamma^2\boldsymbol{I})\mathrm{smax}(\beta\boldsymbol{m}(\boldsymbol{Y}))]$$

$$\boldsymbol{m}(\boldsymbol{X}), \boldsymbol{m}(\boldsymbol{Y}) \sim N(\boldsymbol{0}, \begin{bmatrix} \boldsymbol{X}\boldsymbol{X}^T + \gamma^2\boldsymbol{I} & \boldsymbol{X}\boldsymbol{Y}^T + \gamma^2\boldsymbol{I} \\ \boldsymbol{Y}\boldsymbol{X}^T + \gamma^2\boldsymbol{I} & \boldsymbol{Y}\boldsymbol{Y}^T + \gamma^2\boldsymbol{I} \end{bmatrix}),$$

as stated Section 3.1; see also Section H for the derivation of this kernel in the infinite-width limit of the transformer architecture. For shorthand, we write $\kappa_{\boldsymbol{X},\boldsymbol{Y}}(\beta, \gamma) = K_{\mathsf{attn}}(\boldsymbol{X}, \boldsymbol{Y})$ to emphasize the attention kernel's dependence on the hyperparameters $\beta$ and $\gamma$ which control the softmax's inverse temperature and the weight of the positional embeddings, respectively.

We prove Lemma 3.7, which is that $K_{\mathsf{attn}}$ satisfies the property (10) required by Lemma 3.6 for the transformer random features kernel to succeed at the template task.

Namely, consider any disjoint templates $\boldsymbol{z}_1, \dots, \boldsymbol{z}_r$ and two substitution maps $s, s' : \mathcal{W} \to \mathcal{X}$

- that have disjoint range: $s(\mathcal{W}) \cap s'(\mathcal{W}) = \emptyset$,
- and the substituted tokens do not overlap with any of the tokens in the templates: $s(\mathcal{W}) \cap \mathcal{R} = s'(\mathcal{W}) \cap \mathcal{R} = \emptyset$ where $\mathcal{R} = \cup_{i \in [r], j \in [k]}\{z_j^{(i)}\}$.

Then we define $\boldsymbol{X}_i, \boldsymbol{Y}_i \in \mathbb{R}^{k \times m}$ to be the strings (where we abuse notation slightly by viewing them as matrices with one-hot rows) after substituting $\boldsymbol{z}_i$ by $s, s'$ respectively:

$$\boldsymbol{X}_i = \mathrm{sub}(\boldsymbol{z}_i, s) \quad \boldsymbol{Y}_i = \mathrm{sub}(\boldsymbol{z}_i, s').$$

**Lemma F.1** (Restatement of Lemma 3.7). *Define $g_\tau(\beta, \gamma) = \sum_{i \in [r]} \kappa_{\boldsymbol{X}_i, \boldsymbol{Y}_{\tau(i)}}(\beta, \gamma)$. Then for all but a Lebesgue-measure-zero set of $(\beta, \gamma) \in \mathbb{R}^2$ we have $g_{\mathrm{id}}(\beta, \gamma) \neq g_\tau(\beta, \gamma)$ for all permutations $\tau \neq \mathrm{id}$.*

No closed-form expression is known for $\kappa_{\boldsymbol{X}, \boldsymbol{Y}}(\beta, \gamma)$, so our approach is to analyze its Taylor series expansion around $\beta = \gamma = 0$. Our proof proceeds in stages, where, in each stage, we examine a higher derivative and progressively narrow the set of $\tau$ that might possibly have $g_\tau(\beta, \gamma) = g_{\mathrm{id}}(\beta, \gamma)$. In Section F.1, we list certain low-order derivatives of $\kappa_{\boldsymbol{X}, \boldsymbol{Y}}(\beta, \gamma)$ that will be sufficient for our analysis. In Section F.2, we analyze some of the terms in these expressions. In Section F.3 we put the previous lemmas together to prove Lemma F.1.

To avoid notational overload, in this section we will not use bolded notation to refer to the matrices $\boldsymbol{X}, \boldsymbol{Y}$, but rather use the lowercase $X, Y$.

### F.1 LOW-ORDER DERIVATIVES OF ATTENTION KERNEL

In the following table we collect several relevant derivatives of $\frac{\partial^i}{\partial \beta^i} \frac{\partial^j}{\partial \gamma^j} \kappa_{X,Y}(0,0)$ for $i \leq 6$ and $j \leq 4$. For each $i, j$ we use $c_1, c_2, \ldots$ to denote constants that depend only on $k$, and on the derivative $i, j$ being computed. Certain constants that are important for the proof are provided explicitly. These derivatives were computed using a Python script available in our code. The colors are explained in Section F.2.

| Derivative | Expansion |
|---|---|
| $\kappa_{X,Y}(0,0) =$ | $+c_1 1^T X Y^T 1$ |
| $\frac{\partial^2}{\partial \beta^2} \frac{\partial^2}{\partial \gamma^2} \kappa_{X,Y}(0,0) =$ | $+c_1 1^T X Y^T 1 + c_2 tr(XY^T)$ |
| $\frac{\partial^4}{\partial \beta^4} \kappa_{X,Y}(0,0) =$ | $+c_1 1^T X Y^T 1 + c_2 1^T X X^T X Y^T 1 + c_3 1^T X Y^T Y Y^T 1$ 
 $+c_4 1^T X X^T X X^T X Y^T 1 + c_5 (1^T X Y^T 1)(1^T X X^T 1)$ 
 $+c_6 1^T X Y^T Y X^T X Y^T 1 + c_7 (1^T X Y^T 1)(1^T X Y^T 1)$ 
 $+c_8 1^T Y X^T X Y^T Y Y^T 1 + c_9 (1^T X Y^T 1)(1^T Y Y^T 1)$ 
 $+c_{10} (1^T X X^T X Y^T 1)(1^T X X^T 1) + c_{11} (1^T X Y^T Y Y^T 1)(1^T X X^T 1)$ 
 $+c_{12} (1^T X Y^T 1)(1^T X X^T X Y^T 1) + c_{13} (1^T X Y^T Y Y^T 1)(1^T X Y^T 1)$ 
 $+c_{14} (1^T X X^T X Y^T 1)(1^T Y Y^T 1) + c_{15} (1^T X Y^T Y Y^T 1)(1^T Y Y^T 1)$ 
 $+c_{16} (1^T X Y^T 1)(1^T X X^T 1)(1^T X X^T 1) + c_{17} (1^T X Y^T 1)(1^T X X^T X X^T 1)$ 
 $+c_{18} (1^T X Y^T 1)(1^T X Y^T 1)(1^T X X^T 1)$ 
 $+c_{19} (1^T X Y^T 1)(1^T X Y^T 1)(1^T X Y^T 1)$ 
 $+c_{20} (1^T X Y^T 1)(1^T X X^T 1)(1^T Y Y^T 1)$ 
 $+c_{21} (1^T X Y^T 1)(1^T X Y^T 1)(1^T Y Y^T 1)$ 
 $+c_{22} (1^T X Y^T 1)(1^T Y Y^T 1)(1^T Y Y^T 1) + c_{23} (1^T X Y^T 1)(1^T Y Y^T Y Y^T 1)$ |
| $\frac{\partial^4}{\partial \beta^4} \frac{\partial^2}{\partial \gamma^2} \kappa_{X,Y}(0,0) =$ | $+c_1 1^T X Y^T 1 + c_2 tr(XY^T) + c_3 1^T X X^T X Y^T 1 + c_4 tr(XX^T XY^T)$ 
 $+c_5 1^T X Y^T Y Y^T 1 + c_6 tr(XY^T Y Y^T) + c_7 (1^T X Y^T 1)(1^T X X^T 1)$ 
 $+c_8 (tr(XY^T))(1^T X X^T 1) + c_9 (1^T X Y^T 1)(1^T X Y^T 1)$ 
 $+c_{10} (1^T X Y^T 1)(tr(XY^T)) + c_{11} (1^T X Y^T 1)(1^T Y Y^T 1)$ 
 $+c_{12} 1^T X Y^T X Y^T 1 + c_{13} (tr(XY^T))(1^T Y Y^T 1) + c_{14} 1^T Y X^T Y Y^T 1$ 
 $+c_{15} 1^T X X^T Y X^T 1 + c_{16} 1^T X X^T Y Y^T 1 + c_{17} (1^T Y Y^T 1)(1^T X X^T 1)$ |

$$
\begin{aligned}
\frac{\partial^6}{\partial\beta^6}\frac{\partial^4}{\partial\gamma^4}\kappa_{X,Y}(0,0) = \ & +c_1 1^T X Y^T 1 \ +c_2 tr(XY^T) \ +c_3 1^T XX^T XY^T 1 \ +c_4 tr(XX^T XY^T) \\
& +c_5 1^T XY^T YY^T 1 \ +c_6 tr(XY^T YY^T) \ +c_7 (1^T XY^T 1)(1^T XX^T 1) \\
& +c_8 (tr(XY^T))(1^T XX^T 1) \ +c_9 (tr(XY^T))(1^T XY^T 1) \\
& +c_{10}(1^T XY^T 1)(1^T YY^T 1) \ +c_{11}(1^T XY^T 1)(1^T XY^T 1) \\
& +c_{12} 1^T XY^T XY^T 1 \ +c_{13}(tr(XY^T))(1^T YY^T 1) \ +c_{14} 1^T XX^T YX^T 1 \\
& +c_{15} 1^T YX^T YY^T 1 \ +c_{16} tr(XY^T XY^T) \ +c_{17}(tr(XY^T))(tr(XY^T)) \ +c_{18} \\
& +c_{19} 1^T XX^T 1 \ +c_{20} 1^T XX^T XX^T 1 \ +c_{21} 1^T XX^T YY^T 1 \ +c_{22} 1^T YY^T 1 \\
& +c_{23}(1^T XX^T 1)(1^T XX^T 1) \ +c_{24}(1^T YY^T 1)(1^T XX^T 1) \\
& +c_{25} tr(XX^T YY^T) \ +c_{26} 1^T YY^T YY^T 1 \ +c_{27}(1^T YY^T 1)(1^T YY^T 1)
\end{aligned}
$$

Furthermore,

- in the expression for $\kappa_{X,Y}(0,0)$ we have $c_1 = 1/k^2 > 0$,

- in the expression for $\frac{\partial^2}{\partial\beta^2}\frac{\partial^2}{\partial\gamma^2}\kappa_{X,Y}(0,0)$, we have $c_2 = 8/k^2 > 0$,

- in the expression for $\frac{\partial^4}{\partial\beta^4}\kappa_{X,Y}(0,0)$, we have $c_{20} = 24/k^6 > 0$,

- in the expression for $\frac{\partial^4}{\partial\beta^4}\frac{\partial^2}{\partial\gamma^2}\kappa_{X,Y}(0,0)$, we have $c_{16} = 48/k^4 > 0$,

- and in the expression for $\frac{\partial^6}{\partial\beta^6}\frac{\partial^4}{\partial\gamma^4}\kappa_{X,Y}(0,0)$, we have $c_{25} = 17280/k^4 > 0$.

## F.2 SIMPLIFYING TERMS

Let $X \in \mathbb{R}^{k\times m}$ and $Y \in \mathbb{R}^{k\times m}$ be matrices with one-hot rows (i.e., all entries are zero except for one).

For the submatrix corresponding to rows $S$ and columns $T$, we use the notation $[X]_{S\times T} \in \mathbb{R}^{S\times T}$. If $v$ is a vector, then the subvector consisting of indices $I$ is $[v]_I$.

Let $\mathcal{R} \subseteq [m]$ be a set containing the intersection of the column support of $X$ and $Y$: i.e., for all $i \in [m] \setminus \mathcal{R}$, either $[X]_{[k]\times i} = \mathbf{0}$ or $[Y]_{[k]\times i} = \mathbf{0}$. We analyze the terms in the expressions of Section F.1 below.

### F.2.1 ASSUMING $[1^T X]_\mathcal{R} = [1^T Y]_\mathcal{R}$

Suppose that $[1^T X]_\mathcal{R} = [1^T Y]_\mathcal{R}$. Then any of the pink terms can be written as a function of only $X$ or only $Y$.

- $1^T XY^T 1 = \|[1^T X]_\mathcal{R}\|^2$
- $1^T XX^T XY^T 1 = 1^T X\mathrm{diag}(1^T X)Y^T 1 = (1^T X)^{\odot 2} \cdot (1^T Y) = \|[1^T X]_\mathcal{R}\|_3^3$
- $1^T XY^T YY^T 1 = 1^T X\mathrm{diag}(1^T Y)Y^T 1 = (1^T X) \cdot (1^T Y)^{\odot 2} = \|[1^T X]_\mathcal{R}\|_3^3$
- $1^T XX^T XX^T XY^T 1 = 1^T X\mathrm{diag}(1^T X)\mathrm{diag}(1^T X)Y^T 1 = \|[1^T X]_\mathcal{R}\|_4^4$
- $1^T XY^T YX^T XY^T 1 = 1^T X\mathrm{diag}(1^T Y)\mathrm{diag}(1^T X)Y^T 1 = \|[1^T X]_\mathcal{R}\|_4^4$
- $1^T YX^T XY^T YY^T 1 = 1^T Y\mathrm{diag}(1^T X)\mathrm{diag}(1^T Y)Y^T 1 = \|[1^T X]_\mathcal{R}\|_4^4$
- $\mathrm{trace}(XX^T XY^T) = \mathrm{trace}(X\mathrm{diag}(1^T X)Y^T) = \sum_{i\in[k]}\sum_{v\in[m]} X_{iv}(1^T X)_v Y_{iv} = \sum_{i\in[k]}\sum_{v\in\mathcal{R}} X_{iv}(1^T X)_v = 1^T X\mathrm{diag}(1^T X)1_\mathcal{R} = \|[1^T X]_\mathcal{R}\|^2$
- $\mathrm{trace}(XY^T YY^T) = \|[1^T Y]_\mathcal{R}\|^2 = \|[1^T X]_\mathcal{R}\|^2$

### F.2.2 ASSUMING $[X]_{[k]\times\mathcal{R}} = [Y]_{[k]\times\mathcal{R}}$

Suppose that $X_{[k]\times\mathcal{R}} = Y_{[k]\times\mathcal{R}}$ (i.e., the restriction of $X$ and $Y$ to the $\mathcal{R}$ rows is equal). Then any of the orange terms can be written as a function of only $X$ or only $Y$.

- $tr(XY^T) = \sum_{v\in[m]}\sum_{i\in[k]} X_{iv}Y_{iv} = \sum_{v\in\mathcal{R}}\sum_{i\in[k]} X_{iv}^2 = 1^T X1_\mathcal{R} = 1^T Y1_\mathcal{R}$

- $1^T X Y^T X Y^T 1 = \sum_{a,b,c \in [k]} 1(x_a = y_b)1(x_b = y_c) = 1^T X_{[k] \times \mathcal{R}}(Y_{[k] \times \mathcal{R}})^T X_{[k] \times \mathcal{R}}(Y_{[k] \times \mathcal{R}})^T 1$
  $= 1^T X_{[k] \times \mathcal{R}}(X_{[k] \times \mathcal{R}})^T X_{[k] \times \mathcal{R}}(X_{[k] \times \mathcal{R}})^T$

- $1^T X X^T Y X^T 1 = \sum_{a,b,c} 1(x_a = x_b)1(y_b = x_c) = \sum_{a,b,c} 1(x_a = x_b)1(y_b = x_c \in \mathcal{R})$
  $= \sum_{a,b,c} 1(x_a = x_b \in \mathcal{R})1(y_b = x_c \in \mathcal{R}) = \sum_{a,b,c} 1(x_a = x_b \in \mathcal{R})1(x_b = x_c \in \mathcal{R}) = 1^T X_{[k] \times \mathcal{R}}(X_{[k] \times \mathcal{R}})^T X_{[k] \times \mathcal{R}}(X_{[k] \times \mathcal{R}})^T 1$

- $1^T Y X^T Y Y^T 1 = 1^T X_{[k] \times \mathcal{R}}(X_{[k] \times \mathcal{R}})^T X_{[k] \times \mathcal{R}}(X_{[k] \times \mathcal{R}})^T 1$

- $\text{trace}(X Y^T X Y^T) = \sum_{a,b} 1(x_a = y_b)1(x_b = y_a) = \sum_{a,b} 1(x_a = y_b \in \mathcal{R})1(x_b = y_a \in \mathcal{R}) = \sum_{a,b} 1(x_a = x_b \in \mathcal{R}) = \text{trace}((X_{[k] \times \mathcal{R}})(X_{[k] \times \mathcal{R}})^T)$

### F.2.3 Assuming $1^T X X^T 1 = 1^T Y Y^T 1$

Suppose that $1^T X X^T 1 = 1^T Y Y^T 1$. Then any of the blue terms can be written as a function of only $X$ or only $Y$.

- $1^T X X^T 1 = 1^T Y Y^T 1$
- $1^T Y Y^T 1 = 1^T X X^T 1$

### F.2.4 Assuming $1^T X X^T = 1^T Y Y^T$

Suppose that $1^T X X^T = 1^T Y Y^T$. Then any of the teal terms can be written as a function of only $X$ or only $Y$.

- $1^T X X^T Y Y^T 1 = \|1^T X X^T\|^2 = \|1^T Y Y^T\|^2$

### F.3 Proof of Lemma F.1

We combine the above calculations to prove Lemma F.1.

*Proof.* By the technical Lemma G.1, we know that $g_\tau(\beta, \gamma)$ is an analytic function for each $\tau$. Therefore, by the identity theorem for analytic functions (Mityagin, 2020), it suffices to show that for each $\tau \in S_r \setminus \{\text{id}\}$ we have $g_{id}(\beta, \gamma) \not\equiv g_\tau(\beta, \gamma)$.

*Stage 1. Matching regular token degree distributions.*

**Claim F.2.** *If $g_{id}(0, 0) = g_\tau(0, 0)$, then $[1^T X_i]_\mathcal{R} = [1^T Y_{\tau(i)}]_\mathcal{R}$ for all $i \in [r]$.*

*Proof.* From the table in Section F.1, there is a positive constant $c_1 > 0$ such that

$$
\begin{aligned}
g_\tau(0, 0) &= c_1 \sum_{i \in [r]} 1^T X_i Y_{\tau(i)}^T 1 = c_1 \sum_{i \in [r]} [1^T X_i]_\mathcal{R} [Y_{\tau(i)}^T 1]_\mathcal{R} \\
&\overset{(a)}{\leq} \sum_{i \in [r]} \|[1^T X_i]_\mathcal{R}\| \|[1^T Y_{\tau(i)}]_\mathcal{R}\| \\
&\overset{(b)}{\leq} \sqrt{\sum_{i \in [r]} \|[1^T X_i]_\mathcal{R}\|^2} \sqrt{\sum_{i \in [r]} \|[1^T Y_{\tau(i)}]_\mathcal{R}\|^2} \\
&= \sum_{i \in [r]} \|[1^T X_i]_\mathcal{R}\|^2,
\end{aligned}
$$

where (a) is by Cauchy-Schwarz and holds with equality if and only if $[1^T X_i]_R \propto [1^T Y_{\tau(i)}]_R$ for all $i$. Similarly (b) is by Cauchy-Schwarz and holds with equality if and only if $\|[1^T X_i]_R\| = \|[1^T Y_{\tau(i)}]_R\|$ for all $i$. Notice that (a) and (b) hold with equality if $\tau = \text{id}$, since $[1^T X_i]_R = [1^T Y_i]_R$ for all $i$. $\square$

*Stage 2. Matching regular token positions.*

**Claim F.3.** *If $\frac{\partial^2}{\partial\beta^2}\frac{\partial^2}{\partial\gamma^2}g_\tau(0,0) = \frac{\partial^2}{\partial\beta^2}\frac{\partial^2}{\partial\gamma^2}g_{\mathrm{id}}(0,0)$ and $[1^T X_i]_\mathcal{R} = [1^T Y_{\tau(i)}]_\mathcal{R}$ for all $i \in [r]$, then we must have $[X_i]_{[k]\times\mathcal{R}} = [Y_{\tau(i)}]_{[k]\times\mathcal{R}}$ for all $i \in [r]$.*

*Proof.* For a constant $c_2 > 0$,

$$\frac{\partial^2}{\partial\beta^2}\frac{\partial^2}{\partial\gamma^2}g_\tau(0,0) = \sum_{i\in[r]} c_1 1^T X_i Y_{\tau(i)}^T 1 + c_2 \mathrm{trace}(X_i Y_{\tau(i)}^T)$$

$$= \left(c_1 \sum_{i\in[r]} \|[1^T X_i]_\mathcal{R}\|^2\right) + \left(c_2 \sum_{i\in[r]} \mathrm{trace}(X_i (Y^{\tau(i)})^T)\right),$$

by the calculation in Section F.2.1. The first sum does not depend on $\tau$, so we analyze the second sum. Here,

$$c_2 \sum_{i\in[r]} \mathrm{trace}(X_i Y_{\tau(i)}^T) = c_2 \sum_{i\in[r]} \sum_{a\in[k]} [X_i Y_{\tau(i)}^T]_{aa}$$

$$= c_2 \sum_{i\in[r]} \sum_{v\in\mathcal{R}} \sum_{a\in[k]} [X_i]_{av}[Y_{\tau(i)}]_{av}$$

$$\overset{(a)}{\le} c_2 \sqrt{(\sum_{i\in[r]}\sum_{v\in\mathcal{R}}\sum_{a\in[k]}([X_i]_{av})^2)(\sum_{i\in[r]}\sum_{v\in\mathcal{R}}\sum_{a\in[k]}([Y_{\tau(i)}]_{av})^2}$$

$$= c_2 \sum_{i\in[r]} 1^T X_i 1_\mathcal{R},$$

where (a) is by Cauchy-Schwarz and holds with equality if and only if $X_{av}^{(i)} = c Y_{av}^{(\tau(i))}$ for some constant $c$. We must have $c = 1$ because of the CLS token, so (a) holds with equality if and only if $[X_i]_{[k]\times\mathcal{R}} = [Y_{\tau(i)}]_{[k]\times\mathcal{R}}$ for all $i \in [r]$. Specifically (a) holds with equality if $\tau = \mathrm{id}$. $\square$

*Stage 3. Matching wildcard token degree histogram norm.*

**Claim F.4.** *Suppose that $[1^T X_i]_\mathcal{R} = [1^T Y_{\tau(i)}]_\mathcal{R}$, and that $\frac{\partial^4}{\partial\beta^4}g_\tau(0,0) = \frac{\partial^4}{\partial\beta^4}g_{\mathrm{id}}(0,0)$. Then $1^T X_i X_i^T 1 = 1^T Y_{\tau(i)} Y_{\tau(i)}^T 1$ for all $i \in [r]$.*

*Proof.* Use $[1^T X_i]_\mathcal{R} = [1^T Y_{\tau(i)}]_\mathcal{R}$ and the calculations in Section F.2.1 for the pink terms. Every term of $\frac{\partial^4}{\partial\beta^4}g_\tau(0,0)$ can be written as depending only on one of $X_i$ or $Y_{\tau(i)}$, with the exception of the $c_{20}$ term. Namely, we have

$$\frac{\partial^4}{\partial\beta^4}g_\tau(0,0) = \sum_{i\in[r]} a(X_i) + b(Y_{\tau(i)})$$
$$+ c_{20}(1^T X_i Y_{\tau(i)}^T 1)(1^T X_i X_i^T)(1^T Y_{\tau(i)} Y_{\tau(i)}^T 1),$$

for some functions $a, b$. Since $\tau$ is a permutation, only the term with coefficient $c_{20}$ depends on $\tau$. Here, $c_{20} > 0$. This term corresponds to

$$c_{20} \sum_{i\in[r]} (1^T X_i Y_{\tau(i)}^T 1)(1^T X_i X_i^T 1)(1^T Y_{\tau(i)} Y_{\tau(i)}^T 1)$$

$$= c_{20} \sum_{i\in[r]} \|[1^T X_i]_\mathcal{R}\|\|1^T Y_{\tau(i)}]_\mathcal{R}\|(1^T X_i X_i^T 1)(1^T Y_{\tau(i)} Y_{\tau(i)}^T 1)$$

$$\overset{(a)}{\le} \sqrt{(\sum_{i\in[r]}\|[1^T X_i]_\mathcal{R}\|^2(1^T X_i X_i^T 1)^2)(\sum_{i\in[r]}\|1^T Y_{\tau(i)}]_\mathcal{R}\|^2(1^T Y_{\tau(i)} Y_{\tau(i)}^T 1)^2}$$

$$= \sum_{i\in[r]} \|[1^T X_i]_\mathcal{R}\|^2(1^T X_i X_i^T 1)^2$$

where (a) is by Cauchy-Schwarz and holds with equality if and only if $\|[1^T X_i]_{\mathcal{R}}\|^2 1^T X_i X_i 1 = c\|[1^T Y_{\tau(i)}]_{\mathcal{R}}\|^2 1^T Y_{\tau(i)} Y_{\tau(i)}^T 1$ for all $i$ and some constant $c$. This constant $c = 1$ because the former is a permutation of the latter over $i \in [r]$. Since $\|[1^T X_i]_{\mathcal{R}}\|^2 = \|[1^T Y_i]_{\mathcal{R}}\|^2 \geq 1$ by assumption and since we have the CLS token, we know that (a) holds with equality if and only if $1^T X_i X_i^T 1 = 1^T Y_{\tau(i)} Y_{\tau(i)}^T 1$ for all $i \in [r]$. This is the case for $\tau = \mathrm{id}$ by construction of $X_i$ and $Y_i$. □

*Stage 4. Matching wildcard degree distributions.*

**Claim F.5.** *Suppose that* $[X_i]_{[k] \times \mathcal{R}} = [Y_{\tau(i)}]_{[k] \times \mathcal{R}}$ *and* $1^T X_i X_i^T 1 = 1^T Y_{\tau(i)} Y_{\tau(i)}^T 1$ *for all* $i \in [r]$. *Suppose also that* $\frac{\partial^4}{\partial \beta^4} \frac{\partial^2}{\partial \gamma^2} g_\tau(0,0) = \frac{\partial^4}{\partial \beta^4} \frac{\partial^2}{\partial \gamma^2} g_{\mathrm{id}}(0,0)$. *Then* $1^T X_i X_i^T = 1^T Y_{\tau(i)} Y_{\tau(i)}^T$ *for all* $i \in [r]$.

*Proof.* Similarly to the proof of the previous claim, because of the calculations in Sections F.2.1, F.2.2 and F.2.3 for the pink, orange, and blue terms, respectively, we can write $\frac{\partial^4}{\partial \beta^4} \frac{\partial^2}{\partial \gamma^2}$ as a sum of terms that each depends on either $X_i$ or $Y_{\tau(i)}$, plus $\sum_{i \in [r]} c_{16} 1^T X_i X_i^T Y_{\tau(i)} Y_{\tau(i)}^T 1$. This latter sum is the only term that depends on $\tau$, and the constant $c_{16}$ satisfies $c_{16} > 0$. Similarly to the previous claim, by Cauchy-Schwarz

$$\sum_{i \in [r]} c_{16} 1^T X_i X_i^T Y_{\tau(i)} Y_{\tau(i)}^T 1 \leq \sum_{i \in [r]} c_{16} \|1^T X_i X_i^T\| \|Y_{\tau(i)} Y_{\tau(i)}^T 1\|,$$

with equality if and only if $1^T X_i X_i^T = 1^T Y_{\tau(i)} Y_{\tau(i)}^T$ for all $i$, since $\{X_i X_i^T\}_i$ is a permutation of $\{Y_{\tau(i)} Y_{\tau(i)}^T\}_i$. This condition holds for $\tau = \mathrm{id}$. □

*Stage 5. Matching wildcard positions.*

**Claim F.6.** *Suppose that* $[X_i]_{[k] \times \mathcal{R}} = [Y_{\tau(i)}]_{[k] \times \mathcal{R}}$ *and* $1^T X_i X_i^T = 1^T Y_{\tau(i)} Y_{\tau(i)}^T$ *for all* $i \in [r]$. *Suppose also that* $\frac{\partial^6}{\partial \beta^6} \frac{\partial^4}{\partial \gamma^4} g_\tau(0,0) = \frac{\partial^6}{\partial \beta^6} \frac{\partial^4}{\partial \gamma^4} g_{\mathrm{id}}(0,0)$. *Then* $X_i X_i^T = Y_{\tau(i)} Y_{\tau(i)}^T$ *for all* $i \in [r]$.

*Proof.* Write $\frac{\partial^6}{\partial \beta^6} \frac{\partial^4}{\partial \gamma^4} g_\tau(0,0)$ as a sum of terms each depending only on either $X_i$ or $Y_{\tau(i)}$ by using the calculations in Sections F.2.1, F.2.3, F.2.2, and F.2.4 to handle the pink, orange, blue, and teal terms, plus (for $c_{25} > 0$),

$$\sum_{i \in [r]} c_{25} \mathrm{trace}(X_i X_i^T Y_{\tau(i)} Y_{\tau(i)}^T) \leq \sum_{i \in [r]} c_{25} \|X_i X_i^T\|_F \|Y_{\tau(i)} Y_{\tau(i)}^T\|_F,$$

with equality if and only if $X_i X_i^T = Y_{\tau(i)} Y_{\tau(i)}^T$ for all $i \in [r]$. This equality holds if $\tau = \mathrm{id}$, concluding the claim. □

Combine the above four claims to conclude that if $g_\tau(\beta, \gamma) \equiv g_{\mathrm{id}}(\beta, \gamma)$, then we have $X_i X_i^T = Y_{\tau(i)} Y_{\tau(i)}^T$ and $[X_i]_{[k] \times \mathcal{R}} = [Y_{\tau(i)}]_{[k] \times \mathcal{R}}$ for all $i$, so $\tau = \mathrm{id}$. □

## G  ANALYTICITY OF ATTENTION KERNEL (TECHNICAL RESULT)

We prove the analyticity of $\kappa_{\boldsymbol{X}, \tilde{\boldsymbol{X}}}(\beta, \gamma) = K_{\mathrm{attn}}^{\beta, \gamma}(\boldsymbol{X}, \tilde{\boldsymbol{X}})$ as function of $\beta$ and $\gamma$.

**Lemma G.1** (Analyticity of $K_{\mathrm{attn}}$). *For any* $\boldsymbol{X}, \tilde{\boldsymbol{X}}$, *the function* $\kappa_{\boldsymbol{X}, \tilde{\boldsymbol{X}}}$ *is analytic in* $\mathbb{R}^2$.

*Proof.* Note that we can write

$$\boldsymbol{m} := \boldsymbol{m}(\boldsymbol{X}) = \boldsymbol{X} \boldsymbol{\zeta} + \gamma \boldsymbol{p}, \quad \tilde{\boldsymbol{m}} := \boldsymbol{m}(\tilde{\boldsymbol{X}}) = \tilde{\boldsymbol{X}} \tilde{\boldsymbol{\zeta}} + \gamma \boldsymbol{p},$$

where $\boldsymbol{\zeta}, \tilde{\boldsymbol{\zeta}} \sim \mathcal{N}(0, I_m)$ and $\boldsymbol{p} \sim \mathcal{N}(0, I_k)$ are independent Gaussians. So we can rewrite $\kappa_{\boldsymbol{X}, \tilde{\boldsymbol{X}}}$ as

$$\kappa_{\boldsymbol{X}, \tilde{\boldsymbol{X}}}(\beta, \gamma) = \mathbb{E}_{\boldsymbol{\zeta}, \tilde{\boldsymbol{\zeta}}, \boldsymbol{p}}[f(\beta, \gamma; \boldsymbol{\zeta}, \tilde{\boldsymbol{\zeta}}, \boldsymbol{p})],$$

where

$$f(\beta, \gamma; \boldsymbol{\zeta}, \tilde{\boldsymbol{\zeta}}, \boldsymbol{p}) = \boldsymbol{s}^T (\boldsymbol{X}\tilde{\boldsymbol{X}}^T + \gamma^2 \boldsymbol{I})\tilde{\boldsymbol{s}}.$$

and

$$\boldsymbol{s} = \mathrm{smax}(\beta \boldsymbol{X}\boldsymbol{\zeta} + \beta\gamma\boldsymbol{p})^T, \quad \tilde{\boldsymbol{s}} = \mathrm{smax}(\beta \tilde{\boldsymbol{X}}\tilde{\boldsymbol{\zeta}} + \beta\gamma\boldsymbol{p}).$$

The main obstacle is to prove the technical Lemma G.9, which states that for any $k_1, k_2$, we have

$$\mathbb{E}_{\boldsymbol{\zeta},\tilde{\boldsymbol{\zeta}},\boldsymbol{p}}[|\frac{\partial^{k_1}}{\partial\beta^{k_1}}\frac{\partial^{k_2}}{\partial\gamma^{k_2}}f(\beta,\gamma;\boldsymbol{\zeta},\tilde{\boldsymbol{\zeta}},\boldsymbol{p})|] \le C(1+\gamma^2)k_1!k_2!(C(|\beta|+|\gamma|)^{k_1+k_2})$$

So by smoothness of $f$ and dominated convergence, we know that we can differentiate under the integral sign, and

$$|\frac{d^{k_1}}{d\beta^{k_1}}\frac{d^{k_2}}{d\gamma^{k_2}}\kappa_{\boldsymbol{X},\boldsymbol{X}'}(\beta,\gamma)| = |\mathbb{E}_{\boldsymbol{\zeta},\tilde{\boldsymbol{\zeta}},\boldsymbol{p}}[\frac{\partial^{k_1}}{\partial\beta^{k_1}}\frac{\partial^{k_2}}{\partial\gamma^{k_2}}f(\beta,\gamma;\boldsymbol{X},\tilde{\boldsymbol{X}},\boldsymbol{\zeta},\tilde{\boldsymbol{\zeta}},\boldsymbol{p})]|$$
$$\le C(1+\gamma^2)k_1!k_2!(C(|\beta|+|\gamma|)^{k_1+k_2}).$$

Because of the bound on the derivatives and its smoothness, $\kappa_{\boldsymbol{X},\boldsymbol{X}'}(\beta,\gamma)$ is real-analytic. □

The proof of the technical bound in Lemma G.9 is developed in the subsections below.

## G.1 TECHNICAL LEMMAS FOR QUANTIFYING POWER SERIES CONVERGENCE

In order to show that the values of the attention kernel are real-analytic functions of in terms of $\beta, \gamma$, we will need to make quantitative certain facts about how real-analyticity of is preserved under compositions, products, and sums. For this, we introduce the notion of the convergence-type of a real-analytic function.

**Definition G.2** (Quantifying power series convergence in real-analytic functions). Let $U \subseteq \mathbb{R}^m$ be an open set. We say that a real-analytic function $f : U \to \mathbb{R}$ has $(\tau_1, \tau_2)$-type for functions $\tau_1 : U \to \mathbb{R}_{>0}$ and $\tau_2 : U \to \mathbb{R}_{>0}$ if the following holds. For any $\boldsymbol{\zeta}_0$, consider the power series of $f$ around $\boldsymbol{\zeta}_0$,

$$\sum_{\mu} a_{\boldsymbol{\zeta}_0,\mu}(\boldsymbol{\zeta} - \boldsymbol{\zeta}_0)^{\mu}.$$

Then for any $\boldsymbol{\zeta}$ such that $\|\boldsymbol{\zeta} - \boldsymbol{\zeta}_0\|_\infty \le \tau_1(\boldsymbol{\zeta}_0)$ this power series converges absolutely.

$$\sum_{\mu \text{ s.t. } |\mu|\ge 1} |a_{\boldsymbol{\zeta}_0,\mu}||\boldsymbol{\zeta} - \boldsymbol{\zeta}_0|^{\mu} \le \tau_2(\boldsymbol{\zeta}_0).$$

We provide rules for how convergence type is affected by compositions, products, and sums.

**Lemma G.3** (Composition rule for type; quantitative version of Proposition 2.2.8 of (Krantz & Parks, 2002)). *Let $U \subseteq \mathbb{R}^m$ and let $V \subseteq \mathbb{R}$ be open. Let $f_1, \ldots, f_n : U \to V$ be real-analytic with $(\tau_1, \tau_2)$-type, and let $g : V^n \to \mathbb{R}$ be real-analytic with $(\sigma_1, \sigma_2)$-type. Then the composition $h = g \circ (f_1, \ldots, f_n)$ is real-analytic with $(\min(\tau_1, (\sigma_1 \circ f) \cdot \frac{\tau_1}{\tau_2}), \sigma_2 \circ f)$-type.*

*Proof.* Fix some $\boldsymbol{\zeta}_0$ and let $\boldsymbol{y}_0 = [f_1(\boldsymbol{\zeta}_0), \ldots, f_n(\boldsymbol{\zeta}_0)]$, and let $a^{(i)}_{\boldsymbol{\zeta}_0,\mu}$ be the coefficients of the power series expansion for $f_i$ around $\boldsymbol{\zeta}_0$. Define $\rho = \min(1, \sigma_1(y_0)/\tau_2(\boldsymbol{\zeta}_0))$. Then, for any $\boldsymbol{\zeta}$ such that $\|\boldsymbol{\zeta} - \boldsymbol{\zeta}_0\|_\infty \le \rho\tau_1(\boldsymbol{\zeta}_0)$ and $i \in [n]$ we have

$$\sum_{\mu \text{ s.t. } |\mu|\ge 1} |a^{(i)}_{\boldsymbol{\zeta}_0,\mu}||\boldsymbol{\zeta} - \boldsymbol{\zeta}_0|^{\mu} \le \sum_{\mu \text{ s.t. } |\mu|\ge 1} |a^{(i)}_{\boldsymbol{\zeta}_0,\mu}|\rho^{|\mu|}\tau_1(\boldsymbol{\zeta}_0)^{|\mu|} \le \rho\tau_2(\boldsymbol{\zeta}_0) \le \sigma_1(y_0).$$

So, letting $\sum_{\nu}^{\infty} b_{\boldsymbol{y}_0,\nu}(\boldsymbol{y} - \boldsymbol{y}_0)^{\nu}$ be the series expansion of $g$ around $\boldsymbol{y}_0$, we have the following absolute convergence

$$\sum_{\nu, \text{ s.t. } |\nu|\ge 1}^{\infty} b_{\boldsymbol{y}_0,\nu} \prod_{i=1}^{n} \left| \sum_{\mu \text{ s.t. } |\mu|\ge 1} |a^{(i)}_{\boldsymbol{\zeta}_0,\mu}||\boldsymbol{\zeta} - \boldsymbol{\zeta}_0|^{\mu} \right|^{\nu_i} \le \sigma_2(y_0).$$

So we may rearrange the terms of

$$\sum_{\nu}^{\infty} b_{\boldsymbol{y}_0, \nu} \prod_{i=1}^{n} \left( \sum_{\mu \text{ s.t. } |\mu| \geq 1} a_{\boldsymbol{\zeta}_0, \mu}^{(i)} (\boldsymbol{\zeta} - \boldsymbol{\zeta}_0)^{\mu} \right)^{\nu_i} .$$

as we please, and we get an absolutely convergent series for $g \circ f$ around $\boldsymbol{\zeta}_0$. $\qquad\square$

**Lemma G.4** (Sum and product rules for type). *Let $f : \mathbb{R}^m \to \mathbb{R}$ and $g : \mathbb{R}^m \to \mathbb{R}$ be real-analytic functions of $(\tau_1, \tau_2)$-type and $(\sigma_1, \sigma_2)$-type respectively. Then $h = f + g$ is real-analytic of $(\min(\tau_1, \sigma_1), \tau_2 + \tau_2)$-type, and $h = fg$ is real-analytic of $(\min(\tau_1, \sigma_1), \tau_2 \sigma_2 + \tau_2 |g| + |f| \sigma_2)$-type*

*Proof.* Both of these are straightforward from the definition.

$\qquad\square$

**Lemma G.5** (Derivative bound based on type). *Let $f : \mathbb{R}^m \to \mathbb{R}$ be real-analytic with $(\tau_1, \tau_2)$-type. Then, for any multi-index $\mu$,*

$$|\frac{\partial^{|\mu|}}{\partial \zeta^{\mu}} f(\boldsymbol{\zeta}_0)| \leq \frac{\tau_2(\boldsymbol{\zeta}_0)}{\tau_1(\boldsymbol{\zeta}_0)^{|\mu|}} \mu!$$

*Proof.* Let $a_{\boldsymbol{\zeta}_0, \mu}$ be the coefficients of the power series of $f$ at $\boldsymbol{\zeta}_0$. Since $f$ is of $(\tau_1, \tau_2)$-type, we have

$$\sum_{\mu \text{ s.t. } |\mu| \geq 1} |a_{\boldsymbol{\zeta}_0, \mu}| |\tau_1(\boldsymbol{\zeta}_0)|^{|\mu|} \leq \tau_2(\boldsymbol{\zeta}_0) .$$

Since all terms in the sum are nonnegative, for all $\mu$ with $|\mu| \geq 1$,

$$|a_{\boldsymbol{\zeta}_0, \mu}| \leq \tau_2(\boldsymbol{\zeta}_0) \cdot (1/\tau_1(\boldsymbol{\zeta}_0))^{|\mu|} .$$

The lemma follows by Remark 2.2.4 of Krantz & Parks (2002), which states $\frac{\partial^{|\mu|}}{\partial \zeta^{\nu}} f(\boldsymbol{\zeta}_0)| = |a_{\boldsymbol{\zeta}_0, \mu}| \mu!$.
$\qquad\square$

## G.2    APPLICATION OF TECHNICAL LEMMAS TO ATTENTION KERNEL

We now use the above general technical lemmas to specifically prove that the attention kernel is analytic in terms of $\beta$ and $\gamma$.

**Lemma G.6.** *For any $j \in [m]$, the function $f : \mathbb{R}^m \to \mathbb{R}$ given by $f(\boldsymbol{\zeta}) = \text{smax}(\boldsymbol{\zeta})_j$ is real-analytic of $(1/(2e^2), 1)$-type*

*Proof.* Write $f = g \circ h$ for $g : \mathbb{R}_{>0} \to \mathbb{R}$ and $h : \mathbb{R}^k \to \mathbb{R}_{>0}$ given by $g(y) = 1/y$, and $h(\boldsymbol{\zeta}) = \sum_{i=1}^{m} e^{\zeta_i - \zeta_j}$.

The power expansion of $g(y)$ around $y_0 \in \mathbb{R}_{>0}$, is given by

$$g(y) = \sum_{k=0}^{\infty} \frac{(-1)^{k+1}}{y_0^{k+1}} (y - y_0)^k ,$$

so one can see that $g$ is of $(\rho_1, \rho_2)$-type for $\rho_1(y_0) = y_0/2$ and $\rho_2(y_0) = 1/y_0$. Finally, write the series expansion for $h(\boldsymbol{\zeta})$ around $\boldsymbol{\zeta}_0$

$$h(\boldsymbol{\zeta}) = 1 + e^{-\zeta_j} \sum_{i \in [m] \setminus \{j\}} e^{\zeta_i} = 1 + \sum_{i \in [m] \setminus \{j\}} (\sum_{l=0}^{\infty} e^{-\zeta_{0,j}} \frac{(\zeta_{0,j} - \zeta_j)^l}{l!})(\sum_{k=0}^{\infty} e^{\zeta_{0,i}} \frac{(\zeta_i - \zeta_{0,i})^k}{k!})$$

Note that this expansion converges absolutely for all $\boldsymbol{\zeta}$, as the absolute series is

$$1 + \sum_{i \in [m] \setminus \{j\}} (\sum_{l=0}^{\infty} e^{-\zeta_{0,j}} \frac{|\zeta_{0,j} - \zeta_j|^l}{l!})(\sum_{k=0}^{\infty} e^{\zeta_{0,i}} \frac{|\zeta_i - \zeta_{0,i}|^k}{k!})$$

$$= 1 + \sum_{i \in [m] \setminus \{j\}} e^{-\zeta_{0,j} + \zeta_{0,i} + |\zeta_i - \zeta_{0,i}| + |\zeta_j - \zeta_{0,j}|}$$

$$\leq e^{2\|\boldsymbol{\zeta} - \boldsymbol{\zeta}_0\|_\infty} h(\boldsymbol{\zeta}) .$$

Specifically, $h$ is of $(1, e^2 h)$-type. So by the composition rule of Lemma G.3, it must be that $f$ is real-analytic of $(\tau_1, \tau_2)$-type for $\tau_1 = \min(1, (\rho_1 \circ h) \cdot \frac{1}{e^2 h}) = 1/(2e^2)$ and $\tau_2 = \rho_2 \circ h = 1/h \leq 1$. $\quad\square$

**Lemma G.7.** *For any $j \in [m]$ and $\boldsymbol{X}, \boldsymbol{\zeta}, \boldsymbol{p}$, the function $f : \mathbb{R}^2 \to \mathbb{R}$ given by $f(\beta, \gamma) = \mathrm{smax}(\beta \boldsymbol{X}\boldsymbol{\zeta} + \beta\gamma\boldsymbol{p})_j$ is real-analytic of $(\min(1, 1/(2e^2 \|\boldsymbol{X}\boldsymbol{\zeta}\|_\infty + 2e^2(|\beta| + |\gamma|)\|\boldsymbol{p}\|_\infty), 1)$-type.*

*Proof.* Write $f = g \circ h$ for $g : \mathbb{R}^m \to \mathbb{R}$ and $h : \mathbb{R}^2 \to \mathbb{R}^m$ given by $g(\boldsymbol{v}) = \mathrm{smax}(\boldsymbol{v})_j$ and $h(\beta, \gamma) = \beta \boldsymbol{X}\boldsymbol{\zeta} + \beta\gamma\boldsymbol{p}$. We know from Lemma G.6 that $g$ is real-analytic of $(1/(2e^2), 1)$-type. And it is easy to see that $h$ is real-analytic of $(1, \|\boldsymbol{X}\boldsymbol{\zeta}\|_\infty + (|\beta| + |\gamma|)\|\boldsymbol{p}\|_\infty)$-type. Apply the composition rule of Lemma G.3 to conclude. $\quad\square$

**Lemma G.8.** *For any $\boldsymbol{X}, \tilde{\boldsymbol{X}}, \boldsymbol{\zeta}, \tilde{\boldsymbol{\zeta}}, \boldsymbol{p}$, the function $f : \mathbb{R}^2 \to \mathbb{R}$ given by $f(\beta, \gamma) = \mathrm{smax}(\beta \boldsymbol{X}\boldsymbol{\zeta} + \beta\gamma\boldsymbol{p})^T (\boldsymbol{X}\tilde{\boldsymbol{X}}^T + \gamma^2 \boldsymbol{I})\mathrm{smax}(\beta \tilde{\boldsymbol{X}}\tilde{\boldsymbol{\zeta}} + \beta\gamma\boldsymbol{p})$ is real-analytic and of type*

$$\left(\min(1, \frac{1}{2e^2}\frac{1}{\|\boldsymbol{X}\boldsymbol{\zeta}\|_\infty + (|\beta| + |\gamma|)\|\boldsymbol{p}\|_\infty}, \frac{1}{2e^2}\frac{1}{\|\tilde{\boldsymbol{X}}\tilde{\boldsymbol{\zeta}}\|_\infty + (|\beta| + |\gamma|)\|\boldsymbol{p}\|_\infty}), C(1 + \gamma^2)\right),$$

*where $C$ is a constant depending on the context length $k$.*

*Proof.* Each entry of $(\boldsymbol{X}\tilde{\boldsymbol{X}}^T + \gamma \boldsymbol{I})$ is real-analytic in $\gamma$ and of $(1, \gamma)$-type. So by combining with Lemma G.7 the product rule and sum rule (Lemma G.4), and the fact that each entry of the smax is at most one. $\quad\square$

As a consequence, we can bound the derivatives of $f(\beta, \gamma; \boldsymbol{X}, \tilde{\boldsymbol{X}}, \boldsymbol{\zeta}, \tilde{\boldsymbol{\zeta}}, \boldsymbol{p}) = \mathrm{smax}(\beta \boldsymbol{X}\boldsymbol{\zeta} + \beta\gamma\boldsymbol{p})^T (\boldsymbol{X}\tilde{\boldsymbol{X}}^T + \gamma^2 \boldsymbol{I})\mathrm{smax}(\beta \tilde{\boldsymbol{X}}\tilde{\boldsymbol{\zeta}} + \beta\gamma\boldsymbol{p})$, which was what we needed to prove Lemma G.1.

**Lemma G.9.** *For any $k_1, k_2 \geq 0$,*

$$\left| \frac{\partial^{k_1}}{\partial \beta^{k_1}} \frac{\partial^{k_2}}{\partial \gamma^{k_2}} f(\beta, \gamma; \boldsymbol{X}, \tilde{\boldsymbol{X}}, \boldsymbol{\zeta}, \tilde{\boldsymbol{\zeta}}, \boldsymbol{p}) \right|$$
$$\leq C(1 + \gamma^2) \max(1, ((2e^2)(\|\boldsymbol{X}\boldsymbol{\zeta}\|_\infty + \|\tilde{\boldsymbol{X}}\tilde{\boldsymbol{\zeta}}\|_\infty + (|\beta| + |\gamma|)\|\boldsymbol{p}\|_\infty))^{k_1 + k_2}) k_1! k_2! .$$

*Proof.* Direct consequence of Lemma G.5 and Lemma G.8. $\quad\square$

## H  DERIVATION OF TRANSFORMER KERNEL

We state the transformer architecture and informally derive its random features kernel in the infinite-width limit.

### H.1  TRANSFORMER ARCHITECTURE

We consider a depth-1 transformer architecture (without skip connections or layernorm, for simplicity). This architecture has $H$ heads, each with parameters $\boldsymbol{W}_{K,h}, \boldsymbol{W}_{Q,h}, \boldsymbol{W}_{V,h}, \boldsymbol{W}_{O,h} \in \mathbb{R}^{d_{head} \times d_{emb}}$, and embedding layer $\boldsymbol{W}_E \in \mathbb{R}^{m \times d_{emb}}$, positional embeddings $\boldsymbol{P} \in \mathbb{R}^{k \times d_{emb}}$, an MLP layer with parameters $\boldsymbol{W}_A, \boldsymbol{W}_B \in \mathbb{R}^{d_{mlp} \times d_{emb}}$, and a final unembedding layer with weights $\boldsymbol{w}_U \in \mathbb{R}^{d_{emb}}$. The network takes in $\boldsymbol{X} \in \mathbb{R}^{k \times m}$ and outputs

$$f_{\mathsf{trans}}(\boldsymbol{X}; \boldsymbol{\theta}) = \boldsymbol{w}_U^T \boldsymbol{z}_2 \qquad \text{(Unembedding)}$$

where

$$\boldsymbol{z}_2 = \frac{1}{\sqrt{d_{mlp}}} \boldsymbol{W}_B^T \sigma(\frac{1}{\sqrt{d_{emb}}} \boldsymbol{W}_A \boldsymbol{z}_1) \in \mathbb{R}^{d_{emb}} \qquad \text{(MLP layer)}$$

$$\boldsymbol{z}_1 = \frac{1}{\sqrt{H}} \sum_{h \in [H]} \boldsymbol{A}_h^T \boldsymbol{e}_k \in \mathbb{R}^{d_{emb}} \qquad \text{(Attention layer output at CLS token)}$$

$$\boldsymbol{A}_h = \mathrm{smax}(\frac{\beta \boldsymbol{Z}_0 \boldsymbol{W}_{K,h}^T \boldsymbol{W}_{Q,h} \boldsymbol{Z}_0^T}{d_{emb}\sqrt{d_{head}}}) \boldsymbol{Z}_0 \frac{\boldsymbol{W}_{V,h}^T \boldsymbol{W}_{O,h}}{\sqrt{d_{head} d_{emb}}} \in \mathbb{R}^{k \times d_{emb}} \qquad \text{(Attention heads)}$$

$$\boldsymbol{Z}_0 = \boldsymbol{X}\boldsymbol{W}_E + \gamma \boldsymbol{P} \in \mathbb{R}^{k \times d_{emb}} . \qquad \text{(Embedding layer)}$$

Here $\beta, \gamma \geq 0$ are two hyperparameters that control the inverse temperature of the softmax and the strength of the positional embeddings, respectively. Note that only the output of the attention layer at the final $k$th position CLS token is used, since this is a depth-1 network. The $\mathrm{smax}$ is a softmax applied row-wise.

## H.2 Random features kernel

The derivation of this kernel assumes that every string $\boldsymbol{x}$ ends with a special [CLS] classification token that does not appear elsewhere in the string. We choose that initialization so that each of the entries of the intermediate representations $\boldsymbol{Z}_0, \boldsymbol{z}_1, \boldsymbol{z}_2$ is of order $\Theta(1)$. In order to accomplish this, we initialize $\boldsymbol{W}_E, \boldsymbol{P}, \boldsymbol{W}_{K,h}, \boldsymbol{W}_{Q,h}, \boldsymbol{W}_{V,h}, \boldsymbol{W}_{O,h}, \boldsymbol{W}_A, \boldsymbol{W}_B$ with i.i.d. $N(0, 1)$ entries.

We also initialize $\boldsymbol{w}_U = 0$, and only train $\boldsymbol{w}_U$ while maintaining the rest of parameters at initialization. The random features kernel corresponding to training $\boldsymbol{w}_U$ is

$$\hat{K}_{\mathsf{trans}}(\boldsymbol{X}, \boldsymbol{Y}) = \boldsymbol{z}_2(\boldsymbol{X})^T \boldsymbol{z}_2(\boldsymbol{Y})/d_{emb} \,,$$

where we view $\boldsymbol{z}_2$ as a function of the input (either $\boldsymbol{X}$ or $\boldsymbol{Y}$), and depending on the randomly-initialized parameters of the network.

In the limit of infinitely-many heads $H$, infinite embedding dimension $d_{emb}$ and MLP dimension $d_{mlp}$ and head dimension $d_{head}$, the kernel $\hat{K}_{\mathsf{trans}}$ tends to a deterministic limit $K_{\mathsf{trans}}$, which can be recursively computed (see, e.g., Jacot et al. (2018)). Assuming that the final token of both $\boldsymbol{X}$ and $\boldsymbol{Y}$ is the same token (i.e., a CLS token), the deterministic limiting kernel $K_{\mathsf{trans}}$ is given by:

$$K_{\mathsf{trans}}(\boldsymbol{X}, \boldsymbol{Y}) = \mathbb{E}_{u,v}[\sigma(u)\sigma(v)] \text{ for } u, v \sim N(\boldsymbol{0}, \begin{bmatrix} K_{\mathsf{attn}}(\boldsymbol{X}, \boldsymbol{X}) & K_{\mathsf{attn}}(\boldsymbol{X}, \boldsymbol{Y}) \\ K_{\mathsf{attn}}(\boldsymbol{Y}, \boldsymbol{X}) & K_{\mathsf{attn}}(\boldsymbol{Y}, \boldsymbol{Y}) \end{bmatrix}) \quad (22)$$

$$\text{where } K_{\mathsf{attn}}(\boldsymbol{X}, \boldsymbol{Y}) = \mathbb{E}_{\boldsymbol{m}(\boldsymbol{X}), \boldsymbol{m}(\boldsymbol{Y})}[\mathrm{smax}(\beta \boldsymbol{m}(\boldsymbol{X}))^T (\boldsymbol{X}\boldsymbol{Y}^T + \gamma^2 \boldsymbol{I})\mathrm{smax}(\beta \boldsymbol{m}(\boldsymbol{Y}))]$$

$$\boldsymbol{m}(\boldsymbol{X}), \boldsymbol{m}(\boldsymbol{Y}) \sim N(\boldsymbol{0}, (1 + \gamma^2) \begin{bmatrix} \boldsymbol{X}\boldsymbol{X}^T + \gamma^2 \boldsymbol{I} & \boldsymbol{X}\boldsymbol{Y}^T + \gamma^2 \boldsymbol{I} \\ \boldsymbol{Y}\boldsymbol{X}^T + \gamma^2 \boldsymbol{I} & \boldsymbol{Y}\boldsymbol{Y}^T + \gamma^2 \boldsymbol{I} \end{bmatrix}) \,.$$

Notice that the covariance matrix in the above definition of the distribution of $\boldsymbol{m}(\boldsymbol{X}), \boldsymbol{m}(\boldsymbol{Y})$ is rescaled compared to that in the main text in Section 3.1, but this is inessential, since we can simply reparametrize $\beta$ as $\beta \mapsto \beta/\sqrt{1 + \gamma^2}$ to recover the expression in the main text.

## H.3 Informal derivation

We provide an informal derivation of (22) below. Informally, by law of large numbers we have the following almost sure convergence

$$\hat{K}_{\mathsf{trans}}(\boldsymbol{X}, \boldsymbol{Y}) = \frac{\boldsymbol{z}_2(\boldsymbol{X})^T \boldsymbol{z}_2(\boldsymbol{Y})}{d_{emb}} = \frac{\sigma(\frac{1}{\sqrt{d_{emb}}} \boldsymbol{W}_A \boldsymbol{z}_1(\boldsymbol{X}))^T \boldsymbol{W}_B \boldsymbol{W}_B^T \sigma(\frac{1}{\sqrt{d_{emb}}} \boldsymbol{W}_A \boldsymbol{z}_1(\boldsymbol{Y}))}{d_{emb} d_{mlp}}$$

$$\overset{d_{emb} \to \infty}{\longrightarrow} \frac{\sigma(\frac{1}{\sqrt{d_{emb}}} \boldsymbol{W}_A \boldsymbol{z}_1(\boldsymbol{X}))^T \sigma(\frac{1}{\sqrt{d_{emb}}} \boldsymbol{W}_A \boldsymbol{z}_1(\boldsymbol{Y}))}{d_{mlp}}$$

$$\overset{d_{mlp} \to \infty}{\longrightarrow} \mathbb{E}_{u,v}[\sigma(u)\sigma(v)] \text{ for } u, v \sim N(\boldsymbol{0}, \begin{bmatrix} K_{\mathsf{attn}}(\boldsymbol{X}, \boldsymbol{X}) & K_{\mathsf{attn}}(\boldsymbol{X}, \boldsymbol{Y}) \\ K_{\mathsf{attn}}(\boldsymbol{Y}, \boldsymbol{X}) & K_{\mathsf{attn}}(\boldsymbol{Y}, \boldsymbol{Y}) \end{bmatrix})$$

$$:= K_{\mathsf{trans}}(\boldsymbol{X}, \boldsymbol{Y}) \,,$$

where $K_{\text{attn}}$ is the kernel corresponding to the attention layer in the infinite-width limit, defined as:

$$
\hat{K}_{\text{attn}}(\boldsymbol{X}, \boldsymbol{Y}) := \frac{\boldsymbol{z}_1^T(\boldsymbol{X})\boldsymbol{z}_1^T(\boldsymbol{Y})}{d_{emb}} = \frac{\sum_{h,h' \in [H]} \boldsymbol{e}_k^T \boldsymbol{A}_h(\boldsymbol{X})\boldsymbol{A}_{h'}(\boldsymbol{Y})^T \boldsymbol{e}_k}{H d_{emb}}
$$

$$
= \frac{1}{H d_{head} d_{emb}^2} \sum_{h,h' \in [H]} \boldsymbol{e}_k^T \operatorname{smax}(\frac{\beta \boldsymbol{Z}_0(\boldsymbol{X})\boldsymbol{W}_{K,h}^T \boldsymbol{W}_{Q,h} \boldsymbol{Z}_0(\boldsymbol{X})^T}{d_{emb}\sqrt{d_{head}}})\boldsymbol{Z}_0(\boldsymbol{X})\boldsymbol{W}_{V,h}^T \boldsymbol{W}_{O,h}
$$

$$
\cdot \boldsymbol{W}_{O,h'}^T \boldsymbol{W}_{V,h'} \boldsymbol{Z}_0(\boldsymbol{Y})^T \operatorname{smax}(\frac{\beta \boldsymbol{Z}_0(\boldsymbol{Y})\boldsymbol{W}_{K,h'}^T \boldsymbol{W}_{Q,h'} \boldsymbol{Z}_0(\boldsymbol{Y})^T}{d_{emb}\sqrt{d_{head}}})^T \boldsymbol{e}_k
$$

$$
\overset{d_{head} \to \infty, d_{emb} \to \infty}{\to} \frac{1}{H} \sum_{h \in [H]} \boldsymbol{e}_k^T \operatorname{smax}(\frac{\beta \boldsymbol{Z}_0(\boldsymbol{X})\boldsymbol{W}_{K,h}^T \boldsymbol{W}_{Q,h} \boldsymbol{Z}_0(\boldsymbol{X})^T}{d_{emb}\sqrt{d_{head}}})(\boldsymbol{X}\boldsymbol{Y}^T + \gamma^2 \boldsymbol{I})
$$

$$
\cdot \operatorname{smax}(\frac{\beta \boldsymbol{Z}_0(\boldsymbol{Y})\boldsymbol{W}_{K,h}^T \boldsymbol{W}_{Q,h} \boldsymbol{Z}_0(\boldsymbol{Y})^T}{d_{emb}\sqrt{d_{head}}})^T \boldsymbol{e}_k
$$

$$
\overset{H \to \infty}{\to} \mathbb{E}[\boldsymbol{e}_k^T \operatorname{smax}(\frac{\beta \boldsymbol{Z}_0(\boldsymbol{X})\boldsymbol{W}_{K,h}^T \boldsymbol{W}_{Q,h} \boldsymbol{Z}_0(\boldsymbol{X})^T}{d_{emb}\sqrt{d_{head}}})(\boldsymbol{X}\boldsymbol{Y}^T + \gamma^2 \boldsymbol{I})
$$

$$
\cdot \operatorname{smax}(\frac{\beta \boldsymbol{Z}_0(\boldsymbol{Y})\boldsymbol{W}_{K,h}^T \boldsymbol{W}_{Q,h} \boldsymbol{Z}_0(\boldsymbol{Y})^T}{d_{emb}\sqrt{d_{head}}})^T \boldsymbol{e}_k]
$$

$$
= \mathbb{E}[\operatorname{smax}(\frac{\beta \boldsymbol{e}_k^T \boldsymbol{Z}_0(\boldsymbol{X})\boldsymbol{W}_{K,h}^T \boldsymbol{W}_{Q,h} \boldsymbol{Z}_0(\boldsymbol{X})^T}{d_{emb}\sqrt{d_{head}}})(\boldsymbol{X}\boldsymbol{Y}^T + \gamma^2 \boldsymbol{I})
$$

$$
\cdot \operatorname{smax}(\frac{\beta \boldsymbol{e}_k^T \boldsymbol{Z}_0(\boldsymbol{Y})\boldsymbol{W}_{K,h}^T \boldsymbol{W}_{Q,h} \boldsymbol{Z}_0(\boldsymbol{Y})^T}{d_{emb}\sqrt{d_{head}}})^T]
$$

$$
\overset{d_{emb} \to \infty, d_{head} \to \infty}{\to} \mathbb{E}_{\boldsymbol{m}(\boldsymbol{X}), \boldsymbol{m}(\boldsymbol{Y})}[\operatorname{smax}(\beta \boldsymbol{m}(\boldsymbol{X}))^T(\boldsymbol{X}\boldsymbol{Y}^T + \gamma^2 \boldsymbol{I})\operatorname{smax}(\beta \boldsymbol{m}(\boldsymbol{Y}))]
$$

$$
:= K_{\text{attn}}(\boldsymbol{X}, \boldsymbol{Y}),
$$

where

$$
\boldsymbol{m}(\boldsymbol{X}), \boldsymbol{m}(\boldsymbol{Y}) \sim N(\boldsymbol{0}, (1 + \gamma^2) \begin{bmatrix} \boldsymbol{X}\boldsymbol{X}^T + \gamma^2 \boldsymbol{I} & \boldsymbol{X}\boldsymbol{Y}^T + \gamma^2 \boldsymbol{I} \\ \boldsymbol{Y}\boldsymbol{X}^T + \gamma^2 \boldsymbol{I} & \boldsymbol{Y}\boldsymbol{Y}^T + \gamma^2 \boldsymbol{I} \end{bmatrix}),
$$

because due to the randomness in $\boldsymbol{W}_{K,h}$ and $\boldsymbol{W}_{Q,h}$ we have that

$$
\frac{\boldsymbol{Z}_0(\boldsymbol{X})\boldsymbol{W}_{Q,h}^T \boldsymbol{W}_{K,h} \boldsymbol{Z}_0(\boldsymbol{X})^T \boldsymbol{e}_k}{d_{emb}\sqrt{d_{head}}}
$$

and

$$
\frac{\boldsymbol{Z}_0(\boldsymbol{Y})\boldsymbol{W}_{Q,h}^T \boldsymbol{W}_{K,h} \boldsymbol{Z}_0(\boldsymbol{Y})^T \boldsymbol{e}_k}{d_{emb}\sqrt{d_{head}}}
$$

are jointly Gaussian with covariance:

$$
\Sigma(\boldsymbol{X}, \boldsymbol{Y}) = \mathbb{E}_{\boldsymbol{W}_{K,h}, \boldsymbol{W}_{Q,h}, \boldsymbol{W}_E, \boldsymbol{P}}[\frac{\boldsymbol{Z}_0(\boldsymbol{X})\boldsymbol{W}_{Q,h}^T \boldsymbol{W}_{K,h} \boldsymbol{Z}_0(\boldsymbol{X})^T \boldsymbol{e}_k}{d_{emb}\sqrt{d_{head}}} \frac{\boldsymbol{e}_k^T \boldsymbol{Z}_0(\boldsymbol{Y})\boldsymbol{W}_{K,h}^T \boldsymbol{W}_{Q,h} \boldsymbol{Z}_0(\boldsymbol{Y})^T}{d_{emb}\sqrt{d_{head}}}],.
$$

Since this is an expectation over products of jointly Gaussian variables, for any $i, j \in [k]$ we can calculate:

$$\Sigma_{i,j}(\boldsymbol{X}, \boldsymbol{Y}) = \mathbb{E}_{\boldsymbol{W}_E, \boldsymbol{P}}\Big[\frac{1}{d_{emb}^2} \sum_{r,s \in [d_{emb}]} [\boldsymbol{Z}_0(\boldsymbol{X})]_{ir}[\boldsymbol{Z}_0(\boldsymbol{Y})]_{js} \operatorname{trace}(\boldsymbol{Z}_0(\boldsymbol{X})^T \boldsymbol{e}_k \boldsymbol{e}_k^T \boldsymbol{Z}_0(\boldsymbol{Y}))\Big]$$

$$= \mathbb{E}_{\boldsymbol{W}_E, \boldsymbol{P}}\Big[\frac{1}{d_{emb}^2} \sum_{r,s,t \in [d_{emb}]} [\boldsymbol{Z}_0(\boldsymbol{X})]_{ir}[\boldsymbol{Z}_0(\boldsymbol{Y})]_{js}[\boldsymbol{Z}_0(\boldsymbol{X})]_{kt}[\boldsymbol{Z}_0(\boldsymbol{Y})]_{kt}\Big]$$

$$= \mathbb{E}_{\boldsymbol{W}_E, \boldsymbol{P}}\Big[\frac{1}{d_{emb}^2} \sum_{r,s,t \in [d_{emb}]} [\boldsymbol{X}\boldsymbol{W}_E + \gamma\boldsymbol{P}]_{ir}[\boldsymbol{Y}\boldsymbol{W}_E + \gamma\boldsymbol{P}]_{js}[\boldsymbol{X}\boldsymbol{W}_E + \gamma\boldsymbol{P}]_{kt}[\boldsymbol{Y}\boldsymbol{W}_E + \gamma\boldsymbol{P}]_{kt}\Big]$$

$$\stackrel{(a)}{=} \frac{1}{d_{emb}^2} \sum_{r,s \in [d_{emb}]} \mathbb{E}_{\boldsymbol{W}_E, \boldsymbol{P}}\big[[\boldsymbol{X}\boldsymbol{W}_E + \gamma\boldsymbol{P}]_{ir}[\boldsymbol{Y}\boldsymbol{W}_E + \gamma\boldsymbol{P}]_{js}\big]$$
$$\cdot \sum_{t \in [d_{emb}]} \mathbb{E}_{\boldsymbol{W}_E, \boldsymbol{P}}\big[[\boldsymbol{X}\boldsymbol{W}_E + \gamma\boldsymbol{P}]_{kt}[\boldsymbol{Y}\boldsymbol{W}_E + \gamma\boldsymbol{P}]_{kt}\big] + O(1/d_{emb})$$

$$= \frac{1}{d_{emb}} \sum_{r,s \in [d_{emb}]} \mathbb{E}_{\boldsymbol{W}_E, \boldsymbol{P}}\big[[\boldsymbol{X}\boldsymbol{W}_E + \gamma\boldsymbol{P}]_{ir}[\boldsymbol{Y}\boldsymbol{W}_E + \gamma\boldsymbol{P}]_{js}\big] \cdot (1 + \gamma^2) + O(1/d_{emb})$$

$$\stackrel{(a)}{=} \frac{1}{d_{emb}} \sum_{r \in [d_{emb}]} \mathbb{E}_{\boldsymbol{W}_E, \boldsymbol{P}}\big[[\boldsymbol{X}\boldsymbol{W}_E + \gamma\boldsymbol{P}]_{ir}[\boldsymbol{Y}\boldsymbol{W}_E + \gamma\boldsymbol{P}]_{jr}\big] \cdot (1 + \gamma^2) + O(1/d_{emb})$$

$$= [\boldsymbol{X}\boldsymbol{Y}^T]_{ij} + \gamma^2 \delta_{ij} \cdot (1 + \gamma^2) + O(1/d_{emb}),$$

where in (a) we use that $[\boldsymbol{X}\boldsymbol{W}_E + \gamma\boldsymbol{P}]_{ab}$ and $[\boldsymbol{Y}\boldsymbol{W}_E + \gamma\boldsymbol{P}]_{ab}$ are independent of $[\boldsymbol{X}\boldsymbol{W}_E + \gamma\boldsymbol{P}]_{cd}$ and $[\boldsymbol{Y}\boldsymbol{W}_E + \gamma\boldsymbol{P}]_{cd}$ unless $b = d$. So

$$\Sigma(\boldsymbol{X}, \boldsymbol{Y}) \stackrel{d_{emb} \to \infty}{\to} (1 + \gamma^2) \cdot (\boldsymbol{X}\boldsymbol{Y}^T + \gamma^2\boldsymbol{I}).$$

## I   MLPs FAIL TO GENERALIZE ON UNSEEN SYMBOLS

A natural question is whether classical architectures such as the MLP architecture (a.k.a., fully-connected network) would exhibit the same emergent reasoning properties when trained with enough data. In this section, we prove a negative result: an SGD-trained or Adam-trained MLP will not reach good test performance on the template task. This is in sharp contrast to the positive result for transformers proved in the previous section.

**MLP architecture**   The input to the MLP is a concatenation of the token one-hot encodings. The MLP alternates linear transformations and nonlinear elementwise activations. Formally, the MLP has weights $\boldsymbol{\theta} = \{\boldsymbol{W}_1, \ldots, \boldsymbol{W}_L, \boldsymbol{w}\}$ and outputs

$$f_{\mathsf{MLP}}(\boldsymbol{x}; \boldsymbol{\theta}) = \boldsymbol{w}^T \boldsymbol{z}_L(\boldsymbol{x}; \boldsymbol{\theta}) \in \mathbb{R} \quad \text{where} \tag{23}$$
$$\boldsymbol{z}_\ell(\boldsymbol{x}; \boldsymbol{\theta}) = \phi(\boldsymbol{W}_\ell \boldsymbol{z}_{\ell-1}(\boldsymbol{x}; \boldsymbol{\theta})) \in \mathbb{R}^d \quad \text{for } \ell \geq 1$$
$$\boldsymbol{z}_0(\boldsymbol{x}; \boldsymbol{\theta}) = \boldsymbol{z}_0(\boldsymbol{x}) = [\boldsymbol{e}_{x_1}, \ldots, \boldsymbol{e}_{x_k}] \in \mathbb{R}^{km}.$$

We consider training the MLP with SGD.

**Definition I.1** (One-pass SGD training). The learned weights $\boldsymbol{\theta}^t$ after $t$ steps of SGD training are the random weights given by initializing $\boldsymbol{\theta}^0$ so that each of $\boldsymbol{W}_1^0, \ldots, \boldsymbol{W}_L^0, \boldsymbol{w}^0$ have i.i.d. Gaussian entries, and then updating with $\boldsymbol{\theta}^t = \boldsymbol{\theta}^{t-1} - \eta_t \nabla_{\boldsymbol{\theta}}(f_{\mathsf{MLP}}(\boldsymbol{x}^t; \boldsymbol{\theta}) - y^t)^2 |_{\boldsymbol{\theta}=\boldsymbol{\theta}^{t-1}}$ for $(\boldsymbol{x}^t, y^t) \sim \mathcal{D}$ and some step size $\eta_t > 0$.

We show that SGD-trained MLPs fail at the template task since they do not generalize well in the case when the templates consist only of wildcard tokens. In words, if the template labels $f_*$ are a non-constant function, the MLP will not reach arbitrarily low error no matter how many training steps are taken. Let $\mathcal{X}_{uns} \subset \mathcal{X}$ be the subset of tokens not seen in the train data. We assume that $|\mathcal{X}_{uns}| \geq k$, which guarantees that for any template there is at least one string matching it where all the wildcards are substituted by tokens in $\mathcal{X}_{uns}$. Under this condition:

**Theorem I.2** (Failure of MLPs at generalizing on unseen symbols). *Suppose that the label function $f_*$ is non-constant, and that all templates in the support of $\mu_{\mathsf{tmplt}}$ consist only of wildcards: $z \in \mathcal{W}^k$ for all $z \in \mathrm{supp}(\mu_{\mathsf{tmplt}})$. Then, for any SGD step $t$ there is a string $x \in (\mathcal{X}_{uns})^k$ that matches a template $z \in \mathrm{supp}(\mu_{\mathsf{tmplt}})$ such that*

$$\mathbb{E}_{\theta^t}[(f_{\mathsf{MLP}}(x; \theta^t) - f_*(z))^2] \geq c > 0\,,$$

*where $c$ is constant that depends only on $\mu_{\mathsf{tmplt}}$ and $f_*$.*

The proof relies on the key observation that SGD-training of MLPs satisfies a permutation invariance property (Ng, 2004). This property guarantees that MLP cannot consistently distinguish between the unseen tokens, and therefore, in expectation over the weights $\theta^t$, outputs the same value for any sequence $x \in (\mathcal{X}_{uns})^k$. We make four remarks.

**Remark I.3.** MLPs are universal approximators (Cybenko, 1989), so there are choices of weights $\theta$ such that $f_{\mathsf{MLP}}(\cdot; \theta)$ has good generalization on unseen symbols. The theorem proves that these weights are not found by SGD.

**Remark I.4.** The theorem does not assume that training is in the NTK regime, i.e., it holds even for nonlinear training dynamics.

**Remark I.5.** The theorem also holds for training with Adam, gradient flow, and minibatch-SGD, since the permutation-invariance property of MLP training also holds for these.

**Remark I.6.** As a sanity check, we verify that MLP kernel does not meet the sufficient condition for generalizing on unseen symbols from Lemma 3.5. The kernel for an MLP is an inner product kernel of the form $K_{\mathsf{MLP}}(x, x') = \kappa(\sum_{i=1}^k 1(x_i = x'_i))$ for a function $\kappa : \mathbb{R} \to \mathbb{R}$. Therefore, the matrix $N \in \mathbb{R}^{r \times r}$ has all of its entries equal to $N_{ij} = \kappa(0)$, so it is singular and the condition of Lemma 3.5 is not met.

We now prove Theorem I.2. We first show that trained MLPs cannot differentiate between tokens in the set $\mathcal{X}_{uns}$. Let $\mathcal{X} = \mathcal{X}_{seen} \sqcup \mathcal{X}_{uns}$ be the partition of tokens into those seen and not seen in the train data. Here $\mathcal{X}_{seen}$ is defined as the smallest set such that $x \in \mathcal{X}_{seen}^k$ almost surely for $(x, y) \sim \mathcal{D}$.

**Lemma I.7** (Trained MLPs cannot distinguish unseen tokens). *For any number of SGD steps $t$, and any learning rate schedule $\eta_1, \ldots, \eta_t$, the learned MLP estimator cannot distinguish between sequences of unseen tokens. Formally, for any $x_1, x_2 \in \mathcal{X}_{uns}^k$, we have*

$$\mathbb{E}_{\theta^t}[f_{\mathsf{MLP}}(x_1; \theta^t)] = \mathbb{E}_{\theta^t}[f_{\mathsf{MLP}}(x_2; \theta^t)]\,.$$

*Proof of Lemma I.7.* The proof of this result is based on a well-known permutation-invariance property of MLPs trained by SGD. This property has previously been used to show sample complexity lower bounds for learning with SGD-trained MLPs (Ng, 2004; Li et al., 2020), as well as time-complexity lower bounds (Shamir, 2018; Abbe et al., 2022; Abbe & Boix-Adsera, 2022). In this lemma, we use the permutation invariance property to show poor out-of-distribution generalization of SGD-trained MLPs.

First, construct a permutation $\Pi \in \mathbb{R}^{km \times km}$ such that $\Pi z_0(x_1) = z_0(x_2)$, but which also satisfies that for any $\tilde{x} \in (\mathcal{X}_{seen})^k$ we have $\Pi z_0(\tilde{x}) = z_0(\tilde{x})$. This permutation can be easily constructed since neither $x_1$ nor $x_2$ contains tokens in $\mathcal{X}_{seen}$. Next, define the following network $f_{\mathsf{MLP}}^{\Pi}$, analogously to (23) but with the first-layer inputs permuted by $\Pi$

$$f_{\mathsf{MLP}}^{\Pi}(x; \theta) = w^T z_L^{\Pi}(x; \theta) \in \mathbb{R} \quad \text{where}$$
$$z_\ell^{\Pi}(x; \theta) = \phi(W_\ell z_{\ell-1}^{\Pi}(x; \theta)) \in \mathbb{R}^d \quad \text{for } \ell \geq 1$$
$$z_0^{\Pi}(x; \theta) = z_0^{\Pi}(x) = \Pi[e_{x_1}, \ldots, e_{x_k}] \in \mathbb{R}^{km}\,.$$

Now let us couple the weights $\theta^0, \ldots, \theta^t$ from SGD training of $f_{\mathsf{MLP}}$ on dataset $\mathcal{D}$, with the weights $\theta^{\Pi,0}, \ldots, \theta^{\Pi,t}$ from SGD training of $f_{\mathsf{MLP}}^{\Pi}$ on dataset $\mathcal{D}$. The coupling is performed inductively on the time step, and we can maintain the property that $\theta^\tau = \theta^{\Pi,\tau}$ for all $t$. For the base case $\tau = 0$, we set $\theta^0 = \theta^{\Pi,0}$. For the inductive step, $\tau \geq 1$, we update the weights with the gradient from some sample $(x^\tau, y^\tau)$. Since $x^\tau \in (\mathcal{X}^{seen})^k$ almost surely, we know that $z_0(x^\tau) = z_0^{\Pi}(x^\tau)$ almost

surely, which means that $\boldsymbol{\theta}^\tau = \boldsymbol{\theta}^{\Pi,\tau}$ almost surely. We conclude the equality in distribution of the weights

$$\boldsymbol{\theta}^t \stackrel{d}{=} \boldsymbol{\theta}^{\Pi,t} \,. \tag{24}$$

Next, let us inductively couple the weights $\boldsymbol{\theta}^0, \ldots, \boldsymbol{\theta}^t$ with the weights $\boldsymbol{\theta}^{\Pi,0}, \ldots, \boldsymbol{\theta}^{\Pi,t}$ in a different way, so as to guarantee that for any time $0 \leq \tau \leq t$, we have

$$\boldsymbol{W}_1^\tau = \boldsymbol{W}_1^{\Pi,\tau} \Pi \text{ and } \boldsymbol{W}_\ell^\tau = \boldsymbol{W}_\ell^{\Pi,\tau} \text{ for all } 2 \leq \ell \leq L \text{ and } \boldsymbol{w}^\tau = \boldsymbol{w}^{\Pi,\tau} \,.$$

almost surely. The base case $\tau = 0$ follows because the distribution of $\boldsymbol{W}_1^0$ and $\boldsymbol{W}_1^{\Pi,0}$ is equal and is also invariant to permutations since it is Gaussian. For the inductive step, couple the sample updates so that SGD draws the same sample $(\boldsymbol{x}^\tau, y^\tau) \sim \mathcal{D}$. One can see from the chain rule that the invariant is maintained. We conclude the equality in distribution of the weights

$$\boldsymbol{\theta}^t = \{\boldsymbol{W}_1^t, \ldots, \boldsymbol{W}_L^t, \boldsymbol{w}^t\} \stackrel{d}{=} \{\boldsymbol{W}_1^{\Pi,t}\Pi, \boldsymbol{W}_2^{\Pi,t}, \ldots, \boldsymbol{W}_L^{\Pi,t}, \boldsymbol{w}^{\Pi,t}\} \tag{25}$$

Combining (24) and (25), we get

$$\boldsymbol{\theta}^t = \{\boldsymbol{W}_1^t, \ldots, \boldsymbol{W}_L^t, \boldsymbol{w}^t\} \stackrel{d}{=} \{\boldsymbol{W}_1^t\Pi, \boldsymbol{W}_2^t, \ldots, \boldsymbol{W}_L^t, \boldsymbol{w}^t\} \,,$$

which,since $\Pi \boldsymbol{z}_0(\boldsymbol{x}_1) = \boldsymbol{z}_0(\boldsymbol{x}_2)$, immediately implies

$$f_{\mathsf{MLP}}(\boldsymbol{x}_1; \boldsymbol{\theta}^t) = f_{\mathsf{MLP}}(\boldsymbol{x}_2; \{\boldsymbol{W}_1^t\Pi, \boldsymbol{W}_2^t, \ldots, \boldsymbol{W}_L^t, \boldsymbol{w}^t\}) \stackrel{d}{=} f_{\mathsf{MLP}}(\boldsymbol{x}_2; \boldsymbol{\theta}^t) \,,$$

which proves the lemma. $\qquad \square$

Theorem I.2 follows as a consequence. Note that the key lemma proved above only relied on a permutation invariance property of SGD on MLPs that also holds for Adam training, gradient flow training, and SGD with minibatch (see Li et al. (2020)). Therefore, the result holds for training with those algorithms as well, beyond just SGD.

*Proof of Theorem I.2.* Pick any two templates $\boldsymbol{z}, \boldsymbol{z}' \in \operatorname{supp}(\mu_{\mathsf{tmplt}})$ such that $f_*(\boldsymbol{z}) \neq f_*(\boldsymbol{z}')$. Recall that $\boldsymbol{z}, \boldsymbol{z}' \in \mathcal{W}^k$ by assumption. Since we assumed that $|\mathcal{X}_{uns}| \geq k$, there are strings $\boldsymbol{x}, \boldsymbol{x}' \in \mathcal{X}_{uns}^k$ matching templates $\boldsymbol{z}$ and $\boldsymbol{z}'$, respectively. Furthermore, by Lemma I.7, if we define $a = \mathbb{E}_{\boldsymbol{\theta}^t}[f_{\mathsf{MLP}}(\boldsymbol{x}; \boldsymbol{\theta}^t)] = \mathbb{E}_{\boldsymbol{\theta}^t}[f_{\mathsf{MLP}}(\boldsymbol{x}'; \boldsymbol{\theta}^t)]$, we have

$$\max(\mathbb{E}_{\boldsymbol{\theta}^t}[(f_{\mathsf{MLP}}(\boldsymbol{x}; \boldsymbol{\theta}^t) - f_*(\boldsymbol{z}))^2], \mathbb{E}_{\boldsymbol{\theta}^t}[(f_{\mathsf{MLP}}(\boldsymbol{x}'; \boldsymbol{\theta}^t) - f_*(\boldsymbol{z}'))^2])$$
$$\geq \max((a - f_*(\boldsymbol{z}))^2, (a - f_*(\boldsymbol{z}'))^2)$$
$$\geq \frac{1}{4}(f_*(\boldsymbol{z}) - f_*(\boldsymbol{z}'))^2 = c > 0 \,.$$

$\qquad \square$

## J   Deferred details for next-token-prediction template tasks

### J.1   Definition of next-token-prediction template tasks

In next-token-prediction template tasks, the output is a token in $\mathcal{X}$, with the cross-entropy loss for multiclass classification. The formal definition of these tasks is:

**Definition J.1** (Multi-class prediction version of template). The data distribution $\mathcal{D}_{multiclass} = \mathcal{D}_{multiclass}(\mu_{\mathsf{tmplt}}, \{\mu_{sub,\boldsymbol{z}}\}, f_*)$ is specified by: (i) a template distribution $\mu_{\mathsf{tmplt}}$ supported on $(\mathcal{X} \cup \mathcal{W})^k$; (ii) for each template $\boldsymbol{z}$, a distribution $\mu_{sub,\boldsymbol{z}}$ over substitution maps $s : \mathcal{W} \to \mathcal{X}$; (iii) a labelling function $f_* : \operatorname{supp}(\mu_{\mathsf{tmplt}}) \to \mathcal{X} \cup \mathcal{W}$. A sample $(\boldsymbol{x}, y) \in \mathcal{X}^k \times \mathcal{X}$ drawn from $\mathcal{D}_{multiclass}$ is drawn by taking $\boldsymbol{x} = \operatorname{sub}(\boldsymbol{z}, s)$ and $y = \operatorname{sub}(f_*(\boldsymbol{z}), s)$, where $\boldsymbol{z} \sim \mu_{\mathsf{tmplt}}$ and $s \sim \mu_{sub,\boldsymbol{z}}$.

### J.2   Failure of transformers to copy and modification that succeeds

We provide the deferred proofs for Section 4.

**Attention layer architecture** For simplicity in this section we consider a transformer with the attention layer only, since the MLP layer does not play a role in the ability to copy unseen symbols. Our architecture has $H$ heads with parameters $\boldsymbol{W}_{K,h}, \boldsymbol{W}_{Q,h}, \boldsymbol{W}_{V,h}, \boldsymbol{W}_{O,h} \in \mathbb{R}^{d_{head} \times d_{emb}}$, an embedding/unembedding layer $\boldsymbol{W}_E \in \mathbb{R}^{m \times d_{emb}}$, positional embeddings $\boldsymbol{P} \in \mathbb{R}^{k \times d_{emb}}$, an MLP layer with parameters $\boldsymbol{W}_A, \boldsymbol{W}_B \in \mathbb{R}^{d_{mlp} \times d_{emb}}$, a final unembedding layer , and an activation function $\phi$. The network takes in $\boldsymbol{X} \in \mathbb{R}^{k \times m}$ and outputs

$$f_{\mathsf{attn}}(\boldsymbol{X}; \boldsymbol{\theta}) = \boldsymbol{W}_E \boldsymbol{z}_1 \in \mathbb{R}^m \qquad \text{(Unembedding layer)}$$

where

$$\boldsymbol{z}_1 = \sum_{h \in [H]} \boldsymbol{A}_h^T \boldsymbol{e}_k$$

$$\boldsymbol{A}_h = \mathrm{smax}(\beta \boldsymbol{Z}_0 \boldsymbol{W}_{K,h}^T \boldsymbol{W}_{Q,h} \boldsymbol{Z}_0^T) \boldsymbol{Z}_0 \boldsymbol{W}_{V,h}^T \boldsymbol{W}_{O,h} \in \mathbb{R}^{k \times d_{emb}} \qquad \text{(Attention heads)}$$

$$\boldsymbol{Z}_0 = \boldsymbol{X} \boldsymbol{W}_E + \gamma \boldsymbol{P} \in \mathbb{R}^{k \times d_{emb}} . \qquad \text{(Embedding layer)}$$

and we tie the embedding and unembedding weights, as often done in practice, for example in GPT-2 (Brown et al., 2020). Here $\beta, \gamma \geq 0$ are two hyperparameters that control the inverse temperature of the softmax and the strength of the positional embeddings, respectively.

**Simplification in our case** We consider here a next-token prediction setup, where there is no final [CLS] token appended to the string. Namely, given a string $\boldsymbol{x} \in \mathcal{X}^k$, this is inputted to the network as a stacked matrix of one-hot vectors for the tokens of the string $\boldsymbol{X} = [\boldsymbol{e}_{x_1}, \dots, \boldsymbol{e}_{x_k}]$. We study a very basic template task: template "$\alpha$" labeled by $\alpha$, where $\alpha$ is a wildcard. An example dataset generated from this template could be $\{(A, A), (B, B), (C, C)\}$, where $A, B, C \in \mathcal{X}$ are tokens. Because the template has length $k = 1$, $\boldsymbol{X} \in \mathbb{R}^{k \times m}$ is a one-hot vector encoding the input token. Furthermore, the softmax output is always a $1 \times 1$ matrix with the entry 1, so the architecture simplifies to

$$f_{\mathsf{attn}}(\boldsymbol{X}; \boldsymbol{\theta}) = \boldsymbol{W}_E \big( \sum_{h \in [H]} \boldsymbol{W}_{O,h}^T \boldsymbol{W}_{V,h} \big) (\boldsymbol{W}_E^T \boldsymbol{X}^T + \gamma \boldsymbol{P}^T) . \qquad (26)$$

We initialize the entries of $\boldsymbol{P}$ and $\boldsymbol{W}_E$ be i.i.d. $N(0, 1/d_{emb})$, the entries of $\boldsymbol{W}_{O,h}$ be $N(0, 1/(d_{emb}))$, and the entries of $\boldsymbol{W}_{V,h}$ be $N(0, 1/d_{head})$, so that as $d_{emb} \to \infty$ the variance of the output vanishes as $O(1/d_{emb})$ as in the mean-field scaling (Mei et al., 2018; 2019; Sirignano & Spiliopoulos, 2022; Chizat & Bach, 2018; Rotskoff & Vanden-Eijnden, 2018; Yang & Hu, 2021).

**Derivation of kernels driving dynamics at small times** Despite the simplicity of the task, the architecture does not generalize well on unseen symbols. Our evidence for this will be by analyzing the early times of training. For these times, the dynamics are governed by the neural tangent kernel (NTK) of the network at initialization (Jacot et al., 2018; Chizat et al., 2019). Let us derive the neural tangent kernel of this architecture. This is a network with output of dimension $m$, so for each $i, j \in [m]$ we will derive $K_{ij,O}(\boldsymbol{X}, \boldsymbol{X}'), K_{ij,V}(\boldsymbol{X}, \boldsymbol{X}'), K_{ij,P}(\boldsymbol{X}, \boldsymbol{X}'), K_{ij,E}(\boldsymbol{X}, \boldsymbol{X}')$ which give the dynamics at small times for training the $\{\boldsymbol{W}_{O,h}\}_{h \in [H]}$, the $\{\boldsymbol{W}_{V,h}\}_{h \in [H]}$, the $\boldsymbol{W}_P$, and the $\boldsymbol{W}_E$ weights at small times, respectively. Writing $\boldsymbol{W}_E = [\boldsymbol{w}_{E,1}, \dots, \boldsymbol{w}_{E,m}]^\top$, by the law of large numbers,

$$K_{ij,O}(\boldsymbol{X}, \boldsymbol{X}') = \sum_{h \in [H]} \left( \frac{\partial [f_{\mathsf{attn}}(\boldsymbol{X}; \boldsymbol{\theta})]_i}{\partial \boldsymbol{W}_{O,h}} \right)^T \left( \frac{\partial [f_{\mathsf{attn}}(\boldsymbol{X}'; \boldsymbol{\theta})]_j}{\partial \boldsymbol{W}_{O,h}} \right)$$

$$\propto \frac{1}{H} \sum_{h \in [H]} (\boldsymbol{X} \boldsymbol{W}_E + \gamma \boldsymbol{P}) \boldsymbol{W}_{V,h}^T \boldsymbol{W}_{V,h} (\boldsymbol{W}_E^T \boldsymbol{X}^T + \gamma \boldsymbol{P}^T) \boldsymbol{w}_{E,i}^T \boldsymbol{w}_{E,j}$$

$$\overset{d_{head} \to \infty, d_{emb} \to \infty}{\longrightarrow} \delta_{ij}(\delta_{x_1, x_1'} + \gamma^2)$$

$$K_{ij,V}(\boldsymbol{X}, \boldsymbol{X}') = \sum_{h \in [H]} \left( \frac{\partial [f_{\mathsf{attn}}(\boldsymbol{X}; \boldsymbol{\theta})]_i}{\partial \boldsymbol{W}_{V,h}} \right)^T \left( \frac{\partial [f_{\mathsf{attn}}(\boldsymbol{X}'; \boldsymbol{\theta})]_j}{\partial \boldsymbol{W}_{V,h}} \right)$$

$$\propto \frac{d_{emb}}{d_{head}} \sum_{h \in [H]} \boldsymbol{w}_{E,i}^T \boldsymbol{W}_{O,h}^T \boldsymbol{W}_{O,h} \boldsymbol{w}_{E,j} (\boldsymbol{X}\boldsymbol{W}_E + \gamma \boldsymbol{P})^T (\boldsymbol{X}'\boldsymbol{W}_E + \gamma \boldsymbol{P})$$

$$\overset{d_{head} \to \infty}{\to} \boldsymbol{w}_{E,i}^T \boldsymbol{w}_{E,j} (\boldsymbol{X}\boldsymbol{W}_E + \gamma \boldsymbol{P})^T (\boldsymbol{X}'\boldsymbol{W}_E + \gamma \boldsymbol{P})$$

$$\overset{d_{emb} \to \infty}{\to} \delta_{ij} (\delta_{x_1, x_1'} + \gamma^2)$$

$$K_{ij,P}(\boldsymbol{X}, \boldsymbol{X}') = \left( \frac{\partial [f_{\mathsf{attn}}(\boldsymbol{X}; \boldsymbol{\theta})]_i}{\partial \boldsymbol{P}} \right)^T \left( \frac{\partial [f_{\mathsf{attn}}(\boldsymbol{X}'; \boldsymbol{\theta})]_j}{\partial \boldsymbol{P}} \right) = \gamma^2 \boldsymbol{w}_{E,i}^\top \boldsymbol{w}_{E,j} \overset{d_{emb} \to \infty}{\to} \gamma^2 \delta_{ij}$$

$$K_{ij,E}(\boldsymbol{X}, \boldsymbol{X}') = \left( \frac{\partial [f_{\mathsf{attn}}(\boldsymbol{X}; \boldsymbol{\theta})]_i}{\partial \boldsymbol{W}_E} \right)^T \left( \frac{\partial [f_{\mathsf{attn}}(\boldsymbol{X}'; \boldsymbol{\theta})]_j}{\partial \boldsymbol{W}_E} \right)$$

$$= \delta_{ij} (\boldsymbol{X}\boldsymbol{W}_E + \gamma \boldsymbol{P}) (\sum_{h \in [H]} \boldsymbol{W}_{V,h}^T \boldsymbol{W}_{O,h}) (\sum_{h \in [H]} \boldsymbol{W}_{O,h}^T \boldsymbol{W}_{V,h}) (\boldsymbol{W}_E^T (\boldsymbol{X}')^T + \gamma \boldsymbol{P}^T)$$

$$+ \delta_{x_1, x_1'} \boldsymbol{w}_{E,i}^T (\sum_{h \in [H]} \boldsymbol{W}_{O,h}^T \boldsymbol{W}_{V,h}) (\sum_{h \in [H]} \boldsymbol{W}_{V,h}^T \boldsymbol{W}_{O,h}) \boldsymbol{w}_{E,j}^T$$

$$+ \delta_{i, x_1'} \boldsymbol{w}_{E,j}^T (\sum_{h \in [H]} \boldsymbol{W}_{O,h}^T \boldsymbol{W}_{V,h}) (\sum_{h \in [H]} \boldsymbol{W}_{O,h}^T \boldsymbol{W}_{V,h}) (\boldsymbol{w}_{E,x_1} + \gamma \boldsymbol{P}^T)$$

$$+ \delta_{x_1, j} \boldsymbol{w}_{E,i}^T (\sum_{h \in [H]} \boldsymbol{W}_{O,h}^T \boldsymbol{W}_{V,h}) (\sum_{h \in [H]} \boldsymbol{W}_{O,h}^T \boldsymbol{W}_{V,h}) (\boldsymbol{w}_{E,x_1'} + \gamma \boldsymbol{P}^T)$$

$$\overset{d_{head} \to \infty, d_{emb} \to \infty, H \to \infty}{\to} \delta_{ij} (2\delta_{x_1, x_1'} + \gamma^2),$$

since only the first two terms do not vanish as the embedding dimension and number of heads go to infinity.

**Training loss and testing loss**  Let $(x_1, y_1), \ldots, (x_n, y_n) \in \mathcal{X} \times \mathcal{X}$ be a training set of data points drawn from this task, where due to the structure of the template task each of the context strings is length-1 and we have $x_i = y_i$. We will test the model on a data point $(x^{test}, y^{test})$, which does not appear in the test set: i.e., $x^{test} = y^{test} \notin \{x_1, \ldots, x_n\}$.

The training loss is given by

$$\mathcal{L}_{train}(\boldsymbol{\theta}) = \frac{1}{n} \sum_{i=1}^n \ell(f_{\mathsf{attn}}(x_i; \boldsymbol{\theta}), y_i),$$

where $\ell$ is the cross-entropy loss, and the test loss is given by

$$\mathcal{L}_{test}(\boldsymbol{\theta}) = \ell(f_{\mathsf{attn}}(x^{test}), y^{test}).$$

**Theorem J.2.** *For any learning rates $\eta_O, \eta_V, \eta_P, \eta_E$ such that $|\frac{\partial \mathcal{L}_{train}}{\partial t}| = O(1)$ as $d_{emb}, d_{head}$, and $H \to \infty$, we have $|\frac{\partial \mathcal{L}_{test}}{\partial t}| \le o(1)$. In other words, the error for generalization on unseen symbols does not decrease during training for infinite-width transformers.*

*Proof.* Consider training with gradient flow with learning rates $\eta_O, \eta_V, \eta_P, \eta_E$ on the parameters $\{\boldsymbol{W}_{O,h}\}_{h \in [H]}, \{\boldsymbol{W}_{V,h}\}_{h \in [H]}, \boldsymbol{W}_P$, and $\boldsymbol{W}_E$, respectively. In the limit as $d_{emb} \to \infty$ we have $f_{\mathsf{attn}}(\boldsymbol{X}; \boldsymbol{\theta}_0) \to 0$, so

$$\frac{\partial \mathcal{L}_{train}}{\partial \boldsymbol{\theta}} \Big|_{\boldsymbol{\theta}=\boldsymbol{\theta}_0} = \frac{1}{n} \sum_{i=1}^n (\frac{1}{m} \mathbf{1} - \boldsymbol{e}_{x_i})^T \frac{\partial f_{\mathsf{attn}}(\boldsymbol{X}_i; \boldsymbol{\theta})}{\partial \boldsymbol{\theta}} \Big|_{\boldsymbol{\theta}=\boldsymbol{\theta}_0} .$$

So at time $t = 0$, the training loss decreases as

$$\frac{\partial \mathcal{L}_{train}}{\partial t}\mid_{t=0} \to -\frac{1}{n^2} \sum_{i,i'\in[n]} \sum_{j,j'\in[m]} (1/m - \delta_{j,x_i})(1/m - \delta_{j',x_{i'}})$$
$$\cdot (\eta_V K_{jj',V}(\boldsymbol{X}_i, \boldsymbol{X}_{i'}) + \eta_O K_{jj',O}(\boldsymbol{X}_i, \boldsymbol{X}_{i'})$$
$$+ \eta_P K_{jj',P}(\boldsymbol{X}_i, \boldsymbol{X}_{i'}) + \eta_E K_{jj',E}(\boldsymbol{X}_i, \boldsymbol{X}_{i'})).$$

So we must take $\eta_O = O(1/H), \eta_V = O(d_{emb}/d_{head}), \eta_P = O(1)$, and $\eta_E = O(1)$ for us to have $\frac{\partial \mathcal{L}_{train}}{\partial t} = O(1)$ be bounded by a constant that does not grow with $d_{emb}, d_{head}$, and $H$.

Under these choices of learning rates, the test loss on token $x^{test}$ which is not in the training dataset $\{x_1, \ldots, x_n\}$, evolves as

$$\frac{\partial \mathcal{L}_{test}}{\partial t}\mid_{t=0} \to -\frac{1}{n} \sum_{i\in[n]} \sum_{j,j'\in[m]} (1/m - \delta_{j,x_i})(1/m - \delta_{j',x^{test}})$$
$$\cdot (\eta_V K_{jj',V}(\boldsymbol{X}_i, \boldsymbol{X}^{test}) + \eta_O K_{jj',O}(\boldsymbol{X}_i, \boldsymbol{X}^{test})$$
$$+ \eta_P K_{jj',P}(\boldsymbol{X}_i, \boldsymbol{X}^{test}) + \eta_E K_{jj',E}(\boldsymbol{X}_i, \boldsymbol{X}^{test}))$$
$$\to -\frac{1}{n} \sum_{i\in[n]} \sum_{j,j'\in[m]} (1/m - \delta_{j,x_i})(1/m - \delta_{j',x^{test}})$$
$$\cdot ((\frac{d_{head}}{d_{emb}}\eta_V + H\eta_O)\delta_{j,j'}(\delta_{x_i,x^{test}} + \gamma^2)$$
$$+ \eta_P \gamma^2 \delta_{j,j'} + 2H\eta_E \delta_{j,j'}(\delta_{x_i,x^{test}} + \gamma^2))$$
$$= -\frac{\gamma^2}{n} \sum_{i\in[n]} \sum_{j\in[m]} (1/m - \delta_{j,x_i})(1/m - \delta_{j,x^{test}}) \cdot (\frac{d_{head}}{d_{emb}}\eta_V + H\eta_O + \eta_P + 2\eta_E)$$
$$= -\frac{C}{n} \sum_{i\in[n]} \sum_{j\in[m]} (1/m - \delta_{j,x_i})(1/m - \delta_{j,x^{test}})$$
$$= -C/m + C/m + C/m = C/m \geq 0.$$

$\square$

On the other hand, now we consider the $f_{\text{attn}}$ architecture where in each head we replace $\boldsymbol{W}_{V,h}^T \boldsymbol{W}_{O,h}$ with $\boldsymbol{W}_{V,h}^T \boldsymbol{W}_{O,h} + b_h \boldsymbol{I}$, where $b_h$ is a trainable parameter and $\boldsymbol{I} \in \mathbb{R}^{d_{emb} \times d_{emb}}$ is the identity matrix:

$$f'_{\text{attn}}(\boldsymbol{X}; \boldsymbol{\theta}) = \boldsymbol{W}_E \boldsymbol{z}_1 \in \mathbb{R}^m \qquad \text{(Unembedding layer)}$$

where

$$\boldsymbol{z}'_1 = \sum_{h\in[H]} (\boldsymbol{A}'_h)^T \boldsymbol{e}_k$$
$$\boldsymbol{A}'_h = \text{smax}(\beta \boldsymbol{Z}_0 \boldsymbol{W}_{K,h}^T \boldsymbol{W}_{Q,h} \boldsymbol{Z}_0^T) \boldsymbol{Z}_0 (\boldsymbol{W}_{V,h}^T \boldsymbol{W}_{O,h} + b_h \boldsymbol{I}) \in \mathbb{R}^{k \times d_{emb}} \qquad \text{(Attention heads)}$$
$$\boldsymbol{Z}_0 = \boldsymbol{X}\boldsymbol{W}_E + \gamma \boldsymbol{P} \in \mathbb{R}^{k \times d_{emb}}. \qquad \text{(Embedding layer)}$$

Again, for the case of $k = 1$ that we consider, the network simplifies considerably to

$$f'_{\text{attn}}(\boldsymbol{X}; \boldsymbol{\theta}) = \boldsymbol{W}_E (\sum_{h\in[H]} \boldsymbol{W}_{O,h}^T \boldsymbol{W}_{V,h} + b_h \boldsymbol{I})(\boldsymbol{W}_E^T \boldsymbol{X}^T + \gamma \boldsymbol{P}^T). \qquad (27)$$

We initialize $b_h = 0$ for all $h$, so that the neural tangent kernels $K_{ij,O}, K_{ij,V}, K_{ij,P}, K_{ij,E}$ are the same as above. Now we also have a neural tangent kernel for training the parameters $\{b_h\}_{h\in[H]}$:

$$K_{ij,b}(\boldsymbol{X}, \boldsymbol{X}') = \sum_{h\in[H]} \frac{\partial [f_{\text{attn}}(\boldsymbol{X}; \boldsymbol{\theta})]_i}{\partial b_h} \frac{\partial [f_{\text{attn}}(\boldsymbol{X}'; \boldsymbol{\theta})]_j}{\partial b_h}$$
$$\propto \boldsymbol{w}_{E,i}^\top (\boldsymbol{W}_E^T \boldsymbol{X}^T + \gamma \boldsymbol{P}^T)(\boldsymbol{X}\boldsymbol{W}_E + \gamma \boldsymbol{P}^T)\boldsymbol{w}_{E,j}$$
$$\overset{d_{emb} \to \infty}{\to} \delta_{i,x_1} \delta_{j,x'_1}$$

We prove that under this parametrization the test loss does decrease with training, which shows that adding this trainable identity scaling allows transformers to succeed at this task.

**Theorem J.3.** *There is a choice of learning rates $\eta_b, \eta_V, \eta_O, \eta_E, \eta_P$ such that as $d_{emb}, d_{head}, H \to \infty$ we have $|\frac{\partial \mathcal{L}_{train}}{\partial t}| \mid_{t=0} = O(1)$ and $-\frac{\partial \mathcal{L}_{test}}{\partial t} \mid_{t=0} = \Omega(1)$.*

*Proof.* Training just the parameters $\{b_h\}_{h\in[H]}$ with learning rate $\eta_b$ (keeping the learning rates $\eta_V, \eta_O, \eta_P, \eta_E = 0$, so the training loss decreases as

$$\frac{\partial \mathcal{L}_{train}}{\partial t} \mid_{t=0} \to -\frac{\eta_b}{n^2} \sum_{i,i'\in[n]} \sum_{j,j'\in[m]} (1/m - \delta_{j,x_i})(1/m - \delta_{j',x_{i'}}) K_{jj',b}(\boldsymbol{X}_i, \boldsymbol{X}_{i'}),$$

so we should take $\eta_b = \Theta(1/H)$ for the train loss have derivative on the order of $\Theta(1)$. The test loss decreases as:

$$\begin{aligned}
\frac{\partial \mathcal{L}_{test}}{\partial t} \mid_{t=0} &\to -\frac{\eta_b}{n} \sum_{i\in[n]} \sum_{j,j'\in[m]} (1/m - \delta_{j,x_i})(1/m - \delta_{j',x^{test}}) K_{jj',b}(\boldsymbol{X}_i, \boldsymbol{X}^{test}) \\
&\to -\frac{H\eta_b}{n} \sum_{i\in[n]} \sum_{j,j'\in[m]} (1/m - \delta_{j,x_i})(1/m - \delta_{j',x^{test}}) \delta_{j,x_i} \delta_{j',x^{test}} \\
&= -\frac{H\eta_b}{n} \sum_{i\in[n]} (1/m - 1)(1/m - 1) \\
&= -H\eta_b(1 - 1/m)^2 \\
&= \Omega(1),
\end{aligned}$$

for $\eta_b = \Omega(H)$, as $d_{emb}, H \to \infty$. $\square$