# OpenReview forum: "When can transformers reason with abstract symbols?"
_ICLR.cc/2024/Conference — ICLR 2024 poster_

### Official Review · Reviewer_Fsbi · 2023-10-24

**Soundness:** 4 excellent
**Presentation:** 3 good
**Contribution:** 4 excellent
**Rating:** 8
**Confidence:** 3

**Summary:**

This work presents an analysis of the abstract reasoning capability of the transformer architecture. The analysis yields a novel modification that improves out-of-distribution reasoning in symbolic tasks. The proposed modification also yields improved performance in an LLM (GPT-2) on a more realistic task.

**Strengths:**

- The paper presents a formal analysis of OOD symbolic reasoning in both transformers and MLPs, finding that transformers, but not MLPs, are theoretically capable of a form of abstract symbolic reasoning, given sufficient training data diversity. This provides interesting insights into the source of reasoning capabilities in LLMs, and contrasts with the common view that their capabilities are due exclusively to scale. This sort of analysis is less common in reasoning papers, so it is a very welcome contribution.
- The analysis yields a novel modification to transformers that further improves their abstract reasoning capabilities, improving sample efficiency on one class of tasks, and enabling generalization on another.
- The proposed modification also yields improved performance in a pretrained LLM (GPT-2) on a realistic task (wikitext).

**Weaknesses:**

I think this paper provides a strong contribution. My comments are primarily about clarification and presentation:
- The abstract and introduction refer to 'transformer large language models'. However, if I understand correctly, the experiments in Figures 1 and 2 are performed on transformers trained from scratch, not on LLMs (though an LLM is tested later on). If this is correct, I think it would be clearer to simply refer to transformers (rather than LLMs) in the abstract and introduction.
- It would be helpful if an intuitive characterization of the proposed modifications ($aI$ and $bI$) could be provided earlier in the paper, along with some sense of why this modification helps.
- Do the wikitext experiments involve OOD generalization in any sense (in the same way that the reasoning tasks do)? If not, I wonder whether this partly explains the smaller relative improvement from the modified models vs. GPT-2.
- I was somewhat confused by the distinction between GPT-2 and 'GPT-2 pretrained' -- isn't GPT-2 by default pretrained (since this term refers to the trained model)? Just to clarify, do the results in Figure 4 reflect pretrained GPT-2, which is then fine-tuned on the wikitext datasets, or both of the models in this figure are trained from scratch only on wikitext?

Minor comment:
- The legend is cut off in figure 3a. In general the figure text is very small and difficult to read.

**Questions:**

I found it interesting that GPT-2 already contains heads that implement a strategy similar to the modifications proposed in this paper. A preliminary investigation of GPT-4 suggests that it has a very strong ability to perform the abstract reasoning tasks from this paper based on just a few in-context examples. Do the authors expect that larger models (e.g. GPT-3 and GPT-4) may be able to perform these tasks without the proposed modifications, and if so how does that relate to the theoretical results? Do the authors expect that this capability would similarly depend on the emergence of heads that implement a diagonal attention strategy? Does the analysis in this paper have anything to say about the ability to learn these kinds of abstract reasoning tasks through *in-context* learning (as opposed to direct training of the model parameters)?

I am also curious how the proposal in this work might relate to the idea of 'induction heads' [1]. Intuitively these seem to be related, in that both exploit the inductive biases of transformers to perform abstract tasks involving copying.

[1] Olsson, C., Elhage, N., Nanda, N., Joseph, N., DasSarma, N., Henighan, T., ... & Olah, C. (2022). In-context learning and induction heads. arXiv preprint arXiv:2209.11895.

---

> ### Author Response · Authors · 2023-11-19
> **Response**
>
> ## Thank you for your review
>
> Thank you for your careful reading of the paper, your helpful suggestions on the presentation, and your interesting questions. Please see below for item-by-item responses.
>
> ## Weaknesses
>
> > The abstract and introduction refer to 'transformer large language models'. However, if I understand correctly, the experiments in Figures 1 and 2 are performed on transformers trained from scratch, not on LLMs (though an LLM is tested later on). If this is correct, I think it would be clearer to simply refer to transformers (rather than LLMs) in the abstract and introduction.
>
> Thank you for the suggestion. We will edit the intro and in particular our presentation of Figures 1 and 2 to make it clearer that those experiments are on language transformers and not pretrained language models.
>
> > It would be helpful if an intuitive characterization of the proposed modifications ($aI$ and  $bI$) could be provided earlier in the paper, along with some sense of why this modification helps.
>
> Thank you for the suggestion. We will try to incorporate this into Section 1.1, and will expand on the intuition of the modifications in Section 4.2 and Section 5.
>
> > Do the wikitext experiments involve OOD generalization in any sense (in the same way that the reasoning tasks do)? If not, I wonder whether this partly explains the smaller relative improvement from the modified models vs. GPT-2.
>
> Good point. The Wikitext task is next-token prediction on Wikipedia, so it does not explicitly involve OOD generalization. There may be some implicit reasoning tasks in this problem, which could explain the improvement.
>
> > I was somewhat confused by the distinction between GPT-2 and 'GPT-2 pretrained' -- isn't GPT-2 by default pretrained (since this term refers to the trained model)? Just to clarify, do the results in Figure 4 reflect pretrained GPT-2, which is then fine-tuned on the wikitext datasets, or both of the models in this figure are trained from scratch only on wikitext?
>
> These are models trained from scratch only on wikitext. We will clarify in the text.
>
> > The legend is cut off in figure 3a. In general the figure text is very small and difficult to read.
>
> Our apologies. This got accidentally introduced when we regenerated the figures right before submitting.
>
> ## Questions
> > I found it interesting that GPT-2 already contains heads that implement a strategy similar to the modifications proposed in this paper. A preliminary investigation of GPT-4 suggests that it has a very strong ability to perform the abstract reasoning tasks from this paper based on just a few in-context examples. Do the authors expect that larger models (e.g. GPT-3 and GPT-4) may be able to perform these tasks without the proposed modifications, and if so how does that relate to the theoretical results? Do the authors expect that this capability would similarly depend on the emergence of heads that implement a diagonal attention strategy? Does the analysis in this paper have anything to say about the ability to learn these kinds of abstract reasoning tasks through in-context learning (as opposed to direct training of the model parameters)?
>
> * This is an excellent question. Unfortunately, our theory does not bear upon in-context learning. It seems that to understand that it will be necessary to understand feature-learning for these template tasks – our paper provides a baseline to build off of when conducting those future analyses.
>
> * We expect that the feature learning in the form of strong diagonal weights in the attention heads plays an important role in practice for both copying and for checking whether two symbols are equal. We do not know whether this learned weight structure is necessary for in-context learning to succeed, although our intuition is that it is. This is indeed a fantastic direction for future work.
>
> > I am also curious how the proposal in this work might relate to the idea of 'induction heads' [1]. Intuitively these seem to be related, in that both exploit the inductive biases of transformers to perform abstract tasks involving copying.
>
> There is an interesting relation between our result and that work. Our result on copying, and our experiment in Figure 3a, indicate that the larger the embedding dimension, the more difficult it becomes for the model to learn to copy symbols it has not seen copied at training time. Thus, we should expect infinitely-large models to be unable to learn induction heads that generalize outside of the training data. So it would seem that transformer modifications are necessary for infinitely-large models if you want data-efficiency in learning to copy.

---

> > ### Comment · Reviewer_Fsbi · 2023-11-22
> > **Response to rebuttal**
> >
> > Thank you very much to the authors for the thorough responses. I enthusiastically support the paper's acceptance.

---

### Official Review · Reviewer_xzui · 2023-11-01

**Soundness:** 2 fair
**Presentation:** 2 fair
**Contribution:** 2 fair
**Rating:** 6
**Confidence:** 1

**Summary:**

The paper investigates the "variable-binding" capability of transformers from a theoretic perspective, which refers to the generalization ability to handle unseen symbols during training. To formulate the variable-binding ability, the authors propose a framework called template tasks. Then the authors prove that a simplified transformer block can generalize to unseen input symbols when the output is a binary label, but cannot generalize when the output label is also dependent on novel input symbols. Finally, the authors propose a simple mechanism to augment the attention layer in transformers which is empirically shown to be effective.

**Strengths:**

- The author provides a theoretical formulation of variable-binding tasks called template tasks.
- Proofs are provided for the capability and limitations of a simplified transformer architecture on two instantiations of template tasks.
- Extensive empirical experiments are conducted on the template task framework.

**Weaknesses:**

My main concern lies in the assumption of the proof. Throughout the whole paper, the authors assume a simplified depth-1 transformer architecture without residual connections and layer normalization. However, residual connections and layer normalization are known to be crucial for the performance of transformers [1][2]. The simplification limits the applicability of the main theoretical conclusions to more practical settings. On the other hand, all empirical results use a vanilla depth-2 transformer without modification. This makes it hard to draw direct parallels to justify the practical implications of the theoretical results.

[1] On Layer Normalization in the Transformer Architecture. ICML 2020
[2] Transformer Feed-Forward Layers Are Key-Value Memories. Mor Geva, et al. EMNLP 2021

**Questions:**

As I mentioned in the weakness section, I am especially interested in authors' opinions about how to extend the main proof in the paper to more practical transformer architecture with residual connections and layer normalization.

---

> ### Author Response · Authors · 2023-11-18
> **Response**
>
> ## Thank you for your review
> Thank you for your positive comments and your helpful question. We address this question below.
> ## Weaknesses
> > My main concern lies in the assumption of the proof. Throughout the whole paper, the authors assume a simplified depth-1 transformer architecture without residual connections and layer normalization. However, residual connections and layer normalization are known to be crucial for the performance of transformers [1][2]. The simplification limits the applicability of the main theoretical conclusions to more practical settings. On the other hand, all empirical results use a vanilla depth-2 transformer without modification. This makes it hard to draw direct parallels to justify the practical implications of the theoretical results.
>
> * As far as theoretical results, the self-attention+MLP setting in our work is not unusual and even goes beyond many theoretical works. All known theoretical analyses of transformer training dynamics use a combination of simplifications such as: (a) removing the softmax to get a linear attention mechanism, (b) replacing the $W_KW_Q^T$ matrices with just one matrix $W_{KQ}$ and similarly for $W_VW_O^T$, (c) considering only one layer, (d) assuming diagonal weight matrices, (e) removing layernorms and residual connections, and (f) even removing the MLP layers.
> * Furthermore, we believe that the same ideas in our analysis could be used to prove our theoretical results in the multilayer setting with residual connections – but this would require an even more involved analysis than what we currently have here, and we did not write this because we feel that the more convoluted technical arguments would distract from the main ideas of the proof which only need one self-attention layer + one MLP layer.
>
> * In our experiments, we do train standard multilayer transformers with residual connections and layer norm. These include practical models such as GPT-2 where we show a benefit of our theory-inspired modifications.
>
> * We will be clearer about our experimental details in the revision, explaining more clearly how the theoretical and experimental setups differ. In the revision, we will also  add extra experiments on deeper transformers (we do have some experiments for deeper transformers with up to 16 layers already in Figures 2,6,7). The 2-layer choice for most of our experiments was due to the larger computational cost of training deeper transformers.
>
> * Finally, we believe that our theory is relevant to practice since it suggests modifications to the transformer that improve data-efficiency on the reasoning tasks in practice by an order of magnitude.

---

> > ### Comment · Reviewer_xzui · 2023-11-23
> >
> > Thanks authors for addressing my concerns. I will raise my score to accept.

---

### Official Review · Reviewer_jCDP · 2023-11-02

**Soundness:** 4 excellent
**Presentation:** 4 excellent
**Contribution:** 3 good
**Rating:** 8
**Confidence:** 3

**Summary:**

This manuscript studies the ability of transformers to reason with abstract binding --- or symbol binding. The authors contrast MLPs and transformers and argue how MLPs trained with gradient descent are incapable of abstract reasoning. They then show how transformers on the other hand are capable of abstract reasoning given lots of data. Finally, they propose an a transformer variant that generalizes faster to the symbol binding task and verify the same empirically.

**Strengths:**

The paper was a great read! The authors conduct a thorough theoretical analysis with practical insights suggesting changes to the transformer architecture. The work studies if transformers are capable of symbol binding, which is very important question and of broad interest to the community. The paper presents a diverse set of results, but makes them easy to understand to a broader audience too.

The template tasks are elegant and easy to understand. It grounds the problem in a concrete scenario that can be studied. The authors present concrete insights for real labels and symbolic labels and identify different sample complexity behaviors for both scenarios. The work identifies important factors like data diversity or transformer embedding size that change the convergence behavior.

The authors use their theory to develop practical insights for developing transformer architectures. In particular, they show experiments on a synthetic dataset and on Wikitext to validate their proposed modification.

**Weaknesses:**

The theory for the transformers seems to rely on training only the final layer of the transformer. Does that mean that there are other architectures that can also do symbol binding? It is unclear which aspect of transformers are important for symbol binding: is the attention, the MLP, the invariance properties or something else? It isn't entirely clear to me how

A limitation of most theory that uses the PAC framework often has vacuous bounds. While the paper derives asymptotic trends, it is unclear if they will predictive of what happens in practice as a result. However, I also acknowledge that this is the case for lot of theorems about deep networks. It is unclear if this probabilistic framework is the right tool for analysis.

**Questions:**

1. The proof for the failure of MLPs relies on taking an expectation over $\theta_t$ obtained through gradient descent. Does this also hold for a single run of an MLP training. Is it possible for MLPs to converge some of the time (and not with probability at least $1 - \delta$ )?
2. The proof seems to rely on the kernel ridge regression estimator. Does this mean that representation learning is not critical for symbol binding?
3. If the data diversity is small (data follows a Zipf distribution), then do these results hold? In practice we expect datasets to follows such distributions and I am unsure if I am misunderstanding the theorem incorrectly.
4. Are there technical reasons for using the kernel regression with gradient flow in section 4, but the cross-entropy loss with next-token prediction for section 5? Is it possible to conduct an analysis similar to section 4 if we consider next token prediction?

---

> ### Author Response · Authors · 2023-11-18
> **Response (1/2)**
>
> ## Thank you for your review
> We are very glad that you enjoyed the paper, and we thank you for your thoughtful comments and insightful questions. We address these below.
>
> ## Weaknesses
> > The theory for the transformers seems to rely on training only the final layer of the transformer. Does that mean that there are other architectures that can also do symbol binding? It is unclear which aspect of transformers are important for symbol binding: is the attention, the MLP, the invariance properties or something else? It isn't entirely clear to me how
>
> * Our analysis identifies the condition number of the N matrix as a key driver of the sample complexity for learning the template task. This condition number is always infinite for MLPs, and so they do not perform symbol binding. (In fact, we prove that MLPs do not perform symbol binding even when all parameters are trained.)
>
> * For non-transformer architectures, the condition number of the N matrix might be finite, in which case those architectures would also perform symbol binding. However, we only prove this for transformers in this paper. We will improve the clarity of the proof sketch in Section 4.2 of Lemmas 4.6 and 4.7 in the revision to better explain why the condition number is finite. The main reason is due to the self-attention layer, which, roughly speaking, allows the architecture to access the matrix XX^T. This matrix contains information about which symbols in the string are repeated and where. We will also improve the explanation in Section 4.3 for how this insight motivates the $W_KW_Q^T + aI$ modification. Here it may be illustrative to give concrete bounds on the condition number before/after the modification for the same-different task.
>
> * Re: training only the last layer, please also see our answer to the question 2 below.
>
> > A limitation of most theory that uses the PAC framework often has vacuous bounds. While the paper derives asymptotic trends, it is unclear if they will predictive of what happens in practice as a result. However, I also acknowledge that this is the case for lot of theorems about deep networks. It is unclear if this probabilistic framework is the right tool for analysis.
>
> * Yes, our sample complexity bounds are indeed non-quantitative in terms of the number and structure of the templates. The bounds could in principle be numerically computed by estimating the network’s kernel matrix.
>
> * Nevertheless, we do get a qualitative result on the capabilities of transformers on symbol binding tasks. And we believe that our theory is relevant to practice since it suggests modifications to the transformer that improve data-efficiency on the reasoning tasks in practice by an order of magnitude.

---

> > ### Author Response · Authors · 2023-11-18
> > **Response (2/2)**
> >
> > ## Questions
> > > 1. The proof for the failure of MLPs relies on taking an expectation over $\\theta_t$ obtained through gradient descent. Does this also hold for a single run of an MLP training. Is it possible for MLPs to converge some of the time (and not with probability at least $1−\\delta$)?
> >
> > The proof of failure of MLPs relies on a coupling between the weights of the MLP trained with the true data distribution D, and the weights of the MLP trained with a fake, permuted data distribution D’. This coupling holds almost surely.
> > Using this coupling, we show that the loss of the MLP after training is lower-bounded in expectation over the randomness of the initialization, the random shuffling of the samples, and the noise in the samples. So it is indeed possible that the MLP may in exceptional cases have good test loss due to the randomness of the algorithm (this will just not be true in expectation). It would be interesting to extend the result to bound the probability of these events.
> >
> > > 2. The proof seems to rely on the kernel ridge regression estimator. Does this mean that representation learning is not critical for symbol binding?
> >
> > * It was quite unexpected to us that the transformer kernel analysis is sufficient to characterize when the relational reasoning tasks can be learned. We began this work with the mindset that we would have to study feature-learning dynamics. Nevertheless, with the kernel analysis, we were able to *characterize* when relational reasoning can be learned by large transformers (it can be learned anytime there is no copying of placeholder symbols; it cannot be learned when there is).
> >
> > * A consequence of our analysis is that any separation between NTK and feature learning would only be quantitative in terms of the number of samples. I.e., to show such a separation we would need to prove bounds on the number of samples with explicit quantitative dependence on the structure and number of the templates. We do not presently know how to prove these, but we feel that it is very interesting for future work. This is a very intriguing question and we will add discussion in the final section.
> >
> > * Also, in the revision, we will add experiments comparing feature learning to NTK training – these indicate that feature learning is intriguingly more data-efficient on the relational reasoning tasks (although NTK also learns given enough data).
> >
> >
> > > 3. If the data diversity is small (data follows a Zipf distribution), then do these results hold? In practice we expect datasets to follows such distributions and I am unsure if I am misunderstanding the theorem incorrectly.
> >
> > * This is a great question. If the data follows a Zipf distribution with word k occurring with probability $p_k \propto 1/k$, then it will occur a total number of times $p_k \\approx ck / \\log K$, where $K$ is the total number of words in the language. As the total number of words increases, then this probability vanishes and so the condition on our data holds with diversity parameter $\rho \propto log(K)$.
> >
> > * However, $log(K)$ is a slowly-growing and so we could hope to do better. Perhaps we should not think of common words as those words that substitute placeholder wildcard tokens. Instead, if a word is sufficiently common then maybe it should be modeled as a regular token in a template. For example, by a large margin the most common words in English are words like “the”, “to”, “be”, “and”, and “of”. These words have specific meanings and we should not generally think of them as abstract, interchangeable symbols, in the same way that we think of names like “Alice”, “Bob”, etc… as interchangeable. Interestingly, large language models appear to sometimes fail when these more common words are used as part of abstract reasoning: see [1]. So maybe our analysis is tight in some sense?
> > * There definitely seems more to learn by exploring in this direction. It seems that to get a tighter understanding we may need to derive new sample complexity bounds that depend quantitatively on the number (and potentially structure) of the templates.
> >
> > [1] https://arxiv.org/abs/2305.15507, “The Larger They Are, the Harder They Fail: Language Models do not Recognize Identifier Swaps in Python” by Miceli-Barone et al.
> >
> > > 4. Are there technical reasons for using the kernel regression with gradient flow in section 4, but the cross-entropy loss with next-token prediction for section 5? Is it possible to conduct an analysis similar to section 4 if we consider next token prediction?
> >
> >
> > The technical reason is that we do not have a closed-form solution to the kernel regression with cross-entropy loss. It would be very interesting to find ways to prove the result in this cross-entropy setting without such a closed-form solution. We will mention this in the paper.

---

### Official Review · Reviewer_gjp1 · 2023-11-04

**Soundness:** 3 good
**Presentation:** 3 good
**Contribution:** 3 good
**Rating:** 8
**Confidence:** 4

**Summary:**

In this work, the authors aim to establish and understand the abilities of self-attention architecture to solve variable binding tasks. I find the targeted problem both valuable and exciting---indeed, variable binding is a core intelligence capability and yet we have not seen sufficient work in the community targeting it as a direct benchmark for understanding limitations of existing models. I think this work is a step in the right direction therefore.

**Strengths:**

As noted in the summary, I find the targeted problem valuable and exciting. The theoretical framework itself involves several assumptions that are arguably impractical (making this essentially a kernel regression take on LLMs paper), but nonetheless the tasks established earlier in the paper are very well described and the results are accordingly insightful.

**Weaknesses:**

1. I think the authors are over-claiming their results at several points. The simplified architecture that is studied in this work, i.e., a self-attention + MLP model, is not justified to be labeled a Transformer. Residual connections play a huge role in both optimization stability and, e.g., some of the seemingly emergent abilities of Transformers. In this sense, I would argue the paper is actually focused on understanding inductive biases and limitations of a feedforward, MLP model with self-attention. This is totally fine, but the over-claiming can yield, in my opinion, an incorrect takeaway that Transformers broadly cannot perform the variable binding tasks. It may also be worth mentioning that the results focus on a inference with a single pass through the model, i.e., tools like in-context learning or chain-of-though can impact the results. The latter is just a nitpick to better contextualize the paper's findings.

2. Lack of experimental details: While the authors often state that their theoretical claims focus on a simplified architecture, it is unclear to me if the experiments involve this simplified architecture or an actual Transformer, i.e., with residual connections, is used. Generally, I think the experimental details can be improved.

3. Relation with prior theoretical works on inductive biases of self-attention: A specific work that addresses a similar question as this paper is Edelman et al. [1]. Therein, the authors show a self-attention model can express sparse boolean circuits and performs implicit variable creation during forward inference. I think the work deserves thorough discussion in the current paper's related work. A few more empirical works also exist on this problem, e.g., Davies et al. [2], that will be worth discussing.

4. Discussion with Systematic generalization: I would argue the tasks that the authors studied are extremely related to what NLP community has studied under the label of systematic generalization before. I think such papers warrant discussion as well (e.g., see [3]).



[1] Inductive Biases and Variable Creation in Self-Attention Mechanisms, Edelman et al., 2022.

[2] Discovering Variable Binding Circuitry with Desiderata, Davies et al., 2023

[3] Compositionality Decomposed: How do Neural Networks Generalise?, Hupkes et al., 2020

**Questions:**

The proposed changes based on the theoretical models involve addition of a scaled identity matrix to parts of the attention matrices. This looks, at a surface, like addition of a residual connection in fact. Can the authors clarify this further? I found the discussion of this point to be quite unclear.

---

> ### Author Response · Authors · 2023-11-18
> **Response (1/2)**
>
> ## Thank you for your review
> Thank you for your positive comments and detailed constructive feedback. We are glad that you found this problem exciting and thought our results are a step in the right direction. Your questions and concerns are addressed point by point below.
>
> ## Weaknesses
> > Weaknesses: As noted in the summary, I find the targeted problem valuable and exciting. The theoretical framework itself involves several assumptions that are arguably impractical (making this essentially a kernel regression take on LLMs paper), but nonetheless the tasks established earlier in the paper are very well described and the results are accordingly insightful.
>
> * As far as we are aware, there are **no previous results that prove the ability of transformers to learn to perform symbolic reasoning tasks** when trained. Our paper is the first to formalize a framework of such reasoning tasks and to prove it is learned by transformers when trained.
>
>
> * It was quite **unexpected to us that the transformer kernel analysis is sufficient** to characterize when the relational reasoning tasks can be learned. We began this work with the mindset that we would have to study feature-learning dynamics. Nevertheless, with the kernel analysis, we were able to *characterize* when the relational reasoning can be learned by large transformers (it can be learned anytime there is no copying of placeholder symbols; it cannot be learned when there is).
>
>
> * **The analysis was considerably involved**: (i) This is not a typical kernel method analysis where it is proved that the data can be memorized – instead, we must show that the network *generalizes in the correct way out of distribution*, for any task in our framework. (ii) Also, insofar as we know, this is also the first paper that manages to analyze how the transformer kernel learns a nontrivial task, due to the technical difficulties involved in studying this kernel. The technical tools we develop here may thus be useful for future work.
>
>
>
> * We believe that our theory is relevant since it suggests modifications to the transformer that **improve data-efficiency on the reasoning tasks in practice by an order of magnitude**.
>
>
> * **Our work opens up many directions** that we feel are outside of the scope of the current paper. One direction is to study feature learning. We completely agree that it is an excellent question to ask: can we characterize if/when feature learning is more powerful than NTK? But, to ask this question, an NTK analysis as in this paper is needed first. In the revision, we will add experiments comparing feature learning to NTK training – these indicate that feature learning is intriguingly more data-efficient on the relational reasoning tasks (although NTK also learns given enough data).
>
> * On a more philosophical note, our results raise the question: to what extent is transformer reasoning due to NTK, and to what extent is it due to feature learning such as head specialization? This is a highly relevant question in the context of interpreting the internal workings of transformers. The more NTK-like the solution, the more polysemantic heads that the network will have, and so the harder it will be to interpret a network by extracting sparse subnetworks (which is currently the main method of the mechanistic interpretability literature). It appears quite relevant to the reference [3] that you have pointed out.

---

> ### Author Response · Authors · 2023-11-18
> **Response (2/2)**
>
> ## Questions
> > 1. I think the authors are over-claiming their results at several points. The simplified architecture that is studied in this work, i.e., a self-attention + MLP model, is not justified to be labeled a Transformer. Residual connections play a huge role in both optimization stability and, e.g., some of the seemingly emergent abilities of Transformers. In this sense, I would argue the paper is actually focused on understanding inductive biases and limitations of a feedforward, MLP model with self-attention. This is totally fine, but the over-claiming can yield, in my opinion, an incorrect takeaway that Transformers broadly cannot perform the variable binding tasks. It may also be worth mentioning that the results focus on a inference with a single pass through the model, i.e., tools like in-context learning or chain-of-though can impact the results. The latter is just a nitpick to better contextualize the paper's findings.
>
> * We are sorry that we give the impression of overclaiming. It is not at all our intention to overclaim. Our reasoning is as follows:
>     *  As far as theoretical results, the self-attention+MLP setting in our work is not unusual and even goes beyond many theoretical works. All known theoretical analyses of transformer training dynamics use a combination of simplifications such as: (a) removing the softmax to get a linear attention mechanism, (b) replacing the $W_KW_Q^T$ matrices with just one matrix $W_{KQ}$ and similarly for $W_VW_O^T$, (c) considering only one layer, (d) assuming diagonal weight matrices, (e) removing layernorms and residual connections, and (f) even removing the MLP layers.
>     * Furthermore, we believe that the same ideas in our analysis could be used to prove our theoretical results in the multilayer setting with residual connections – but this would require an even more involved analysis than what we currently have here, and we did not write this because we feel that the more convoluted technical arguments would distract from the main ideas of the proof which only need one self-attention layer + one MLP layer.
>
>     * In our experiments, we do train standard multilayer transformers, including practical models such as GPT-2 where we show a benefit of our theory-inspired modifications.
>
> * With respect to “the takeaway that Transformers broadly cannot perform the variable binding tasks”, this was not our intention. Our intention was only to highlight that a large amount of data is needed for simple variable binding tasks, and that theory-inspired modifications can lower that amount of data substantially. Also, we uncovered an intriguing inverse scaling law for copying which shows that making transformers larger can actually break down some of their reasoning abilities on unseen symbols.
>
> > 2. Lack of experimental details: While the authors often state that their theoretical claims focus on a simplified architecture, it is unclear to me if the experiments involve this simplified architecture or an actual Transformer, i.e., with residual connections, is used. Generally, I think the experimental details can be improved.
> * Thank you for your suggestion. We will make it clearer that our experiments are on standard multilayer transformers with residual connections and layer-norm.
>
> > 3. Relation with prior theoretical works on inductive biases of self-attention: A specific work that addresses a similar question as this paper is Edelman et al. [1]. Therein, the authors show a self-attention model can express sparse boolean circuits and performs implicit variable creation during forward inference. I think the work deserves thorough discussion in the current paper's related work. A few more empirical works also exist on this problem, e.g., Davies et al. [2], that will be worth discussing.
>
> * Thank you for the references. These seem interesting and we will certainly incorporate discussion in the revision.
>
>
> > 4. Discussion with Systematic generalization: I would argue the tasks that the authors studied are extremely related to what NLP community has studied under the label of systematic generalization before. I think such papers warrant discussion as well (e.g., see [3]).
> * Thank you. We were unaware of this reference. Interestingly, our results show that reasoning is possible even without neuron/head specialization, which appears quite relevant to this reference. We will incorporate a discussion in the revision.

---

> ### Author Response · Authors · 2023-11-19
> **Response (3/2)**
>
> Our apologies: we missed answering your question earlier.
>
> > The proposed changes based on the theoretical models involve addition of a scaled identity matrix to parts of the attention matrices. This looks, at a surface, like addition of a residual connection in fact. Can the authors clarify this further? I found the discussion of this point to be quite unclear.
>
> The $W_VW_O^T + bI$ modification is adding an "attention-modulated" skip connection. It is a skip connection except that it is multiplied by the attention matrix.
>
> The $W_KW_Q^T + aI$ modification does not seem to have this interpretation. Instead, it encourages the attention head to use the matrix XX^T of the data, which encodes which symbols are repeated and where.

---

> ### Comment · Reviewer_gjp1 · 2023-11-22
>
> Thank you for the response. I want to emphasize that I'm still critical of the writeup and argue that the writing is over-claiming what is shown. For example, quoting the abstract, "we prove that Transformers can generalize to unseen symbols but require astonishingly large training data". This is not true, however---the proved result is for a simplified scenario. The authors' rebuttal, i.e., that they *believe* that their theoretical results can be extended is insufficient to make a broad claim. If they prefer to make such a claim, it is important to say that they prove the result for a simplified setup and *hypothesize* the proof can be extended further. Similarly, the argument that empirical results indicate practical models fail to perform variable binding is not appropriate to vindicate the statement that "we prove Transformers can generalize to unseen symbols but require astonishingly large training data".
>
> Overall, I stress that it is important that over-claiming is avoided. I sincerely hope the writing will be addressed in the final version of the paper and am choosing to keep my score as is assuming that the authors will make the change.

---

> ### Author Response · Authors · 2023-11-22
>
> Thank you for getting back to us. We will definitely make changes to the abstract + intro to make it clearer earlier on what exactly it is that we show. To summarize:
>
> * Our theory showing that transformers can learn to reason is for one-layer transformer (= self-attention + MLP). This theory
> shows that this simple architecture is already capable of learning these relational reasoning tasks.
>
> * Our experiments showing that transformers need a large number of samples to learn to reason are for multilayer transformers & even for non-pretrained GPT-2. Also, our theory indicates that the need for a large number of samples is due to the poor conditioning of the matrix N associated with the architecture & template task. (Note: Once you pretrain GPT-2, then it is requires many fewer samples to learn to reason, and we have some discussion on this in Section 6.)
>
> * To be more clear about our previous comment in this thread, we are fairly confident that we could extend our theory to show that
> multilayer transformers with residual connections learn, but the analysis would be messy & would not give qualitative insights beyond what we already have in this paper.
>
> So our revision will make these changes: In the abstract, we will specify that the theoretical achievability result is for one-layer transformers & that our observation on large # of samples is mainly empirical. In the introduction, we will also put the "one-layer" qualifier when appropriate. In general, we will edit to clarify what our theory proves on one-layer transformers, versus how it informs the experiments & transformer modifications, which are conducted for multi-layer transformers.

---

> > ### Comment · Reviewer_gjp1 · 2023-11-22
> >
> > Perfect, thank you for taking the comments into thorough consideration! I've increased my score accordingly.

---

> > > ### Author Response · Authors · 2023-11-22
> > >
> > > Thank you for adjusting your score. Let us know if you have any more questions.

---

### Official Review · Reviewer_5tFF · 2023-11-09

**Soundness:** 4 excellent
**Presentation:** 3 good
**Contribution:** 3 good
**Rating:** 8
**Confidence:** 4

**Summary:**

This work studies the ability / inability of transformers to learn certain symbolic reasoning tasks, specifically real valued (“a=+1, print(a)” -> +1) and symbolic valued (“a = `q’ , print(a)” => q) template tasks. To study whether models can learn a rule for an arbitrary new input symbols, they invoke a slightly different concept of generalization under distribution shift known as “generalization to the unseen.” This measures the intuitive notion of generalization of a rule to arbitrary inputs rather than inputs from the same distribution used for training (which is the usual notion of generalization in ML).  For both of these symbolic tasks, the authors argue that the permutation symmetry of multi-layer perceptrons (MLPs) trained with SGD prevents them from generalizing to unseen symbols (Appendix I). For transformer architectures, an analysis of the initial feature kernel under random Gaussian weights reveals that transformers can generalize on real-valued symbolic reasoning tasks by training the readouts only. The reason for this generalization follows from an approximation of the empirical kernel matrix as block diagonal with each block representing the inputs from a single template: K(x_a, x_{a’}) = K(x_b,x_{b’}) for x_a, x_b from template 1 and x_a’ , x_b’ from template 2. This approximation becomes increasingly valid as the “data diversity” metric introduced by the authors increases. This approximation allows the authors to work with a # template x # template gram matrix N, which they show to be non-singular. The importance of having a well-conditioned the N matrix suggests a possible improvement to transformers where an identity matrix is added to the weight innerproducts WK WQ -> WK Wq +  a * Identity, etc. The authors show this improves generalization in their task. For symbolic tasks, the authors show that the gradient descent updates are orthogonal to the directions which would reduce the loss when the embedding dimension is large. They provide experiments verifying that large embedding dimension worsens performance on copy tasks.

**Strengths:**

This paper studies an important problem of generalization to unseen symbols in transformer architectures. Much of the contribution is formalizing a set of symbolic reasoning tasks which can be analyzed in different architectures. The authors give a theoretical analysis that suggests implementation tweaks (adding the identity matrices) which improve performance on real-valued template tasks and preserve good performance on natural language modeling. They also give an important separation result between multi-layer perceptrons and transformer architectures and explain why the transformers can learn the real-valued regression tasks at large width. Lastly they observe that large embedding dimension can be harmful in transformers on copy tasks. I appreciate the effort the authors took in performing experiments (several in the Appendix) to evaluate their theoretical claims, which are often lacking in theory papers.

**Weaknesses:**

Though the paper provides some generic results about architectures for MLPs, the main result for transformers relies on operating the model in the kernel regime. It is unclear at the moment if any of the results would change (1) at finite width or (2) in the feature learning regime (see questions below). Further, the proposed theory bounds the generalization error in terms of the data diversity metric, which does not have an obvious scaling with samples n, making it hard to reason about the number of samples needed to attain good generalization.

**Questions:**

1. Do the sample complexity results / rank of N matrix change if you allow the network to learn its internal weights rather than just the readout weights?
2. Do the authors have any empirical evidence showing the emergence of this block diagonal structure in the NNGP kernel at initialization (I imagine this would be easy to compute)? The block-diagonal approximation seems reasonable, but some empirical support + visualization could be useful. A similar comparison of the predictor from kernel regression and the average of the labels over a template could also be useful.
3. Do the authors have a sense of how wide the transformer must be compared to samples n, templates r, etc for the theory to be accurate (N non-singular etc)? Wouldn’t the kernels become singular at some finite width? What if every layer is trained at large width (NTK vs NNGP)? Do real networks (which learn features) have to be as wide as kernel analysis would suggest?
4. The perplexity improvements on GPT + Wikitext seem rather small. Does this suggest that transformers trained in practice evolve to have well conditioned N matrices, removing the need to add an explicit identity matrix? Is this related to the pronounced diagonals in Wv Wo the authors report in the next section? Or is the idea that real data behaves differently than the proposed template tasks?
5. In Theorem 1.1 it may be helpful to clarify that the gradient flow is only performed on the last layer.


Small typos
1. Figure 3b caption should read W_V W_O^T  (transpose dropped in second writing)
2. Page 28 d/dx in the Wronskian should be d/dt

---

> ### Author Response · Authors · 2023-11-18
> **Response (1/2)**
>
> ## Thank you for your review
>
> Thank you for the in-depth reading of our paper and your insightful and constructive comments. We are glad that you found the problem setting significant, the new formalization interesting, and the separation with fully-connected networks important, and that you also appreciated the in-depth experimental analyses that support our claims. We address your questions and concerns item-by-item below.
>
> ## Weaknesses
> > Though the paper provides some generic results about architectures for MLPs, the main result for transformers relies on operating the model in the kernel regime. It is unclear at the moment if any of the results would change (1) at finite width or (2) in the feature learning regime (see questions below).
>
> Re: (1) finite width
>
> * We will add experiments in the appendix to the revision where we vary the width of the network.
>
> Re: (2) feature learning vs. kernel analysis
>
> * As far as we are aware, there are **no previous results that prove the ability of transformers to learn to perform symbolic reasoning tasks** when trained. Our paper is the first to formalize a framework of such reasoning tasks and to prove it is learned by transformers when trained.
>
>
> * It was quite **unexpected to us that the transformer kernel analysis is sufficient** to characterize when the relational reasoning tasks can be learned. We began this work with the mindset that we would have to study feature-learning dynamics. Nevertheless, with the kernel analysis, we were able to *characterize* when the relational reasoning can be learned by large transformers (it can be learned anytime there is no copying of placeholder symbols; it cannot be learned when there is).
>
>
> * **The analysis was considerably involved**: (i) This is not a typical kernel method analysis where it is proved that the data can be memorized – instead, we must show that the network *generalizes in the correct way out of distribution*, for any task in our framework. (ii) Also, insofar as we know, this is also the first paper that manages to analyze how the transformer kernel learns a nontrivial task, due to the technical difficulties involved in studying this kernel. The technical tools we develop here may thus be useful for future work.
>
>
> * Note also that **all known theoretical analyses of transformer dynamics use a combination of simplifications** such as: (a) removing the softmax to get a linear attention mechanism, (b) replacing the $W_KW_Q^T$ matrices with just one matrix $W_{KQ}$ and similarly for $W_VW_O^T$, (c) considering only one layer, (d) assuming diagonal weight matrices, (e) removing layernorms and residual connections.
>
>
>
> * We believe that our theory is relevant since it suggests modifications to the transformer that **improve data-efficiency on the reasoning tasks in practice by an order of magnitude**.
>
>
> * **Our work opens up many directions** that we feel are outside of the scope of the current paper. One direction is to study feature learning. We completely agree that it is an excellent question to ask: can we characterize if/when feature learning is more powerful than NTK? But, to ask this question, an NTK analysis as in this paper is needed first. In the revision, we will add experiments comparing feature learning to NTK training – these indicate that feature learning is intriguingly more data-efficient on the relational reasoning tasks (although NTK also learns given enough data).
>
> * On a more philosophical note, our results raise the question: to what extent is transformer reasoning due to NTK, and to what extent is it due to feature learning such as head specialization? This is a highly relevant question in the context of interpreting the internal workings of transformers. The more NTK-like the solution, the more polysemantic heads that the network will have, and so the harder it will be to interpret a network by extracting sparse subnetworks (which is currently the main method of the mechanistic interpretability literature).
>
>
> > Further, the proposed theory bounds the generalization error in terms of the data diversity metric, which does not have an obvious scaling with samples n, making it hard to reason about the number of samples needed to attain good generalization.
>
> A large number of samples n is not enough, because all n samples could be copies of each other, in which case the algorithm will not be able to generalize beyond the training data distribution. Therefore, it is necessary to have a control on the data diversity. We opt in our case to have a parameter rho which measures diversity, but otherwise we are agnostic to the data distribution.

---

> > ### Author Response · Authors · 2023-11-18
> > **Response (2/2)**
> >
> > ### Questions
> > > 1. Do the sample complexity results / rank of N matrix change if you allow the network to learn its internal weights rather than just the readout weights?
> >
> > Thanks for the question – we will put discussion of this in the paper: Our result would not change; what we have proved about NNGP is a strictly stronger statement than the corresponding statement if all layers are trained together.
> >
> > > 2. Do the authors have any empirical evidence showing the emergence of this block diagonal structure in the NNGP kernel at initialization (I imagine this would be easy to compute)? The block-diagonal approximation seems reasonable, but some empirical support + visualization could be useful. A similar comparison of the predictor from kernel regression and the average of the labels over a template could also be useful.
> >
> > As we promised above, we will add experiments for feature learning vs. NTK. These show that the NTK does succeed albeit with more samples than in feature learning. We will also add explicit approximation of the matrix N for the same-different task before/after the addition of the $W_KW_Q^T$ identity perturbations, illustrating how the condition number is vastly improved by our modifications.
> >
> > > 3. Do the authors have a sense of how wide the transformer must be compared to samples n, templates r, etc for the theory to be accurate (N non-singular etc)?  Wouldn’t the kernels become singular at some finite width? What if every layer is trained at large width (NTK vs NNGP)? Do real networks (which learn features) have to be as wide as kernel analysis would suggest?
> >
> > This is an excellent question for future work, but is outside of the scope of our current paper. The width and number of samples would depend on the particular number of templates and potentially on the structure of these templates. In this setting with quantitative bounds, we could hope for a separation between training just readout vs. also training the internal weights. However, we do not presently know how this would work. We will add discussion of this question in the final section.
> >
> > > 4. The perplexity improvements on GPT + Wikitext seem rather small. Does this suggest that transformers trained in practice evolve to have well conditioned N matrices, removing the need to add an explicit identity matrix? Is this related to the pronounced diagonals in Wv Wo the authors report in the next section? Or is the idea that real data behaves differently than the proposed template tasks?
> >
> > * We are adding a total of 288 parameters to a 117M parameter architecture, which is a minuscule fraction of the total number of parameters. Per parameter, the drop in perplexity is significant. The transformer architecture is state-of-the-art and ubiquitous, so we believe that any improvement can be impactful.
> >
> > * For comparison, for this rebuttal we ran GPT-2 medium (345M parameters) on Wikitext103 for 5 epochs on a V100. This takes 2.14 days and gets a perplexity of 17.904. Stopping GPT-2 small at those same 2.14 days gives a perplexity of 17.08, and stopping GPT-2 small plus the modifications for 2.14 days gives a perplexity of 16.693. Despite the fact that GPT-2 medium has 228M more parameters it actually underperforms because with the same computational budget it has not yet converged (we are continuing to run this comparison).
> >
> > * Finally, you are correct that Wikitext is not a reasoning task, so seeing any significant improvements at all is a welcome surprise. We will try to benchmark our modifications on more pure reasoning tasks in the revision if we have time, but we do recall that this is not the main point of the paper, which is to theoretically characterize when transformer architectures can learn to perform symbol binding reasoning.
> >
> >
> > * > 5. In Theorem 1.1 it may be helpful to clarify that the gradient flow is only performed on the last layer.
> >
> > Thanks for the suggestion. It is in the formal statement, and in the revision we will add it to the informal statement.
> >
> > * >Typos
> >
> > Thank you, we will correct these.

---

> ### Comment · Reviewer_5tFF · 2023-11-18
> **Response to Rebuttal**
>
> I thank the authors for their detailed responses to all of my questions. I really appreciate the novelty and analysis of this paper so I am in favor of acceptance and will raise my score.
>
> With the promised additional numerical tests, I think this paper would be a nice addition to ICLR.

---

> > ### Author Response · Authors · 2023-11-18
> > **Thank you**
> >
> > Thank you for adjusting your score. Do let us know if you have any more questions.

---

### Meta-Review · Area_Chair_63Md · 2023-12-10

**Metareview:**

The submission provides a careful formalization of an important and interesting capacity of neural nets, namely symbolic generalization abilities, and investigates the impact of both data and architecture on this capacity. The paper's claims are evidenced by both theory and experiments with a rare degree of complementarity in this line of research, and the empirical investigation includes both an observational study and a counterfactual reparameterization. This paper will be an excellent contribution to the conference.

**Justification For Why Not Higher Score:**

scope: limited to the lazy learning regime, and thus may be limited in accounting for feature learning in standard nets

**Justification For Why Not Lower Score:**

solid theoretical and empirical contributions

---

### Decision · Program_Chairs · 2024-01-16

Accept (poster)